# $I^2$-NeRF: Learning Neural Radiance Fields Under Physically-Grounded Media Interactions

**Shuhong Liu[1], Lin Gu[2,†], Ziteng Cui[1], Xuangeng Chu[1], Tatsuya Harada[1,3]**
[1]The University of Tokyo, [2]Tohoku University, [3]RIKEN
{s-liu,lingu,cui,xuangeng.chu,harada}@mi.t.u-tokyo.ac.jp

## Abstract

Participating in efforts to endow generative AI with the 3D physical world perception, we propose $I^2$-NeRF, a novel neural radiance field framework that enhances isometric and isotropic metric perception under media degradation. While existing NeRF models predominantly rely on object-centric sampling, $I^2$-NeRF introduces a reverse-stratified upsampling strategy to achieve near-uniform sampling across 3D space, thereby preserving isometry. We further present a general radiative formulation for media degradation that unifies emission, absorption, and scattering into a particle model governed by the Beer–Lambert attenuation law. By composing the direct and media-induced in-scatter radiance, this formulation extends naturally to complex media environments such as underwater, haze, and even low-light scenes. By treating light propagation uniformly in both vertical and horizontal directions, $I^2$-NeRF enables isotropic metric perception and can even estimate medium properties such as water depth. Experiments on real-world datasets demonstrate that our method significantly improves both reconstruction fidelity and physical plausibility compared to existing approaches. The source code is available at https://github.com/ShuhongLL/I2-NeRF.

## 1 Introduction

Recent breakthroughs in generative models [20, 12, 59] have sparked growing belief that Artificial General Intelligence (AGI) may be within reach. However, these models primarily operate on virtual data and often lack a grounded understanding of space, time, and causality [24, 1, 28, 50], limiting their applicability to real-world tasks [62]. To bridge this gap, World Models [30, 29, 65, 31] have been introduced in navigation [5], multi-task control [32], long-horizon planning [73, 47], and robot embodiment [78]. Incorporating physical constraints has also been shown to enhance continuous-time prediction [37], enable physically-consistent generation [9, 53], and facilitate the development of foundation models for simulation, education, entertainment, and embodied AI [19].

In this work, we focus on integrating two fundamental physical principles—**isometry** and **isotropy**—into Neural Radiance Fields (NeRF), particularly for modeling volumetric media outside the object. Classical NeRF [56] and its variants [6, 8] assume a clear-air medium, where no radiance is sampled or accumulated between the camera and the object surface. While this assumption simplifies computation, it discards nearly 93.36%[1] of the 3D volume, thereby failing to capture the full metric structure of the space. Recent efforts [16, 81] relax this assumption by introducing virtual attenuation in the surrounding air to simulate illumination effects, enabling robustness under low-light and overexposed conditions. In parallel, [45, 63] address non-air media by incorporating absorption and

---

[†]Corresponding author.

[1]Average space occupancy ratio across eight synthetic indoor scenes from the Replica dataset [67]. The ratio is computed as ground-truth meshes divided by the total grid size.

39th Conference on Neural Information Processing Systems (NeurIPS 2025).

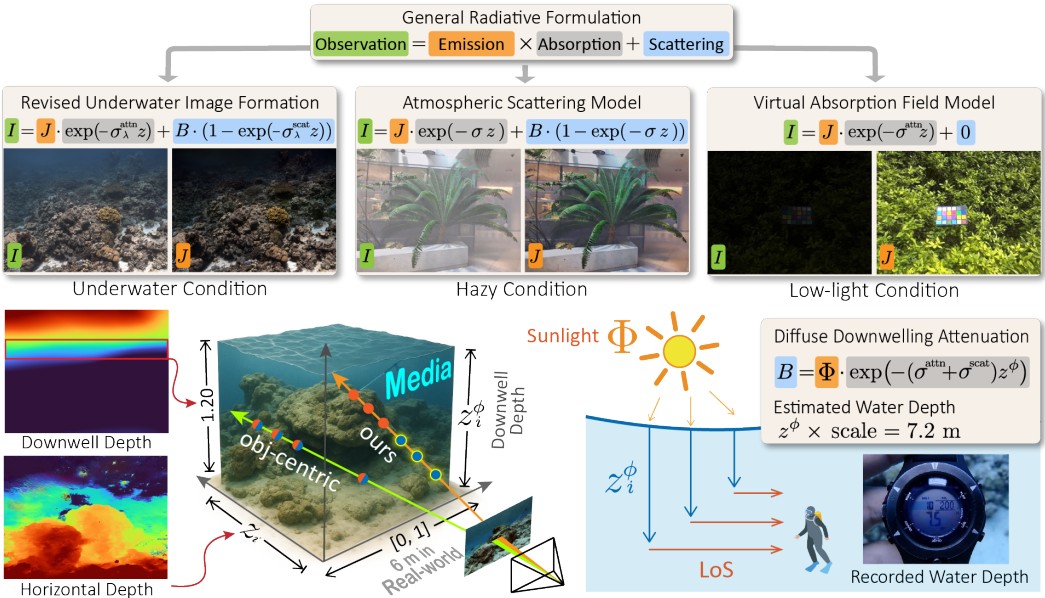

Figure 1: The proposed general radiative formulation accommodates diverse media conditions, including underwater, hazy, and low-light environments. The scene shown here was captured in the Pacific Ocean near Okinawa. As a result of isometric and isotropic metric perception, the estimated depth closely matches the actual depth of the scene.

scattering based on the Beer–Lambert Law and Henyey–Greenstein phase functions. However, they do not explicitly disentangle object geometry from volumetric media, limiting interpretability and physical fidelity.

Existing NeRF-based methods predominantly rely on object-centric sampling, which places most sampling points near high-density object surfaces. As shown in the bottom row of Figure 1, SeaThru-NeRF [45] concentrates samples around rock surfaces while leaving large portions of the surrounding medium undersampled. This imbalance limits the representation of the full 3D volume and compromises spatial consistency. To address this, we propose a reverse-stratified upsampling strategy that explicitly allocates samples within the medium. Our method adopts a two-phase process: we first perform standard sampling in high-density regions, such as the rocks illustrated in orange. We then compute a reverse weight of the object density to guide stratified sampling in low-density regions, placing additional points in the semi-transparent medium, shown as blue points. This enforces the isometric property of the radiance field by ensuring that spatial sampling reflects the actual spatial extent of the scene geometry. By treating the medium as an equal component of the radiance field, our approach enables a more faithful mapping between the physical real-world and the radiance space.

Building on the isometry-preserving particle sampling, we further introduce a general radiative formulation. This formulation models the observation $I$ as the composition of attenuated direct radiance $J$ and backscattered ambient radiance $B$, as illustrated at the top of Figure 1. It unifies emission, absorption, and scattering in a direction-independent manner and generalizes naturally to a variety of degradation conditions. In underwater scenes, the coefficients $\sigma_\lambda^{\mathrm{attn}}$ and $\sigma_\lambda^{\mathrm{scat}}$ governing direct and backscattered radiance differ and exhibit wavelength dependence reflecting the spectral sensitivity of the camera. When absorption and scattering share a common coefficient $\sigma$, the formulation reduces to classical atmospheric scattering models [60]. Remarkably, this formulation can even explain low-light conditions by introducing a virtual absorption medium, where darkness emerges from radiance attenuation along the viewing direction.

Moreover, existing scattering-aware NeRFs [45, 63, 14, 83] perform radiative modeling only along the horizontal line of sight (LoS), neglecting downwelling attenuation from ambient illumination. This effect is particularly important in media, where the radiance at each point physically depends on its vertical depth relative to the medium surface. We address this by explicitly modeling downwelling attenuation at vertical depth $z^\Phi$ for each ray. This allows the Beer–Lambert Law to apply uniformly

in both horizontal and vertical directions, forming an isotropic radiative model. As a result, our method enables metric perception across the full 3D space. For instance, with a known scale, our model estimates the water depth in Figure 1 as 7.2 meters, closely matching the real depth, 7.5 meters, shown on the diving watch.

With the general radiative formulation established, we apply our model to various media-degraded applications, including low-light, hazy, and underwater scenes. On the low-light benchmark [16], our method achieves state-of-the-art (SOTA) performance, outperforming Gaussian Splatting–based approaches [17, 84] by approximately 2 dB in PSNR. For haze and underwater datasets [45], we deliver competitive results that closely match current SOTA methods. These improvements stem directly from the integration of isometry and isotropy as core physical principles within the radiance field. Unlike previous methods that rely on heuristic tuning or pixel-wise manipulation, our model ensures high-fidelity reconstruction while preserving physical plausibility.

Overall, we make the following contributions:

- We propose a general radiative formulation that unifies the modeling of underwater, hazy, and low-light environments through an isotropic modeling.

- We introduce a novel reverse-stratified upsampling strategy that enables metric-preserving sampling in NeRF space. Combined with absorption modeling, our method extends spatial representation beyond the one-dimensional line of sight to recover real-world scene structure.

- Experiments on real-world datasets demonstrate that $I^2$-NeRF substantially improves both reconstruction fidelity and physical plausibility compared to existing approaches.

## 2   Related Works

**Neural Radiance Field**   Neural Radiance Fields (NeRF) [56] and 3D Gaussian Splatting (3DGS) [40] have achieved remarkable performance in novel-view synthesis (NVS). Building on this, physically grounded formulations have been introduced to improve fidelity and robustness under real-world conditions. For realistic image formation, [64] reinterprets neural fields probabilistically to simulate lens effects, while PAC-NeRF [49] adopts a hybrid Eulerian-Lagrangian framework for dynamic media. Mip-NeRF and its successors [6–8] reduce aliasing via conical frustum sampling. Ref-NeRF [72] and PBR-NeRF [79] model view-dependent reflectance using BRDFs. DP-NeRF [43] and Deblurring-GS [42] address motion and defocus blur. Atmospheric scattering models are used in [63, 70, 14, 83] to dehaze scenes. RawNeRF [57] handles low-light inputs in linear space, while LL-NeRF [74] and LL-GS [68] apply Retinex-based decomposition. Aleth-NeRF [16] introduces a concealing field, and Luminance-GS [17] and LITA-GS [84] leverage illumination-adaptive priors. In underwater settings, SeaThru-NeRF [3], Proposed-T [69], and Watersplatting [46] model wavelength-dependent attenuation and backscatter. NeuroPump [27] accounts for lens refraction to improve realism.

**Single-Image Restoration**   Recent progress in single image restoration tackles the challenges of low-light, haze, and underwater conditions through deep learning architectures and novel learning strategies. Low-light enhancement has shifted toward end-to-end learning [77, 71, 38, 21], utilizing techniques such as direct curve estimation [26, 15] and self-calibration [54] to improve visibility in darkness. Haze removal has evolved from prior-based methods [33] to data-driven approaches with a focus on handling complex atmospheric degradations [13, 10, 48, 18]. Underwater image restoration addresses unique optical distortions using generalized priors [23, 11, 51], physics-based modeling [61, 3], and unsupervised techniques [22, 52, 34].

## 3   Isometry and Isotropy NeRF

**Preliminaries**   NeRF [56] implicitly represents a scene with a continuous function $f_\Theta : (\mathrm{x}, \mathrm{d}) \to (\mathrm{c}, \sigma)$ that encodes the 3D position $\mathrm{x} \in \mathbb{R}^3$ and viewing direction $\mathrm{d}$ to predict the density $\sigma$ and view-dependent color $\mathrm{c} = (r, g, b)$. This color $\mathrm{c}$ is interpreted as the radiance emitted by the objects along direction $\mathrm{d}$. For a ray $r(t) = \mathrm{o} + t\mathrm{d}$ emitted from the center of a camera $\mathrm{o}$ and range $t \in \mathbb{R}^+$, NeRF uses numerical quadrature to approximate the integral of volume rendering as

$\hat{C}(r) = \sum_{i=1}^{N} \hat{C}_i(r) = \sum_{i=1}^{N} T(r_i)(1 - \exp(-\sigma(r_i)\delta_i))c(r_i)$, where $T(r_i) = \exp(-\sum_{j=1}^{i-1} \sigma(r_j)\delta_j)$ is the cumulative transmittance, and $\delta_i$ is the distance between two points.

## 3.1 General Radiative Formulation via Matting

Traditional image matting [44] refers to the process of compositing a target image $I$ from a foreground $J$ and a background component $B$, using a blending coefficient $\alpha \in [0, 1]$. In this classical problem, $\alpha$ is typically estimated empirically and determines the mixing ratio between $J$ and $B$ at each pixel.

In participating media such as fog, haze, or water, light is subject to volumetric scattering, blur, and exponential attenuation. These effects result in an observed view that also arises from the combination of two components: the direct scene radiance $J$ and the ambient backscatter light $B$. This similarity in structure allows us to reinterpret media-induced degradation as a physically grounded extension of matting. In our isotropic radiative formulation, the blending coefficient $\alpha$ is no longer an abstract mixing weight but a transmittance function derived from the Beer–Lambert Law, relative to the scene depth $z$. This yields a general radiative formulation as:

$$I = J \cdot \alpha + B \cdot (1 - \alpha) \rightarrow I = J \cdot \exp(-\sigma^{\text{attn}}z) + B \cdot (1 - \exp(-\sigma^{\text{scat}}z)). \quad (1)$$

The coefficients $\sigma^{\text{attn}}$ and $\sigma^{\text{scat}}$ respectively encode the medium's direct extinction and in-scattering properties under the single-scattering assumption.

**Haze Condition**   Haze arises from aerosol particles in the atmosphere, causing both scattering and absorption. Koschmieder's Law [41] introduces a single extinction coefficient, implicitly assuming equal scattering and absorption contributions. In our general formulation, setting $\sigma^{\text{attn}} = \sigma^{\text{scat}}$ regresses Equation (1) to the classical Atmospheric Scattering Model (ASM).

**Underwater Condition**   In underwater scattering, scattering and absorption differ due to distinct optical mechanisms. Scattering depends on particle size and concentration, while absorption is influenced by molecular composition and the inherent optical properties of water [36, 58]. Therefore, we define $\sigma^{\text{attn}} \neq \sigma^{\text{scat}}$ and generalize Equation (1) to the Revised Underwater Image Formation (RUIF) model [2].

**Low-Light Condition**   Underexposed images can be interpreted through a simplified pixel-scaling model $I = K \cdot J$, where $J$ represents the signal of well-lit scenes and $K$ is a per-pixel scaling factor. To incorporate this model into our radiative formulation, we introduce a **virtual absorption medium** that attenuates the photon flux reaching the sensor. This corresponds to a special case of Equation (1), where the backscatter component $B = 0$ and the scaling factor $K = \exp(-\sigma^{\text{attn}}z)$ gains a physical interpretation as the transmittance through the medium. Further details are provided in Appendix A.3.

## 3.2 Proposed General Particle Model

In our neural radiance model, we explicitly represent both the objects and the intervening medium as volumetric particles characterized by two separate spatial density fields. As shown in Figure 2, these particles may emit, absorb, or scatter light according to the same radiative transfer rules [39].

**Emission**   Under clear-air conditions, only object particles of density $\sigma_j^{\text{obj}}$ emit radiance directly toward the camera. Therefore, the volume rendering formation of direct clear randiance $J$ is identical to the vanilla NeRF:

$$\hat{J} = \sum_{i=1}^{N} \hat{C}_i^{\text{obj}} = \sum_{i=1}^{N} T_i(1 - \exp(-\sigma_i^{\text{obj}}\delta_i))c_i^{\text{obj}}, \text{ where } T_i = \exp(-\sum_{j=1}^{i-1} \sigma_j^{\text{obj}}\delta_j). \quad (2)$$

**Absorption**   Absorption reduces transmitted radiance without re-emitting energy. The received degraded radiance $I$ at the camera is attenuated by the absorption density $\sigma^{\text{attn}}$:

$$\hat{I} = \sum_{i=1}^{N} \hat{C}_i^{\text{obj}} = \sum_{i=1}^{N} T_i^D(1 - \exp(-\sigma_i^{\text{obj}}\delta_i))c_i^{\text{obj}}, \text{ where } T_i^D = \exp(-\sum_{j=1}^{i-1} (\sigma_j^{\text{obj}} + \sigma_j^{\text{attn}})\delta_j). \quad (3)$$

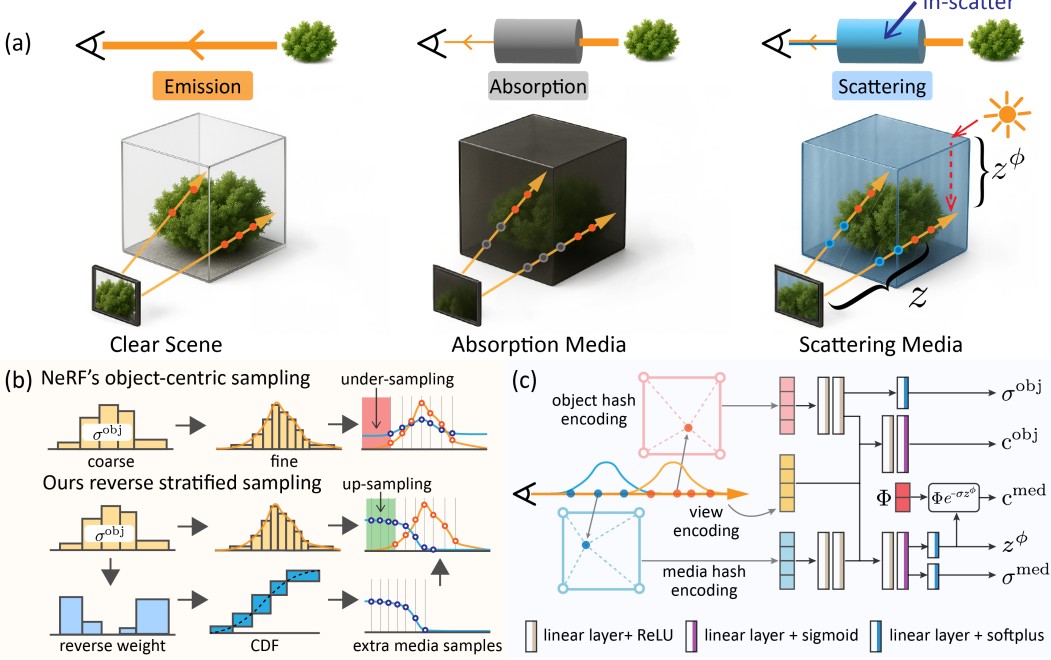

Figure 2: Illustration of our particle model and pipeline. (a) depicts three types of particles in the media field that emit, absorb, or scatter radiance. (b) presents the media upsampling strategy designed to prevent media field collapse. (c) shows the model architecture, in which the object and the upsampled media points are processed by hash encoders followed by separate MLPs to predict density, color, and downwelling depth (in the case of scattering).

**Scattering** Particles such as suspended aerosols in haze or colloidal particles in water can contribute additional backscattering radiance $\hat{C}^{\text{med}}$ by scattering ambient illumination into the viewing direction. Under single-scattering approximation, the scattered radiance from each sampled point $i$ along the ray is:

$$\hat{C}_i^{\text{med}} = T_i^B(1 - \exp(-\sigma_i^{\text{scat}}\delta_i))c_i^{\text{med}}, \text{ where } T_i^B = \exp(-\sum_{j=1}^{i-1}(\sigma_j^{\text{obj}} + \sigma_j^{\text{scat}})\delta_j) \qquad (4)$$

Here, $\sigma^{scat}$ represents the scattering coefficient, $c^{med}$ denotes the backscattered color contributed by ambient illumination, and $T^B$ is the accumulated backscatter transmittance. Building upon our isotropic radiative formulation, we further model the backscatter color $c_i^{med}$ via Beer–Lambert attenuation of sunlight traveling vertically from the media surface. This approach reflects empirical observations, such as the progressive darkening with increasing water depth. At a sampled point $i$, the in-scattered medium color can be modeled as:

$$c_i^{\text{med}} = \Phi \cdot \exp\left(-(\sigma_i^{\text{attn}} + \sigma_i^{\text{scat}})z_i^{\phi}\right), \qquad (5)$$

where $\Phi$ is a learnable sunlight constant [2], $z^{\phi}$ denotes the downwelling distance to the medium surface, and $\sigma^{\text{attn}} + \sigma^{\text{scat}}$ represents the diffuse attenuation coefficient [66]. This formulation further disentangles the formerly view-dependent medium color into an explicit vertical attenuation model, enabling our radiance field to perceive the scene geometry both horizontally through LoS and vertically. Because $z^{\phi}$ is expressed in the same metric units as the NeRF sampling space, our model maintains an isometric relationship to real-world distances through a global scale factor. Although this scattering model allows per-sample specification of $\sigma_i^{\text{attn}}$, $\sigma_i^{\text{scat}}$, and $z_i^{\phi}$, such fine-grained parameterization significantly enlarges the optimization space. To make the problem tractable while preserving physical interpretability, we adopt the simplified RUIF model [2] that assumes horizontal rays with constant scattering coefficients and downwelling depth along each ray, as further described in Section 4. The detailed derivation of Equation (5) is shown in Appendix C.

---

[2]Equal to the empirical measurement of irradiance $E(z, \lambda)$ at the medium surface ($z = 0$).

**Final Model**  Considering all particle behaviors, the total radiance $I$ received by the camera comprises the sum of direct radiance and potential in-scattered components:

$$\hat{\mathrm{I}} = \sum_{i=1}^{N} \hat{\mathrm{C}}_i^{\mathrm{obj}} + \sum_{i=1}^{N} \hat{\mathrm{C}}_i^{\mathrm{med}} = \sum_{i=1}^{N} T_i^D (1 - \exp(-\sigma_i^{\mathrm{obj}} \delta_i)) \mathrm{c}_i^{\mathrm{obj}} + \sum_{i=1}^{N} T_i^B (1 - \exp(-\sigma_i^{\mathrm{scat}} \delta_i)) \mathrm{c}_i^{\mathrm{med}}. \quad (6)$$

In Appendix B, we demonstrate that our volume rendering formulation in Equation (6) is equivalent to the generalized image formation in Equation (1) across all degradation scenarios.

### 3.3 Reverse-Stratified Upsampling

Original NeRF [56] and its variants [63, 45, 16] employ a hierarchical coarse-to-fine sampling scheme [56, 6, 7] designed to capture object densities in clear-air conditions. In real-world scenes, however, media particles reside exclusively along the LoS in front of objects, absorbing or scattering incoming light before it reaches the camera. This object-centric sampling thus under-samples the medium volume, which can cause the medium field to collapse into the object color during decomposition.

To realize the surrounding media in the NeRF model, we propose a novel reverse-stratified upsampling (RSU) strategy. As illustrated in Figure 2, our method upsamples media particles near-uniformly in regions where the object density is low or zero. Specifically, to sample media positions outside the solid object, we perform an additional sampling phase tailored specifically for media points. In this phase, each current interval $t_i^{\mathrm{obj}}$ is assigned a reverse weight derived from the object density $\sigma^{\mathrm{obj}}$ that favors low-density gaps:

$$w_i^{\mathrm{med}} = \Delta_i \big( \max_{1 \leq j \leq N} \hat{\sigma}_j^{\mathrm{obj}} - \hat{\sigma}_i^{\mathrm{obj}} \big) + \epsilon, \qquad \hat{\sigma}_i^{\mathrm{obj}} = \frac{1}{\Delta_i} \int_{t_i^{\mathrm{obj}}}^{t_{i+1}^{\mathrm{obj}}} \sigma^{\mathrm{obj}}(t) \, \mathrm{d}t, \quad (7)$$

where $\Delta_i = t_{i+1}^{\mathrm{obj}} - t_i^{\mathrm{obj}}$ is the interval length, and $\epsilon$ is a small constant to prevent zero weight. Next, we normalize $\{w_i^{\mathrm{med}}\}$ into a cumulative distribution function (CDF) $F$ and perform stratified sampling over $[0, 1]$ to draw $N_{\mathrm{add}}$ new media points:

$$u_j \sim \mathcal{U}\big( \tfrac{j-1}{N_{\mathrm{add}}}, \tfrac{j}{N_{\mathrm{add}}} \big), \quad t_j^{\mathrm{med}} \sim \mathcal{U}\big( t_i^{\mathrm{obj}}, t_{i+1}^{\mathrm{obj}} \big) \quad \text{whenever } F(i-1) < u_j \leq F(i), \quad (8)$$

where $\mathcal{U}$ denotes uniform distribution. Finally, we merge these new media samples with the original object samples $\{t_i^{\mathrm{obj}}\}$, sort the union $\{t_k\}$, and obtain the upsampled set of ray-marching positions.

### 3.4 Objective Functions

We employ the **reconstruction loss** $\mathcal{L}_{\mathrm{recon}}$ proposed by RawNeRF [57], which integrates inherent tone-mapping to reduce sensitivity to scale variations during loss computation as:

$$\mathcal{L}_{\mathrm{recon}} = \frac{1}{M} \sum_{k=1}^{M} \left( \frac{\hat{\mathrm{I}}_k - \mathrm{I}_k}{\mathrm{sg}(\hat{\mathrm{I}}_k) + \epsilon} \right)^2, \quad (9)$$

where $\mathrm{sg}(\cdot)$ denotes the stop-gradient operator, $M$ is the number of pixels, and $\epsilon = 10^{-3}$ to avoid division by zero. To ensure geometric consistency, we introduce a **geometry loss** $\mathcal{L}_{\mathrm{geo}}$ between the normalized rendered depth $\hat{\mathrm{D}}_k$ and pseudo-depth map $\tilde{\mathrm{D}}_k$ obtained from a pretrained model [82] using:

$$\mathcal{L}_{\mathrm{geo}} = \frac{1}{M} \sum_{k=1}^{M} \left\| \hat{\mathrm{D}}_k - \tilde{\mathrm{D}}_k \right\|^2, \quad (10)$$

To effectively recover the clean radiance structure $\hat{J}$ from the degraded observation $I$, we utilize a **compensated structure similarity loss** $\mathcal{L}_{\mathrm{comp}}$ that accounts for ideal illuminance $\tilde{\nu}_J$ and contrast factor $\tilde{\kappa}_J$. This loss is calculated over $H$ stochastic ray sub-patches following [80], expressed as:

$$\mathcal{L}_{\mathrm{comp}} = \frac{1}{H} \sum_{h=1}^{H} \mathcal{L}_{\mathrm{SSIM}}(\mathcal{P}_h(\hat{J}), \mathcal{P}_h(I); \tilde{\nu}_J, \tilde{\kappa}_J), \quad (11)$$

where $\mathcal{P}_h$ denotes the $h$-th randomly extracted patch. Each patch-wise loss employs the SSIM index [76], adjusted for the desired mean and variance.

Considering the physical constraint that space is occupied either by objects or media, we apply a **mutual exclusion loss** $\mathcal{L}_{\mathbf{mutex}}$ to prevent simultaneous high densities of object and media at the same spatial location:

$$\mathcal{L}_{\mathrm{mutex}} = \frac{1}{N} \sum_{i=1}^{N} (\max(0, \sigma_i^{\mathrm{obj}} - \eta) \times \sigma_i^{\mathrm{med}}), \tag{12}$$

where $N$ is the number of sampled points in a ray, and the threshold $\eta = 0.1$ helps avoid the trivial solution of zero media. Moreover, if a media transmittance prior $\tilde{T}$ is available, we define the **media transmittance loss** $\mathcal{L}_{\mathbf{media}}$ to constrain media density at the object's surface depth $z$:

$$\mathcal{L}_{\mathrm{media}} = \frac{1}{M} \sum_{k=1}^{M} |T_k - \tilde{T}_k|^2 = \frac{1}{M} \sum_{k=1}^{M} | \exp(-\sum_{j}^{z_k} (\sigma_j^{\mathrm{obj}} + \sigma_j^{\mathrm{med}}) \delta_j) - \tilde{T}_k|^2. \tag{13}$$

Additionally, to ensure a physically plausible monotonic decay in media density, which is particularly critical in nonhomogeneous media, we introduce a monotonic decay loss $\mathcal{L}_{\mathrm{mono}} = \frac{1}{MN} \sum_{k=1}^{M} \sum_{i=1}^{N} \mathrm{ReLU}(\sigma_{k,i}^{\mathrm{med}} - \sigma_{k,i-1}^{\mathrm{med}})$. Thus, the **total media transmittance regularization loss** is defined as $\mathcal{L}_{\mathrm{trans}} = \mathcal{L}_{\mathrm{media}} + \mathcal{L}_{\mathrm{mono}}$. The complete optimization objective is thereby expressed as:

$$\mathcal{L} = \mathcal{L}_{\mathrm{recon}} + \lambda_{\mathrm{comp}} \mathcal{L}_{\mathrm{comp}} + \lambda_{\mathrm{geo}} \mathcal{L}_{\mathrm{geo}} + \lambda_{\mathrm{mutex}} \mathcal{L}_{\mathrm{mutex}} + \lambda_{\mathrm{trans}} \mathcal{L}_{\mathrm{trans}}, \tag{14}$$

where the hyperparameters $\lambda_{\mathrm{comp}}$, $\lambda_{\mathrm{geo}}$, $\lambda_{\mathrm{mutex}}$, and $\lambda_{\mathrm{trans}}$ are specific to application scenarios. Media density $\sigma^{\mathrm{med}}$ in the above context subject to $\sigma^{\mathrm{attn}}$ or $\sigma^{\mathrm{scat}}$, as applicable. Detailed derivations and justifications for the loss components are provided in Appendix E.

## 4 Applications

**Underwater Scene**  To reflect the near-homogeneous distribution of water particles and make the computation of scattering radiance tractable, we adopt the RUIF model [2] which assumes horizontally propagating rays (i.e., an inclination angle of $90°$ in the nadir direction) in Equation (4). Under this simplification, the attenuation and scattering coefficients are treated as constant along each ray $r$, i.e., $\sigma_j^{\mathrm{attn}}(r) = \sigma^{\mathrm{attn}}(r)$ and $\sigma_j^{\mathrm{scat}}(r) = \sigma^{\mathrm{scat}}(r)$. Consequently, each point along the ray is approximated as having an identical downwelling depth $z_i^{\phi}(r) = z^{\phi}(r)$ and media color $c_i^{\mathrm{med}}(r) = c^{\mathrm{med}}(r)$. The clean radiance $J$ is obtained by rendering the scene with the scattering field removed. The model is optimized by minimizing Equation (14), with the $\lambda$ hyperparameters set to 0, $10^{-2}$, $10^{-4}$, and 0, respectively.

**Hazy Scene**  We apply the identical settings and assumptions used for underwater scattering to the case of air-based scattering. The model is optimized using Equation (14), with the $\lambda$ hyperparameters set to 0, $10^{-2}$, $10^{-4}$, and 0, respectively.

**Low-light Scene**  In low-light conditions, spatially varying illumination necessitates modeling the absorption medium as nonhomogeneous along each ray. To regularize this inherently ill-posed setting, we incorporate an external supervision signal $T_{\mathcal{P}}$ derived from the Bright Channel Prior (BCP) [75]. BCP estimates a 2D illumination map $T_{\mathcal{P}}$ from the bright channel of each local patch $\mathcal{P} \subseteq I$, and models the low-light image as:

$$I = T_{\mathcal{P}} \cdot J, \text{ and } T_{\mathcal{P}} = 1 - \max_{c \in (r,g,b)} \left( \max_{q \in \mathcal{P}} \left( \frac{1 - I_q^c}{1 - B^c} \right) \right). \tag{15}$$

Within our volumetric framework, where attenuation is solely attributed to the absorption field, this illumination

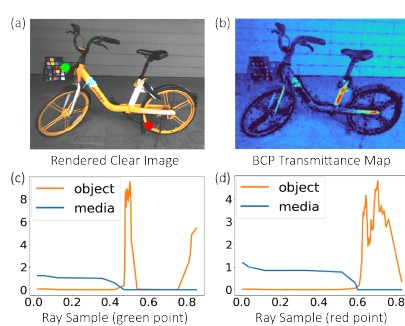

Figure 3: Visualization of virtual absorption fields on the LOM dataset [16]: (a) restored clean radiance $J$; (b) illumination map $T_{\mathcal{P}}$ estimated unsupervisedly via the BCP; (c,d) Particle density profiles along the rays, corresponding to the green and red points illustrated in (a).

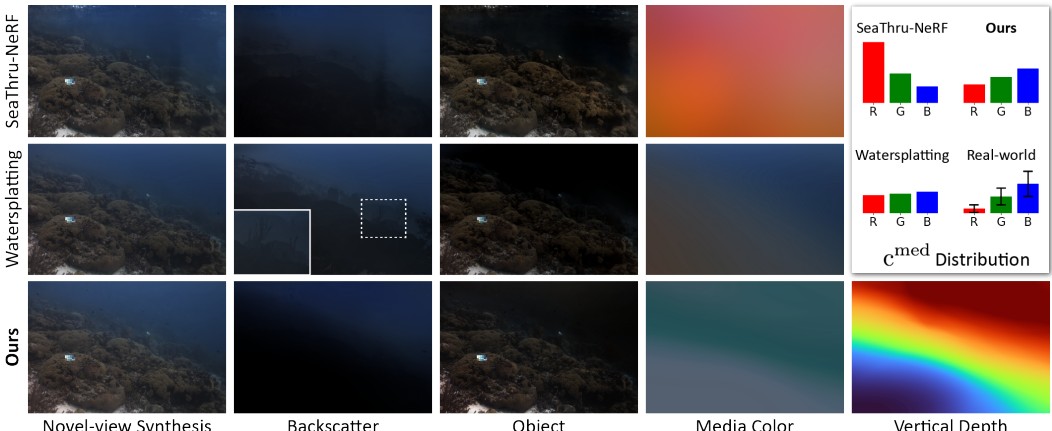

Figure 4: Visualization of NVS and media decomposition on the SeaThru-NeRF dataset [45]. Our method precisely estimates the vertical water depth, which is inaccessible in baseline methods. Top-right corner displays the mean RGB values of the medium color $c^{med}$ from test views, alongside the real-world ranges simulated in [4] across various Jerlov water types [35] within depths 1-20 meters.

Table 1: Quantitative comparison of hazy and underwater scenes on SeaThru-NeRF dataset [45].

| Model | "Hazy Fern" PSNR | SSIM | LPIPS | "Curaçao" PSNR | SSIM | LPIPS | "IUI3 Red Sea" PSNR | SSIM | LPIPS | "J.G. Red Sea" PSNR | SSIM | LPIPS | "Panama" PSNR | SSIM | LPIPS |
|---|---|---|---|---|---|---|---|---|---|---|---|---|---|---|---|
| MipNeRF360 [7] | 30.23 | 0.880 | 0.150 | 28.23 | 0.683 | 0.571 | 19.55 | 0.510 | 0.520 | 19.62 | 0.624 | 0.492 | 18.32 | 0.556 | 0.595 |
| ZipNeRF [8] | 30.34 | 0.875 | 0.142 | 19.96 | 0.442 | 0.421 | 16.94 | 0.474 | 0.412 | 19.02 | 0.349 | 0.483 | 19.01 | 0.349 | 0.482 |
| 3DGS [40] | 25.96 | 0.782 | 0.303 | 28.31 | 0.873 | 0.221 | 22.98 | 0.843 | 0.249 | 21.49 | 0.854 | 0.216 | 29.20 | 0.893 | 0.152 |
| SeaTh.NeRF [45] | 30.75 | 0.870 | 0.160 | 30.96 | 0.915 | 0.133 | 26.76 | 0.826 | 0.168 | 23.28 | 0.876 | 0.111 | 31.28 | 0.937 | 0.071 |
| Proposed-T [69] | - | - | - | 30.03 | 0.828 | 0.238 | 22.70 | 0.624 | 0.348 | 25.81 | 0.853 | 0.183 | 23.75 | 0.687 | 0.263 |
| Watersplat. [46] | 29.35 | 0.880 | 0.181 | 32.20 | 0.948 | 0.116 | 29.84 | 0.889 | 0.203 | 24.74 | 0.892 | 0.116 | 31.62 | 0.942 | 0.080 |
| **Ours** | 30.59 | 0.862 | 0.139 | 32.70 | 0.947 | 0.144 | 27.33 | 0.870 | 0.269 | 24.11 | 0.884 | 0.168 | 31.55 | 0.938 | 0.096 |

map acquires a concrete physical interpretation as the accumulated transmittance of each ray through the medium up to the object surface. This allows the BCP-derived illumination map $T_{\mathcal{P}}$ to serve as a physically grounded transmittance prior $\tilde{T}$ in Equation (13) that constrains the transmittance at object surfaces. Leveraging this BCP-derived supervision, the media density converges toward a physically plausible distribution that maintains clear spatial separation from the object geometry, as shown in Figure 3. To incorporate this constraint during training, the model is optimized using Equation (14) with $\lambda$ hyperparameters set to 1, $10^{-2}$, $10^{-4}$, and $10^{-3}$, respectively.

## 5   Experiments

### 5.1   Implementation Details

Our model is built upon the ZipNeRF codebase [8], leveraging its hash-based encoding for efficient training. For NeRF sampling, we use 64 points for the object and 32 additional points for upsampling the media field. The global sunlight constant is initialized to CIE D65. Additional hyperparameter details and visualizations of experimental results are provided in Appendices G and I.

**Datasets**   For underwater environments, we use the SeaThru-NeRF dataset [45]. For low-light conditions, we conduct evaluations on the LOM dataset [16]. Both datasets follow the original train-test split. In addition, we captured two real-world underwater scenes in Okinawa, Pacific Ocean, using an OLYMPUS Tough TG-6 underwater camera and recorded the corresponding water depths.

**Metrics**   To assess reconstruction quality, we adhere to common metrics including PSNR, SSIM, and LPIPS. The best results are shaded as **first**, **second**, and **third** for each metric.

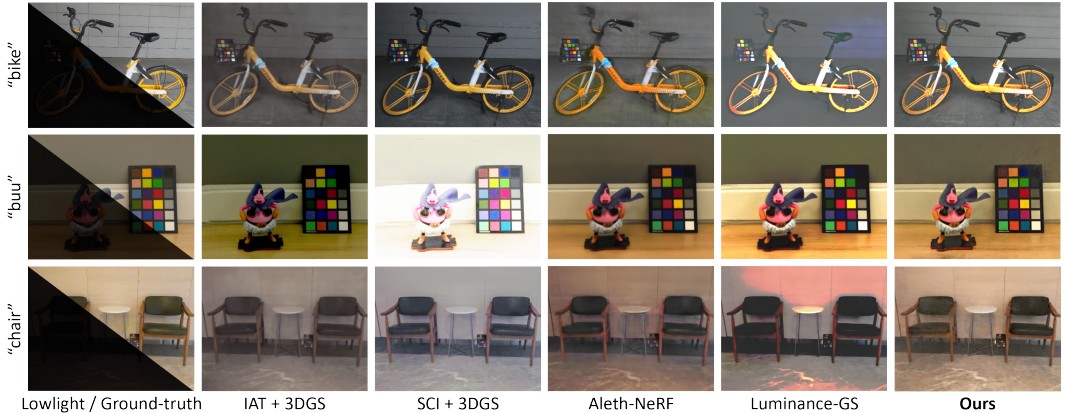

Figure 5: Qualitative comparison of NVS on the LOM dataset [16].

Table 2: Quantitative comparisons of low-light restoration performance on the LOM dataset [16].

| Model | "*bike*" | | | "*buu*" | | | "*chair*" | | | "*shrub*" | | | "*sofa*" | | |
|---|---|---|---|---|---|---|---|---|---|---|---|---|---|---|---|
| | PSNR | SSIM | LPIPS | PSNR | SSIM | LPIPS | PSNR | SSIM | LPIPS | PSNR | SSIM | LPIPS | PSNR | SSIM | LPIPS |
| ZipNeRF [8] | 6.64 | 0.083 | 0.611 | 7.72 | 0.291 | 0.445 | 6.19 | 0.151 | 0.590 | 8.48 | 0.037 | 0.658 | 6.30 | 0.212 | 0.553 |
| 3DGS [40] | 6.38 | 0.071 | 0.822 | 7.74 | 0.292 | 0.459 | 6.26 | 0.146 | 0.761 | 8.74 | 0.039 | 0.604 | 6.21 | 0.201 | 0.918 |
| SCI[54]+NeRF | 13.44 | 0.658 | 0.435 | 7.76 | 0.692 | 0.525 | 19.77 | 0.802 | 0.674 | 18.16 | 0.503 | 0.475 | 10.08 | 0.772 | 0.520 |
| IAT[15]+NeRF | 13.65 | 0.616 | 0.528 | 14.46 | 0.705 | 0.386 | 18.70 | 0.780 | 0.665 | 13.87 | 0.317 | 0.536 | 17.88 | 0.829 | 0.547 |
| SCI[54]+GS | 13.67 | 0.677 | **0.324** | 7.95 | 0.695 | 0.501 | **21.77** | **0.866** | 0.350 | **18.67** | 0.657 | **0.153** | 9.99 | 0.750 | 0.452 |
| IAT[15]+GS | 11.55 | 0.570 | 0.593 | 14.23 | 0.727 | 0.207 | 17.59 | **0.858** | **0.344** | 14.26 | 0.517 | 0.366 | 16.96 | 0.841 | 0.347 |
| Aleth-NeRF [16] | **20.46** | 0.727 | 0.499 | **20.22** | **0.859** | 0.315 | 20.93 | 0.818 | 0.468 | 18.24 | 0.511 | 0.448 | 19.52 | 0.857 | 0.354 |
| Lumin-GS [17] | 18.27 | **0.749** | 0.412 | 18.09 | **0.878** | **0.193** | 19.83 | 0.836 | 0.367 | 15.41 | **0.666** | **0.242** | 20.12 | **0.871** | **0.259** |
| LITA-GS [84] | 22.75 | **0.819** | **0.282** | 20.59 | 0.897 | 0.175 | 22.60 | 0.873 | 0.223 | 19.35 | 0.659 | 0.217 | 20.43 | 0.895 | 0.268 |
| **Ours** | **22.87** | 0.803 | **0.278** | **22.35** | 0.845 | **0.203** | **23.82** | 0.847 | **0.280** | **19.44** | **0.674** | 0.249 | **25.89** | **0.859** | **0.234** |

**Baselines** For underwater scenes, we compare against SeaThru-NeRF [45], Proposed-T [69], and Watersplatting [46]. For low-light scenarios, we compare our method with end-to-end reconstruction approaches, including Aleth-NeRF [16], Luminance-GS [17], and LITA-GS [84]. We also evaluate hybrid pipelines that incorporate 2D low-light enhancement models [15, 54].

## 5.2 Evaluation of Underwater Scenes

Table 1 reports quantitative results of NVS on the SeaThru-NeRF dataset [45], where our method achieves competitive performance. Figure 4 illustrates both NVS and scattering decomposition results. While Watersplatting [46] produces visually compelling scene reconstructions, it struggles to capture volumetric scattering effects and erroneously attributes object radiance and shadows to the scattered media component. Compared to SeaThru-NeRF [45], our method additionally estimates downwelling water depth. By incorporating downwelling attenuation modeling that accounts for water depth, our predicted medium color more accurately reproduces the spectral characteristics of natural oceanic water, where red light attenuates more rapidly than green and blue [4].

To evaluate the accuracy of the predicted downwelling depth, we applied our model to scenes captured in the Kerama Islands National Park, northwestern Pacific Ocean. As shown in Figure 1, the subsea anchor has an approximate scene footprint of six meters. Using this as a reference, we compute a scaling factor that maps the NeRF sampling space $[0, 1]$ to real-world metric units. Multiplying the predicted depth value in the region of the vanishing line by this factor yields an estimated downwelling distance of 7.2 meters. This result closely matches the recorded water depth of 7.5 meters as measured by the diving computer, demonstrating the accuracy and physical consistency of our model.

## 5.3 Evaluation of Low-light Scenes

For low-light restoration, we conduct experiments on the LOM dataset [16]. As shown in Table 2, our method achieves state-of-the-art PSNR performance across all scenes, outperforming both NeRF-based and GS-based approaches [16, 17, 84]. Figure 5 presents qualitative results on three representative scenes. Our physically grounded volumetric model enables consistent and balanced color restoration across views by explicitly modeling media degradation. In contrast, pixel-scaling-based baselines exhibit noticeable color inconsistencies. In Appendix G.4, we include a low-light underwater case study where our physically grounded model decomposes multiple media components and reconstructs clean radiance under hybrid conditions

## 5.4 Ablation Study

**Effect of RSU** We demonstrate the effectiveness of our upsampling strategy. Figure 6 shows two views from low-light and underwater scenes, displaying the restored radiance and corresponding density distributions in terms of sampling positions. Our strategy

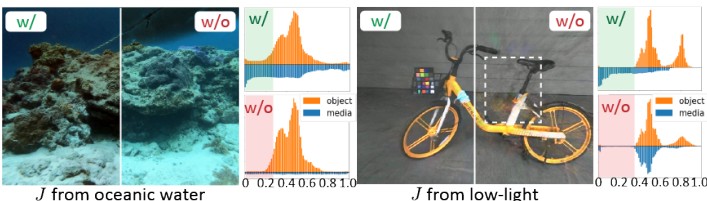

Figure 6: Ablation study of RSU upsampling strategy.

egy recovers the space-filling media in front of objects by explicitly allocating samples, in contrast to object-centric methods that neglect these regions and cause media field collapse to near-zero densities.

**Effect of Loss Terms** We present ablation analyses for each loss term under low-light or underwater scenarios in Table 3 and Table 4. Due to the inherent ill-posedness of reconstructing scenes through media, our proposed loss terms are specifically designed to enforce physical plausibility. While omitting $\mathcal{L}_{\mathrm{mutex}}$ or permitting a non-homogeneous scattering field may enhance the rendering quality of

Table 3: Ablation of absorption field on the "bike" [16].

| Method | PSNR | SSIM | LPIPS |
|---|---|---|---|
| w/o upsampling | 20.92 | 0.792 | 0.284 |
| w/o $\mathcal{L}_{\mathrm{geo}}$ | 20.66 | 0.776 | 0.297 |
| w/o $\mathcal{L}_{\mathrm{comp}}$ | 6.55 | 0.078 | 0.712 |
| w/o $\mathcal{L}_{\mathrm{mutex}}$ | 22.71 | 0.793 | 0.281 |
| w/o $\mathcal{L}_{\mathrm{trans}}$ | 21.87 | 0.790 | 0.285 |
| Full model | **22.87** | **0.803** | **0.278** |

Table 4: Ablation of underwater field on our captured scene in Okinawa.

| Method | PSNR | SSIM | LPIPS |
|---|---|---|---|
| inconstant media | **33.97** | **0.913** | **0.172** |
| w/o upsampling | 32.08 | 0.885 | 0.215 |
| w/o $\mathcal{L}_{\mathrm{geo}}$ | 29.95 | 0.852 | 0.319 |
| w/o $\mathcal{L}_{\mathrm{mutex}}$ | 33.11 | 0.902 | 0.178 |
| Full model | 31.25 | 0.887 | 0.201 |

the observation $I$, it hinders the accurate recovery of the clean radiance $J$ due to the lack of explicit density regulation, leading to media field collapse. More analysis and visualizations are provided in Appendix F.

## 6 Conclusion

We propose $I^2$-NeRF, a physically grounded volumetric model for radiative interactions in participating media. The model preserves isometry and isotropy through explicit sampling of dense objects and semi-transparent media, and with a unified attenuation model applied consistently across viewing and vertical axes. This design enables metric-preserving scene reconstruction that faithfully reflects real-world spatial structure. Experiments across diverse media conditions demonstrate the effectiveness of our approach in achieving high reconstruction quality and physical plausibility.

**Limitations** While $I^2$-NeRF demonstrates competitive performance in various degraded media conditions, it has several limitations. First, compared to rasterization-based methods [40, 17, 46], our NeRF model requires longer training time due to its neural implicit representation. Second, our model assumes static media with fixed optical properties, limiting its applicability to dynamic scenes such as moving water or drifting fog. Moreover, under scattering conditions, media properties such as downwelling depth have dependency on the encoded viewing directions, leading to variations that may not be strictly tied to spatial position. In low-light settings, varying light across views can also degrade performance, as our static virtual absorption medium cannot adapt to dynamic illumination.

## Acknowledgement

This work was partially supported by JST Moonshot R&D Grant Number JPMJPS2011, CREST Grant Number JPMJCR2015 and Basic Research Grant (Super AI) of Institute for AI and Beyond of the University of Tokyo. In addition, this work was also partially supported by JST SPRING, Grant Number JPMJSP2108.

The authors sincerely thank Mr. Zongqi Pan and the DeepSea Diving Center (Okinawa, Japan) for their valuable assistance in capturing the underwater scenes used in this study.

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

# — Appendix —

## Table of Contents

## A  Image Formation Under Media Degradation

In Section 3.2, we introduced a radiative model describing image formation under media degradation conditions. Here, we provide a detailed explanation of the typical image formation process in such environments. Specifically, we show that image formation under media degradation can be simplified and described clearly as a matting process, which combines direct radiance from objects and ambient illumination scattered by particles present in the medium.

### A.1  Degenerated Atmosphere Scattering Model (ASM)

In a hazy atmosphere, airborne particles such as water droplets, dust, and aerosols scatter and absorb incident light. These effects form a veil that reduces contrast and desaturates color [55]. Under the

single-scattering assumption of the Atmospheric Scattering Model (ASM), the intensity observed at each pixel of observation $I$ is composed of two components: direct radiance from the object attenuated exponentially with distance, and ambient illumination scattered toward the camera by airborne particles. This is formulated as:

$$\underbrace{I}_{observed\ radiance} = \overbrace{\underbrace{J}_{object\ radiance} \cdot \underbrace{(e^{-\sigma \cdot z})}_{attenuation}}^{direct} + \overbrace{\underbrace{B^{\infty}}_{airlight} \cdot \underbrace{(1 - e^{-\sigma \cdot z})}_{accumulation}}^{backscatter}, \tag{16}$$

where $z$ denotes the horizontal distance from the object surface to the camera. $J$ represents the radiance of a clean image without media degradation, and $B^{\infty}$ indicates the ambient airlight originating from distant illumination sources. In ASM, a single attenuation coefficient $\sigma$ characterizes both direct transmission and backscatter. This shared attenuation coefficient exhibits wavelength independence due to the dominance of Mie scattering. Mie scattering occurs when particle sizes in haze significantly exceed the wavelengths of visible light, resulting in nearly equal scattering across all wavelengths.

## A.2 Revised Underwater Image Formation (RUIF) Model

Unlike in air, image formation underwater is strongly dependent on wavelength $\lambda$. This occurs because water molecules and suspended particles selectively absorb and scatter different wavelengths of visible light. Define the beam absorption coefficient as $a(\lambda)$, the beam scattering coefficient as $b(\lambda)$, and the beam attenuation coefficient as $\beta(\lambda) = a(\lambda) + b(\lambda)$. Assuming a horizontal line-of-sight (LoS), the narrow-sense underwater image formation model can be expressed as:

$$\underbrace{I}_{observed\ radiance} = \overbrace{\underbrace{J}_{object\ radiance} \cdot \underbrace{(e^{-(a(\lambda)+b(\lambda))\cdot z})}_{attenuation}}^{direct} + \overbrace{\underbrace{E(z,\lambda)}_{irradiance} \cdot \underbrace{\frac{b(\lambda)}{a(\lambda)+b(\lambda)}}_{albedo} \cdot \underbrace{(1 - e^{-(a(\lambda)+b(\lambda))\cdot z})}_{accumulation}}^{backscatter}. \tag{17}$$

where $E(z, \lambda)$ denotes the ambient irradiance at distance $z$ along horizontal viewing directions. Moreover, due to differences in optical pathways and directional sensitivity of the imaging system, the camera response to direct and scattered radiance signals differs significantly. As a result, two distinct coefficients, $\sigma^{\mathrm{attn}}(\lambda)$ for direct signal attenuation and $\sigma^{\mathrm{scat}}(\lambda)$ for backscattered signal accumulation, are introduced. Incorporating these separate coefficients simplifies Equation (17)) into the Revised Underwater Image Formation (RUIF) model proposed by [2]:

$$\underbrace{I}_{observed\ radiance} = \overbrace{\underbrace{J}_{object\ radiance} \cdot \underbrace{(e^{-\sigma^{\mathrm{attn}}(\lambda)\cdot z})}_{attenuation}}^{direct} + \overbrace{\underbrace{B^{\infty}(\lambda)}_{ambient\ light} \cdot \underbrace{(1 - e^{-\sigma^{\mathrm{scat}}(\lambda)\cdot z})}_{accumulation}}^{backscatter}, \tag{18}$$

In this context, $B^{\infty}(\lambda)$ denotes the veiling light originating from ambient illumination at an infinite viewing distance, defined as:

$$B^{\infty}(\lambda) = E(z^{\Phi}, \lambda) \cdot \frac{b(\lambda)}{a(\lambda) + b(\lambda)}, \tag{19}$$

where $z^{\Phi}$ represents the vertical downwelling distance from the water surface. Although similar in form to the ASM, this underwater model differs in two critical ways. First, the direct attenuation coefficient $\sigma^{\mathrm{attn}}(\lambda)$ and the scattering accumulation coefficient $\sigma^{\mathrm{scat}}(\lambda)$ are distinct values. Second, both coefficients exhibit wavelength-dependent values, accurately reflecting the selective absorption and scattering characteristics of underwater media across different color channels.

## A.3 Simple Low-light Scaling Model

As shown in Section 3.1, under low-light conditions, the image formation can be effectively approximated using a simple scaling model. This simplification is valid because, in low-light conditions,

veiling light is negligible relative to the significantly reduced direct radiance. The simplified scaling model can thus be expressed as:

$$\underbrace{I}_{observed\ color} = \overbrace{\underbrace{J}_{object\ color} \cdot \underbrace{K}_{scaling\ factor}}^{direct} + \overbrace{0}^{backscatter} \tag{20}$$

where $K$ denotes a scaling factor and the backscatter component is set to zero ($B^{\infty} = 0$). In the general radiative formulation, this simplified scaling scenario can be physically interpreted by introducing an absorbing medium positioned between the camera and the object, attenuating the photon flux along the viewing ray. Given the attenuation coefficient $\sigma^{\mathrm{attn}}$ and the object depth $z$, the low-light image formation can then be represented explicitly as:

$$\underbrace{I}_{observed\ radiance} = \overbrace{\underbrace{J}_{object\ radiance} \cdot \underbrace{\left(e^{-\sigma^{\mathrm{attn}} \cdot z}\right)}_{attenuation}}^{direct} + \overbrace{0}^{backscatter} \tag{21}$$

## A.4 Low-light Bright Channel Prior (BCP) Model

Inspired by the Dark Channel Prior (DCP) model [33], which was originally developed for modeling hazy environments, the Bright Channel Prior (BCP) model [75] was introduced to characterize illumination degradation in low-light images and to support local exposure correction. BCP formulates low-light image formation using an image matting approach. Specifically, the observed low-light image $I$ is modeled as:

$$\underbrace{I}_{observed\ color} = \overbrace{\underbrace{J}_{object\ color} \cdot \underbrace{T_{\mathcal{P}}}_{illumination\ intensity}}^{direct} + \overbrace{\underbrace{B}_{ambient\ color} \cdot \underbrace{(1 - T_{\mathcal{P}})}_{illumination\ compensation}}^{compensation} \tag{22}$$

Here, $T_{\mathcal{P}}$ represents the local illumination intensity computed over a patch $\mathcal{P}$ centered at each pixel. Compared to the simple scaling model introduced in Equation (20), this formulation additionally includes a global ambient term $B$ that accounts for veiling light in underexposed regions.

BCP is based on the assumption that in well-lit images, the bright channel—defined as the maximum intensity value across color channels within a local patch—approaches one. Formally, for a clean image $J$, the bright channel is:

$$J_{BC} = \max_{c \in \{r,g,b\}} \left( \max_{q \in \mathcal{P}} J_q^c \right) \tag{23}$$

Empirically, $J_{BC} \approx 1$ for well-lit images. In contrast, for a low-light image $I$, the corresponding bright channel $I_{BC} \ll 1$ due to suppressed illumination.

To derive the illumination map $T_{\mathcal{P}}$ from low-light observation $I$, we first express Equation (22) in channel-wise as:

$$I^c = J^c \cdot T_{\mathcal{P}} + B^c \cdot (1 - T_{\mathcal{P}}), \quad c \in \{r,g,b\} \tag{24}$$

Solving for $T_{\mathcal{P}}$ yields:

$$T_{\mathcal{P}} = \frac{I^c - B^c}{J^c - B^c} \tag{25}$$

To estimate $T_{\mathcal{P}}$, a maximum operator is applied over the patch $\mathcal{P} \subseteq I$:

$$T_{\mathcal{P}} = \max_{c \in \{r,g,b\}} \left( \max_{q \in \mathcal{P}} \left( \frac{I_q^c - B^c}{J^c - B^c} \right) \right) \tag{26}$$

Assuming $J_{BC} = 1$ and that $T_{\mathcal{P}}$ is constant within the patch, this simplifies to:

$$T_{\mathcal{P}} = \max_{c \in \{r,g,b\}} \left( \max_{q \in \mathcal{P}} \left( \frac{I_q^c - B^c}{1 - B^c} \right) \right) \tag{27}$$

Rewriting this expression leads to the final BCP formulation:

$$T_{\mathcal{P}} = 1 - \max_{c \in \{r,g,b\}} \left( \max_{q \in \mathcal{P}} \left( \frac{1 - I_q^c}{1 - B^c} \right) \right) \tag{28}$$

The ambient constant $B$ can be estimated empirically as the average of the darkest 0.1% pixels in the bright channel $I_{BC}$ [75]. Since $B$ is typically small, we can approximate $B \approx 0$, which simplifies Equation (22) to:

$$I = J \cdot T_{\mathcal{P}} \tag{29}$$

This simplified form is equivalent to our low-light radiative formulation in Equation (21) with veiling light $B = 0$. Consequently, the BCP-derived local illumination map $T_{\mathcal{P}}$ acquires a physical interpretation as the transmittance through a virtual absorption medium. We leverage this transmittance map to supervise the estimation of the medium density in our training process, as described in Section 4.

## B  Relation of Radiative Formulation and Particle Model

In Sections 3.2 and 3.3, we introduced a general radiative formulation for modeling image degradation in participating media, along with a volumetric particle model that incorporates emission, absorption, and scattering processes. In this section, we establish the equivalence between the two by demonstrating how our particle-based volumetric rendering aligns with the radiative formulation.

Using isometric sampling for the participating medium, we discretize the volume along each camera ray into intervals $[s_i, s_{i+1}]$, and assume uniform spacing where $s_i = i \cdot \delta$ and $\delta$ denotes the constant sampling step. Suppose a single opaque object lies along the ray at the end of the interval $[s_k, s_{k+1}]$, the object depth is then given by $z = k \cdot \delta$. The ray terminates at index $i = k$ when it intersects the object surface, and the transmittance drops to zero beyond this point.

Under this setting, we define the object density $\sigma^{\mathrm{obj}}$ to be nonzero only at index $i = k$, and zero for all $i < k$. Conversely, the medium densities, such as the absorption coefficient $\sigma^{\mathrm{attn}}$ or the scattering coefficient $\sigma^{\mathrm{scat}}$, are defined only for intervals prior to the object surface, i.e., $\sigma^{\mathrm{attn}}, \sigma^{\mathrm{scat}} = 0$ for $i \geq k$. This setup reflects a spatial separation between the opaque object and the participating medium, capturing the physical intuition that dense objects and sparse media do not occupy the same space along the ray.

### B.1  Low-light Conditions

Under low-light conditions, we simulate reduced visibility by introducing a virtual absorption medium. In this case, Equation (3) simplifies to:

$$\hat{C} = \sum_{i=1}^{N} \hat{C}_i^{\mathrm{obj}} = \hat{C}_k^{\mathrm{obj}} = \underbrace{T_k^D \cdot \left( 1 - \exp(-\sigma_k^{\mathrm{obj}} \delta_k) \right) c_k^{\mathrm{obj}}}_{\textit{direct radiance}} \tag{30}$$

$$= \exp\left( -\sum_{j=1}^{k} (\sigma_j^{\mathrm{obj}} + \sigma_j^{\mathrm{attn}}) \delta_j \right) \cdot \left( 1 - \exp(-\sigma_k^{\mathrm{obj}} \delta_k) \right) c_k^{\mathrm{obj}} \tag{31}$$

Assuming $\sigma_j^{\mathrm{obj}} = 0$ for all $j < k$, and taking both the interval length $\delta_j = \delta$ and the attenuation coefficient $\sigma_j^{\mathrm{attn}} \approx \sigma^{\mathrm{attn}}$ as roughly constants, the expression simplifies to:

$$\hat{C} = \exp(-\sigma^{\mathrm{attn}} \cdot k\delta) \cdot \left( 1 - \exp(-\sigma_k^{\mathrm{obj}} \delta) \right) c_k^{\mathrm{obj}} \tag{32}$$

In typical low-light scenarios, the object density $\sigma_k^{\mathrm{obj}}$ is sufficiently large that the ray terminates at the object surface, resulting in $\exp(-\sigma_k^{\mathrm{obj}} \delta) \approx 0$. Therefore, we further simplify the expression to:

$$\hat{C} = c_k^{\mathrm{obj}} \cdot \exp(-\sigma^{\mathrm{attn}} \cdot k\delta) \tag{33}$$

$$= c_k^{\mathrm{obj}} \cdot \exp(-\sigma^{\mathrm{attn}} \cdot z) \tag{34}$$

This result corresponds to a special case of our general radiative formulation under low-light conditions, as presented in Section 3.2, where the observed color is modeled as the attenuated object radiance with $J = c_k^{\text{obj}}$ and no ambient component ($B = 0$).

## B.2  Haze and Underwater Conditions

In scattering-dominant environments, where backscattered radiance contributes significantly to the observed signal, we follow the derivation approach of [45] to formulate Equation (6) as:

$$\hat{C} = \sum_{i=1}^{N} \hat{C}_i^{\text{obj}} + \sum_{i=1}^{N} \hat{C}_i^{\text{med}} = \hat{C}_k^{\text{obj}} + \sum_{i=1}^{N} \hat{C}_i^{\text{med}} \tag{35}$$

$$= \underbrace{T_k^D \left(1 - \exp(-\sigma_k^{\text{obj}}\delta_k)\right) c_k^{\text{obj}}}_{\text{direct radiance}} + \underbrace{\sum_{i=1}^{k} T_i^B \left(1 - \exp(-\sigma_i^{\text{scat}}\delta_i)\right) c_i^{\text{med}}}_{\text{backscatter radiance}} \tag{36}$$

The direct radiance term in Equation (36) is identical to the expression derived in (30), and can be simplified using the result in (34). We now focus on the backscatter term:

$$\sum_{i=1}^{k} \hat{C}_i^{\text{med}} = \sum_{i=1}^{k} \exp\left(-\sum_{j=1}^{i}(\sigma_j^{\text{obj}} + \sigma_j^{\text{scat}})\delta_j\right) \left(1 - \exp(-\sigma_i^{\text{scat}}\delta_i)\right) c_i^{\text{med}} \tag{37}$$

$$= \sum_{i=1}^{k} \exp(-\sigma^{\text{scat}} \cdot i\delta) \left(1 - \exp(-\sigma^{\text{scat}}\delta)\right) c_i^{\text{med}} \tag{38}$$

Assuming the single scattering approximation holds and that the medium properties are approximately uniform along the ray, we treat $c_i^{\text{med}}$ as a constant value $c^{\text{med}}$, and obtain:

$$\sum_{i=1}^{k} \hat{C}_i^{\text{med}} = \left(1 - \exp(-\sigma^{\text{scat}}\delta)\right) c^{\text{med}} \sum_{i=1}^{k} \exp(-\sigma^{\text{scat}} \cdot i\delta) \tag{39}$$

$$= \left(1 - \exp(-\sigma^{\text{scat}}\delta)\right) c^{\text{med}} \cdot \left(\frac{1 - \exp(-\sigma^{\text{scat}} \cdot k\delta)}{1 - \exp(-\sigma^{\text{scat}}\delta)}\right) \tag{40}$$

$$= 1 - \exp(-\sigma^{\text{scat}} \cdot k\delta)c^{\text{med}} \tag{41}$$

Substituting (41) back into (36) gives:

$$\hat{C} = \underbrace{c_k^{\text{obj}} \cdot \exp(-\sigma^{\text{attn}} \cdot k\delta)}_{\text{direct}} + \underbrace{c^{\text{med}}(1 - \exp(-\sigma^{\text{scat}} \cdot k\delta))}_{\text{backscatter}} \tag{42}$$

$$= c_k^{\text{obj}} \cdot \exp(-\sigma^{\text{attn}} \cdot z) + c^{\text{med}}(1 - \exp(-\sigma^{\text{scat}} \cdot z)) \tag{43}$$

This expression is consistent with our general radiative formulation, where $J = c_k^{\text{obj}}$ and $B = c^{\text{med}}$. In atmospheric environments, the absorption and scattering components are often combined into a single extinction coefficient, such that $\sigma^{\text{attn}} = \sigma^{\text{scat}}$, due to the relatively uniform interaction between light and small airborne particles. In contrast, underwater environments exhibit distinct optical behaviors, where $\sigma^{\text{attn}} \neq \sigma^{\text{scat}}$.

## C  Downwelling Attenuation Model

In Equation (19), the ambient light at infinity $B^{\infty}(\lambda)$ is proportional to the irradiance $E(z^{\Phi}, \lambda)$, which models the downwelling light reaching the medium along vertical paths. This irradiance accounts for the cumulative attenuation of sunlight as it travels downward through the medium. According to [66], the downwelling irradiance at a vertical distance $z^{\Phi}$ can be modeled as:

$$E(z^{\Phi}, \lambda) = E(0, \lambda) \cdot \exp(-K^{\Phi}(\lambda) \cdot z^{\Phi}) \tag{44}$$

Here, $E(0, \lambda)$ denotes the ambient light at the surface of the medium, and $K^{\Phi}(\lambda)$ is the diffuse downwelling attenuation coefficient. As derived in Equation (43), we can establish a connection between the medium radiance $c^{\text{med}}(\lambda)$ in our volume rendering framework and the downwelling irradiance at depth via:

$$c^{\text{med}}(\lambda) = B^{\infty}(\lambda) = \frac{b(\lambda)E(z^{\Phi})}{a(\lambda) + b(\lambda)} = \frac{b(\lambda)E(0)}{a(\lambda) + b(\lambda)} \cdot \exp(-K^{\Phi}(\lambda) \cdot z^{\Phi}). \quad (45)$$

According to [25], the diffuse downwelling attenuation coefficient $K^{\Phi}(\lambda)$ can be approximated as a function of the beam attenuation coefficients and the solar incident angle $\theta^{\Phi}$:

$$K^{\Phi}(\lambda) \approx (a(\lambda) + b(\lambda)) \cdot \cos \theta^{\Phi} \quad (46)$$

In practical scenarios, the beam absorption coefficient $a(\lambda)$ and scattering coefficient $b(\lambda)$ are difficult to estimate directly due to their dependence on local environmental and optical conditions. As an alternative, we approximate them using the learned attenuation coefficients $\sigma^{\text{attn}}(\lambda)$ and $\sigma^{\text{scat}}(\lambda)$ obtained from our neural radiance field. Assuming direct overhead sunlight with $\cos \theta^{\Phi} = 1$, substituting Equation (46) into Equation (45) yields:

$$c^{\text{med}}(\lambda) \approx \frac{b(\lambda)E(0, \lambda)}{a(\lambda) + b(\lambda)} \cdot \exp(-(\sigma^{\text{attn}}(\lambda) + \sigma^{\text{scat}}(\lambda)) \cdot z^{\Phi}) \quad (47)$$

$$= \Phi \cdot \exp(-(\sigma^{\text{attn}}(\lambda) + \sigma^{\text{scat}}(\lambda)) \cdot z^{\Phi}) \quad (48)$$

Here, $\Phi$ denotes an ambient light coefficient at the medium surface. Following the setup in [2], where $E(0, \lambda)$ is empirically initialized to solar illumination, we similarly initialize $\Phi$ to the CIE D65 daylight spectrum. This per-scene constant is then optimized jointly with the parameters of our NeRF model during training.

## D   Reverse-Stratified Upsampling

In Section 3.4, we introduce a media upsampling strategy designed to explicitly model the media field as spatially separate from the object geometry. In this section, we provide a detailed explanation of the subsequent stratified sampling and merging procedure. Specifically, given the reverse weights $w_i^{\text{med}}$ over the original $N$ intervals derived from Equation (7), we compute their cumulative distribution function (CDF) as:

$$\Gamma_i = \sum_{k=1}^{i} w_k^{\text{med}}, \quad F(i) = \frac{\Gamma_i}{\Gamma_N}, \quad i = 1, \dots, N. \quad (49)$$

We then draw $N_{\text{add}}$ stratified uniform samples:

$$u_j \sim \mathcal{U}\left(\tfrac{j-1}{N_{\text{add}}}, \tfrac{j}{N_{\text{add}}}\right), \quad j = 1, \dots, N_{\text{add}}, \quad (50)$$

Invert the CDF by identifying the unique index $i$ such that $F(i-1) < u_j \leq F(i)$, and sample within that interval as:

$$t_j^{\text{med}} \sim \mathcal{U}\left(t_i^{\text{obj}}, t_{i+1}^{\text{obj}}\right). \quad (51)$$

The complete set of sampled positions is formed by merging and sorting the object-centric and media-centric samples:

$$\{t_k\} = \text{sort}\left(\{t_i^{\text{obj}}\}_{i=1}^{N} \cup \{t_j^{\text{med}}\}_{j=1}^{N_{\text{add}}}\right). \quad (52)$$

The resulting sorted set $\{t_k\}$ is then forwarded to our neural radiance model.

## E   Objective Functions

In this section, we provide detailed formulations of the reconstruction loss $\mathcal{L}_{\text{recon}}$, the geometry loss $\mathcal{L}_{\text{geo}}$, and the revisited structural similarity loss $\mathcal{L}_{\text{struct}}$, which were introduced in Section 3.4.

## E.1 Reconstruction Loss $\mathcal{L}_{\text{recon}}$

In degraded images exhibiting low contrast, the standard L2 reconstruction loss employed in vanilla NeRF [56] often fails to emphasize differences in low-intensity regions. To address this limitation, we follow [57] to incorporate a tone curve linearization that places greater emphasis on reconstruction fidelity in low-intensity pixels. Specifically, we use the loss:

$$\mathcal{L}_{\text{recon}} = \frac{1}{M} \sum_{k=1}^{M} \left( \frac{\hat{I}_k - I_k}{\text{sg}(\hat{C}_k) + \epsilon} \right)^2, \tag{53}$$

where $\text{sg}(\cdot)$ denotes the stop-gradient operator, $M$ is the number of pixels in the batch, and $\epsilon = 10^{-3}$ to avoid division by zero. This loss implicitly applies a gradient supervision of the tone-mapping curve $\log(x + \epsilon)$ and produces high reconstruction quality in degraded conditions.

## E.2 Geometry Loss $\mathcal{L}_{\text{geo}}$

To regularize scene geometry under visually degraded conditions, we utilize the pretrained DeepAnything-V2 model [82] to generate pseudo depth maps as a geometric prior. Since the predicted depth is the relative depth map, we normalize the rendered depth to the range $[0, 1]$. The geometry loss is then defined as

$$\mathcal{L}_{\text{geo}} = \frac{1}{M} \sum_{k=1}^{M} \left\| \hat{D}_k - \tilde{D}_k \right\|^2, \tag{54}$$

where $\hat{D}_k$ denotes the normalized depth rendered from our NeRF model, $\tilde{D}_k$ is the corresponding depth prior from the pretrained model, and $M$ is the number of pixels in the batch.

## E.3 Compensated Structural Similarity Loss $\mathcal{L}_{\text{comp}}$

The Structural Similarity Index (SSIM) is a perceptual metric that evaluates image quality by comparing local patterns of pixel intensities. It decomposes similarity into three components: luminance, contrast, and structure. Formally, SSIM measures local means to assess luminance consistency, variances for contrast similarity, and normalized covariance to evaluate structural alignment.

However, in severely degraded conditions, especially under low-light environments where the observed radiance is strongly attenuated by the absorption medium, the observed image $I$ often deviates significantly in luminance and contrast from the rendered object radiance $\hat{J}$. To mitigate this discrepancy, we propose a revised structural similarity loss that compensates SSIM using the ideal luminance $\tilde{\nu}_J$ and contrast scaling factor $\tilde{\kappa}_J$. Such prior information can be derived from common well-lit images or tuned as hyperparameters during experiments. Specifically, let $G$ be a Gaussian kernel. We first compute local means of the rendered object radiance $\hat{J}$ and the observation $I$:

$$\hat{\nu}_J = G * \hat{J}, \qquad \nu_I = G * I, \tag{55}$$

along with the global mean of the observation $I$ as:

$$m_I = \frac{1}{M} \sum_{p_i}^{M} I_{p_i}. \tag{56}$$

Here, $M$ denotes the number of pixels within the batch. We then compensate the observation's luminance and contrast using the prior $\tilde{\nu}_J$ and $\tilde{\kappa}_J$ as follows:

$$\nu_I^{\text{comp}} = (\nu_I - m_I)\,\tilde{\kappa}_J + \tilde{\nu}_J, \qquad (\sigma_I^{\text{comp}})^2 = \tilde{\kappa}_J^2 \left( G * I^2 - \nu_I^2 \right). \tag{57}$$

For the rendered radiance $\hat{J}$, we compute local contrast and cross-correlation as:

$$\sigma_J^2 = G * \hat{J}^2 - \hat{\nu}_J^2, \qquad \sigma_{JI} = G * (\hat{J} \cdot I) - \hat{\nu}_J \cdot \nu_I, \qquad \sigma_{JI}^{\text{comp}} = \tilde{\kappa}_J \cdot \sigma_{JI}. \tag{58}$$

With constants $C_1 = (0.01)^2$ and $C_2 = (0.03)^2$, the compensated SSIM index is then derived as:

$$\text{SSIM}^{\text{comp}} = \underbrace{\frac{2\,\hat{\nu}_J\,\nu_I^{\text{comp}} + C_1}{\hat{\nu}_J^2 + (\nu_I^{\text{comp}})^2 + C_1}}_{\text{luminance}} \times \underbrace{\frac{2\,\sigma_{JI}^{\text{comp}} + C_2}{\sigma_J^2 + (\sigma_I^{\text{comp}})^2 + C_2}}_{\text{contrast–structure}}. \tag{59}$$

And the SSIM loss is given by:

$$\mathcal{L}_{\mathrm{SSIM}} = 1 - \mathrm{SSIM}^{\mathrm{comp}}(\hat{J}, I, \tilde{\nu}_J, \tilde{\kappa}_J). \tag{60}$$

Since NeRF training randomizes ray sampling and disrupts spatial correlations, we adopt the strategy from [80] and apply our revised SSIM loss over stochastic image patches. Specifically, we define the final structural similarity loss as

$$\mathcal{L}_{\mathrm{comp}} = \frac{1}{H} \sum_{h=1}^{H} \mathcal{L}_{\mathrm{SSIM}}(\mathcal{P}_h(\hat{J}), \mathcal{P}_h(I); \tilde{\nu}_J, \tilde{\kappa}_J), \tag{61}$$

where $\mathcal{P}$ extracts the $h$-th randomly partitioned ray patch, and repeat $H$ times to obtain averaged loss.

## F  Additional Ablation Study

### F.1  Underwater Scene

In Section 5.4, we present a quantitative ablation study on each component of our NeRF model. Specifically, for the LOM dataset [16], the evaluation was performed by comparing the restored object radiance against the ground-truth well-illuminated images. In contrast, for the SeaThru-NeRF dataset [45], the evaluation was based on the rendering quality of degraded novel-view images, since obtaining clean object radiance free from scattering is infeasible in real-world underwater conditions. Due to this unavoidable difference in evaluation protocols between low-light and underwater scattering settings, the quantitative results reported in Table 4 do not fully capture the model's ability to disentangle and reconstruct scene components.

In this section, we elaborate on the ablation study for underwater scattering scenes and provide more intuitive qualitative and quantitative evaluations. As shown in Figure 7, we visualize novel-view synthesis results on an underwater scene captured by ourselves in Okinawa to highlight performance differences under well-illuminated yet challenging conditions, where backscattering is intensive. In such environments, backscattered radiance tends to collapse into object radiance, making the separation between the two more difficult. This radiance blending may superficially improve novel-view synthesis quality from the observed image perspective, as it effectively ignores the backscatter component and simplifies the task to resemble a clear-air scenario. However, doing so leads to inaccurate estimation of the media field, undermining the physical plausibility of the model.

We observe that omitting reverse-stratified upsampling (RSU), the geometry loss $\mathcal{L}_{\mathrm{geo}}$, or the mutual exclusivity loss $\mathcal{L}_{\mathrm{mutex}}$ results in the collapse of scattered radiance into the object radiance, as evident in the rendered object and backscatter components. Furthermore, this collapse leads to erroneous depth estimations in both horizontal and vertical directions, making the recovered scene geometry physically unreliable.

We also examine the case where spatially varying scattering coefficients $\sigma^{\mathrm{attn}}$ and $\sigma^{\mathrm{scat}}$ are used along each ray. In Section 4, we discuss that, to accommodate the near-homogeneous nature of the scattering medium and simplify the problem, we follow [45, 63] to assume constant media densities along the ray. While allowing inconstant scattering coefficients offers greater flexibility, it introduces extreme ill-posedness in the absence of prior knowledge about the water medium. In practice, real-world underwater conditions are highly complex, and acquiring reliable priors for water properties is challenging. Without such constraints, the decomposition of scene radiance becomes physically inconsistent and results in severely degraded reconstruction quality.

Table 5: Mean Chromatic Dissimilarity Metric (MCDM) between the restored object radiance $J$ and the original underwater observation $I$ for each ablation component of our model. Higher MCDM values indicate a greater chromatic separation from the original scatter-degraded image.

| Ablation | w/o RSU | w/o $\mathcal{L}_{\mathrm{mutex}}$ | w/o $\mathcal{L}_{\mathrm{geo}}$ | **full model** | inconstant $\sigma$ |
|---|---|---|---|---|---|
| Object | 6.06 | 11.64 | 13.94 | 15.88 | 24.22 |

To quantitatively assess the discrepancy between the recovered object radiance $\hat{J}$ and the observed image $I$, we employ the Mean Chromatic Dissimilarity Metric (MCDM). MCDM measures the average perceptual difference in chromaticity between two images, where higher values indicate greater deviation in color. Importantly, MCDM is not designed to reflect reconstruction fidelity;

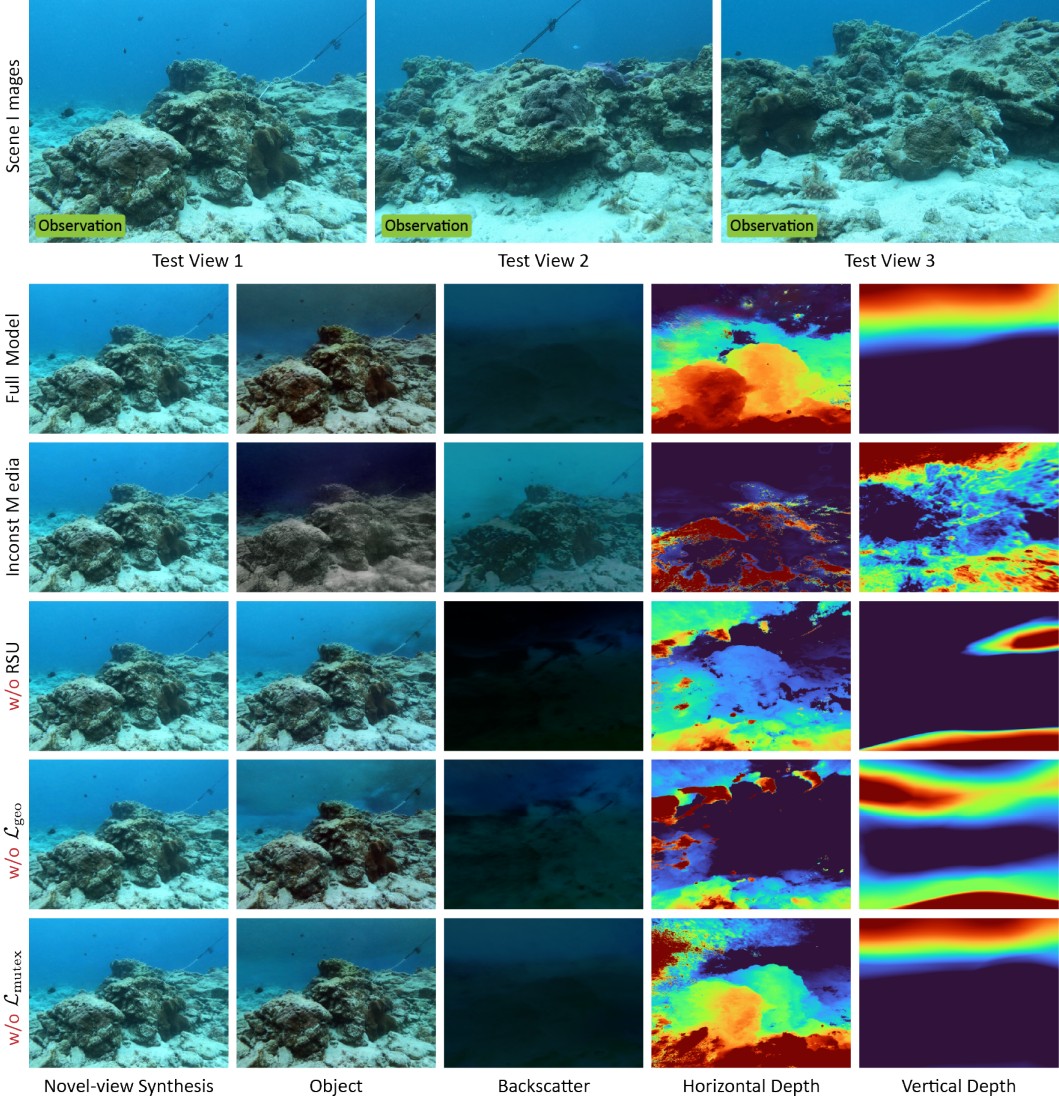

Figure 7: Visualization of the ablation study on an underwater scene captured in Okinawa. In this well-illuminated environment, stronger backscattering occurs due to the increased ambient light being scattered by suspended particles. As a result, distinguishing between the object radiance and the backscattered radiance becomes more challenging, especially when foreground objects exhibit colors similar to the background water body.

rather, it provides an objective measure of chromatic separation. As shown in Table 5, in the ablated components, lower MCDM values indicate that the recovered object radiance closely resembles the observed image, suggesting a collapse of the media field in which backscattered radiance is incorrectly absorbed into the object signal. On the other hand, excessively high MCDM values, such as using inconstant scattering coefficients, reflect a failure case in scene decomposition.

## F.2 Low-light Scene

In Table 3, we present an ablation study on the application of our model in low-light environments. In this section, we further investigate the effect of two critical hyperparameters: the ideal illumination $\tilde{\nu}_J$ and the contrast factor $\tilde{\kappa}_J$, both of which are used in the compensated structural similarity loss $\mathcal{L}_{\mathrm{comp}}$ as defined in Equation (61).

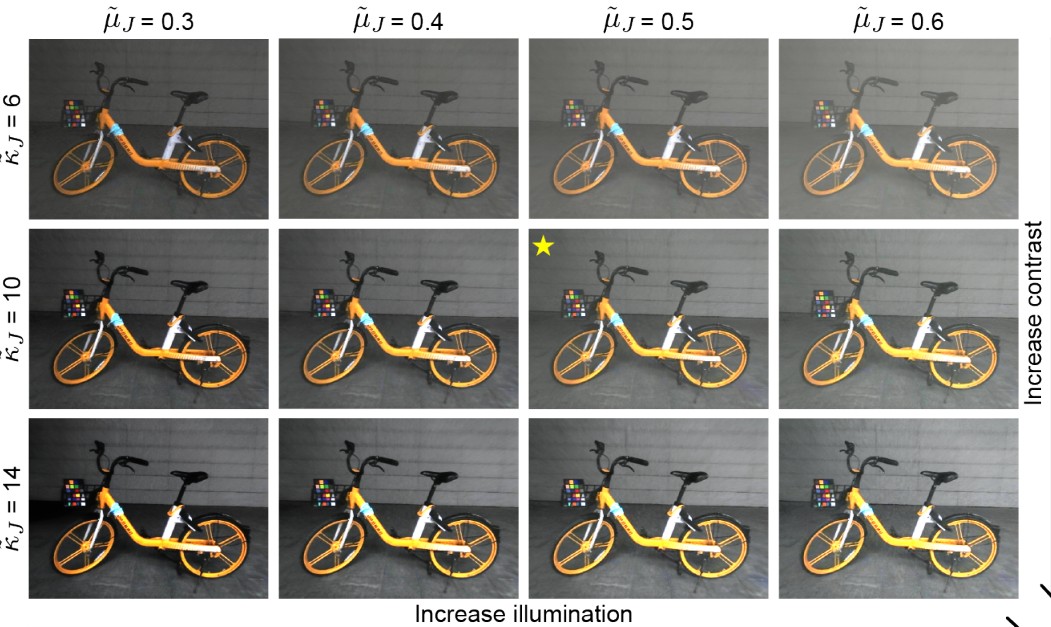

Figure 8: Ablation study on the hyperparameters for ideal illumination $\tilde{\nu}_J$ and contrast factor $\tilde{\kappa}_J$ in our model, conducted on the "bike" scene from the LOM dataset [16].

As shown in Figure 8, $\tilde{\nu}_J$ primarily controls the brightness of the restored clean object radiance $\hat{J}$, with higher values leading to increased pixel intensities. $\tilde{\kappa}_J$ governs the contrast of $\hat{J}$, with higher values enhancing edge sharpness. The yellow pentagram indicates the selected hyperparameter values used in our final model, which are close to the appearance of naturally well-lit images.

## G  More Experimental Results

### G.1  Underwater Scene

As shown in Figure 9, we compare our method against SeaThru-NeRF [45] and Watersplatting [46] on the SeaThru-NeRF dataset [45]. In the case of SeaThru-NeRF, the horizontal depth maps are truncated at regions with low accumulated radiance weights, and the oceanic background is removed during depth map rendering. Compared to the baseline methods, our model produces more accurate depth estimation along LoS and enables spatial perception that extends beyond the typical one-dimensional ray to incorporate vertical distance.

While Watersplatting [46] achieves state-of-the-art performance in full-image rendering, it struggles to accurately decompose scene radiance. Consequently, its backscatter component often contains erroneous elements from the object signal, such as shadows or textures. In contrast, our approach yields more physically faithful geometry and radiance decomposition, leading to superior scene reconstruction under media degradation.

### G.2  Hazy Scene

For the hazy environment, we conduct experiments on the synthetic hazy Fern scene provided in the SeaThru-NeRF dataset [45], as capturing real-world multi-view hazy scenes is challenging. The synthetic scene is generated by first estimating the depth along the LoS and then applying the ASM to synthesize haze-induced degradation. Such hazy conditions present notable challenges for degradation removal, as the additional backscattered radiance significantly increases pixel intensity. As a result, the strong backscatter signal is often misinterpreted by models as part of the scene geometry, leading to inaccurate estimation of the medium properties.

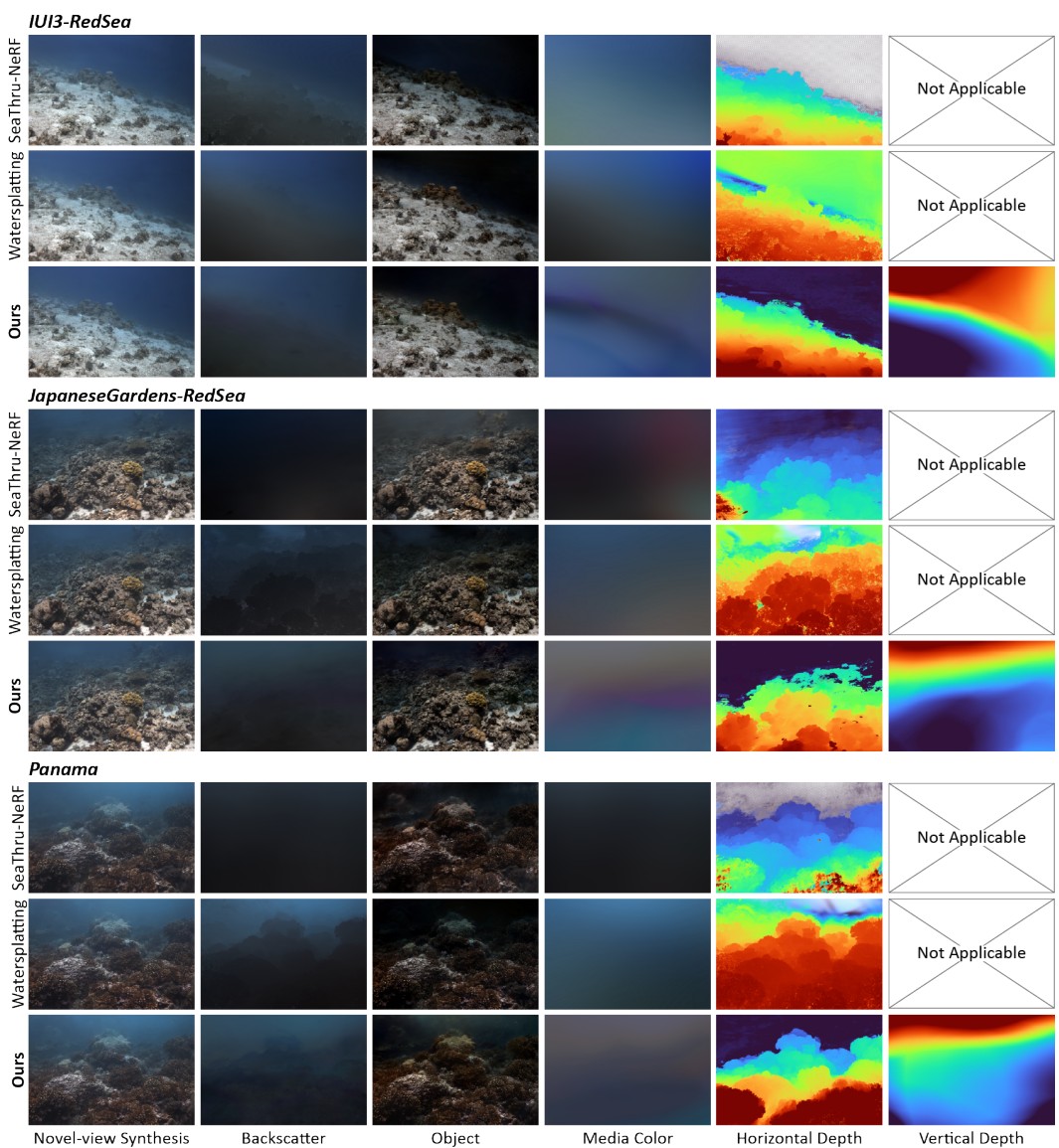

Figure 9: Visualization of our method and baseline approaches on the SeaThru-NeRF dataset [45].

As shown in Figure 10, SeaThru-NeRF [45] exhibits media field collapse, where the estimated backscattered radiance is nearly zero and the restored image fails to remove haze-induced degradation. In contrast, Watersplatting [46] suffers from an opposite failure case, which overestimates backscatter and erroneously attributes object radiance to the medium component. This results in underexposed restorations with blurred boundaries and degraded structural fidelity, as shadows and fine edges are inaccurately absorbed into the backscattered signal.

In contrast, our model achieves a well-balanced decomposition by incorporating physically grounded constraints and a metric-preserving perception strategy, producing sharper restorations and more accurate scattering estimation. Furthermore, our isotropic attenuation model with downwelling distance enables more realistic prediction of medium color, better capturing the color pattern of haze compared to SeaThru-NeRF [45].

It is worth noting that the synthetic scene is constructed using a globally constant scattering coefficient $\sigma$, which differs from real-world scattering environments where the scattering varies spatially with medium depth and thickness. Consequently, our model predicts a uniform vertical depth map, which

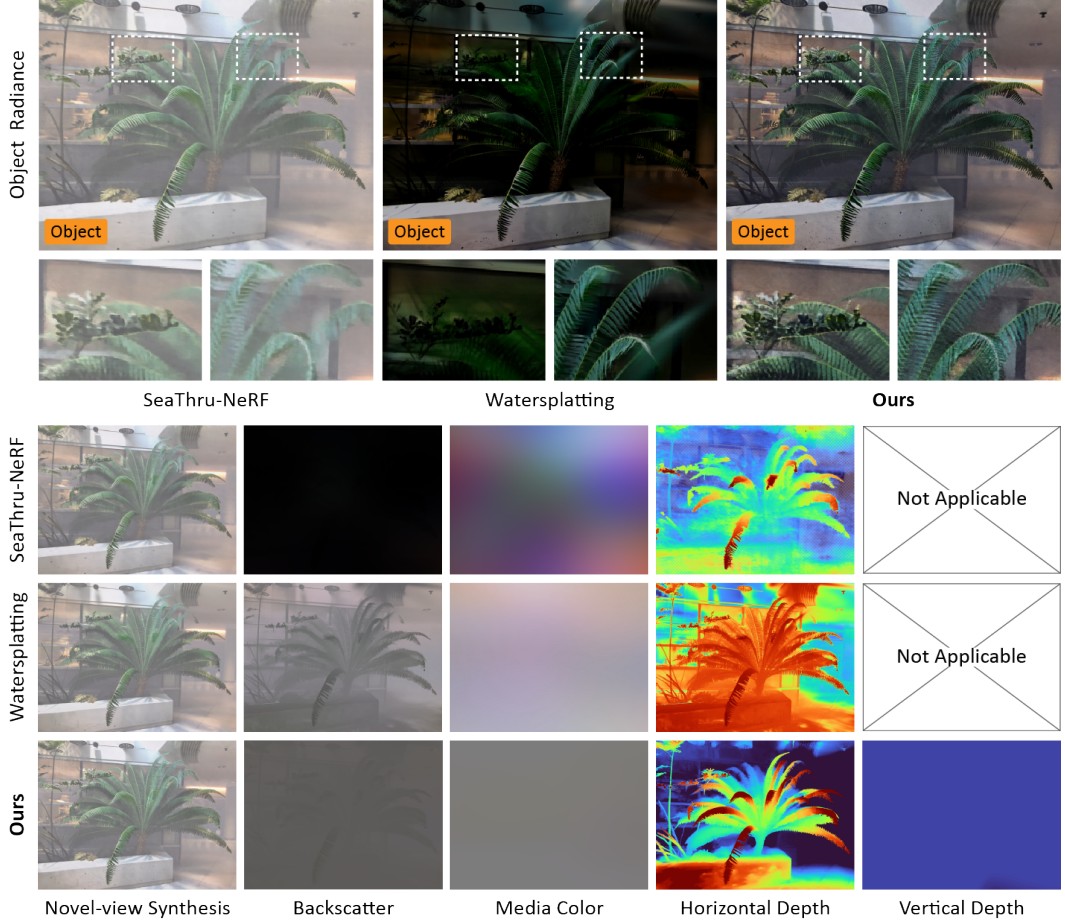

Figure 10: Visualization of our method and baseline approaches on a synthetic hazy scene from the SeaThru-NeRF dataset [45]. The top row shows the restored clean object radiance, while the bottom row compares full image rendering, backscattered radiance, estimated medium color $c^{med}$, and depth maps.

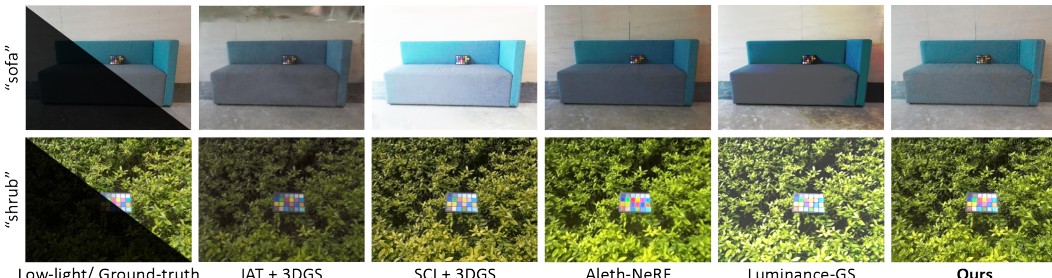

Figure 11: Low-light restoration results of our method compared to baseline approaches on the LOM dataset [16].

actually reflects the characteristics of this synthetic setup. This outcome also serves as a sanity check, reinforcing that our model provides a physically plausible interpretation of the scene.

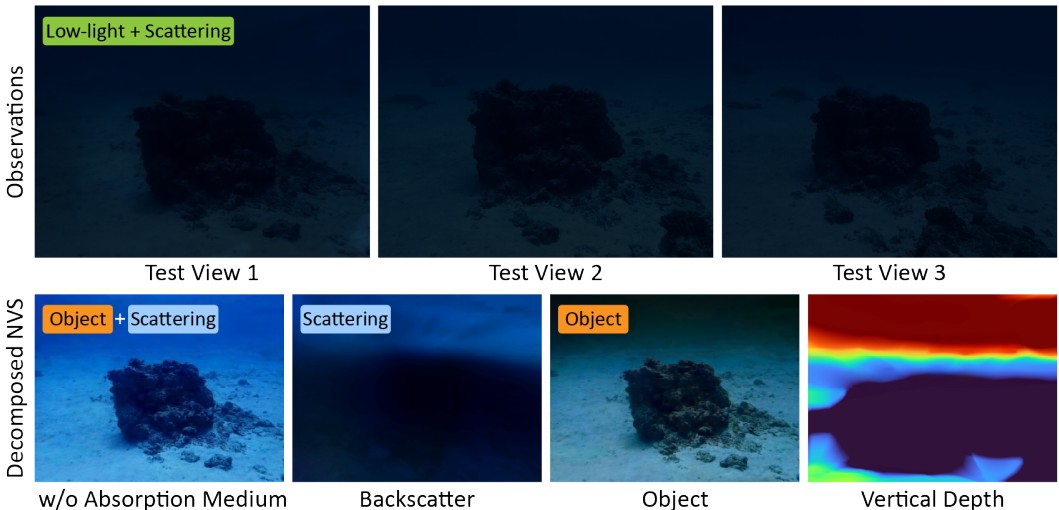

Figure 12: Visualization of a real-world low-light underwater scene captured in Okinawa. We also present the decomposed radiance components, such as removing the virtual absorption medium used to model low-light conditions, the direct object radiance, and backscattered ambient light.

### G.3 Low-light Scene

As shown in Figure 11, we present visual results on the remaining two scenes from the LOM dataset [16]. Our method achieves more color-consistent restoration compared to both NeRF-based and Gaussian Splatting-based approaches.

### G.4 Low-light and Underwater Hybrid Scene

Benefiting from our general radiative formulation, the proposed model supports the simultaneous incorporation of multiple types of media, making it adaptable to complex real-world environments. As a case study, we captured low-light underwater scenes using an OLYMPUS Tough TG-6 camera under short-exposure settings. Within our particle-based neural radiance framework, we jointly model both low-light and scattering effects. As shown in Figure 12, our method effectively decomposes the scene and reconstructs clean object radiance under hybrid low-light and scattering conditions, demonstrating its robustness in real-world physical settings. Although the vertical depth estimation becomes less precise due to interference from multiple interacting media, it still faithfully reflects the depth property of the scene.

## H Downstream Applications

[45] demonstrates that modeling both the object and the participating medium in 3D space enables a range of downstream applications, such as scene geometry reconstruction, clear radiance restoration, and medium property estimation. Building on this foundation, we explore several physically grounded applications enabled by our $I^2$-NeRF framework, particularly in underwater environments.

### H.1 Single-Image Volume Estimation

Benefiting from our metric-preserving framework, our model enables the estimation of real-world physical quantities. In this section, we demonstrate how the volume of subsurface water can be estimated from a single underwater image, based on certain valid approximations and simplified geometric assumptions.

We begin by assuming an orthographic camera model, where all rays emitted from image pixels are parallel to the horizontal axis. For each pixel $p_{i,j}$ in an image of size $H_I \times W_I$, the emitted ray travels horizontally until it intersects with an object surface. The horizontal distance from the camera

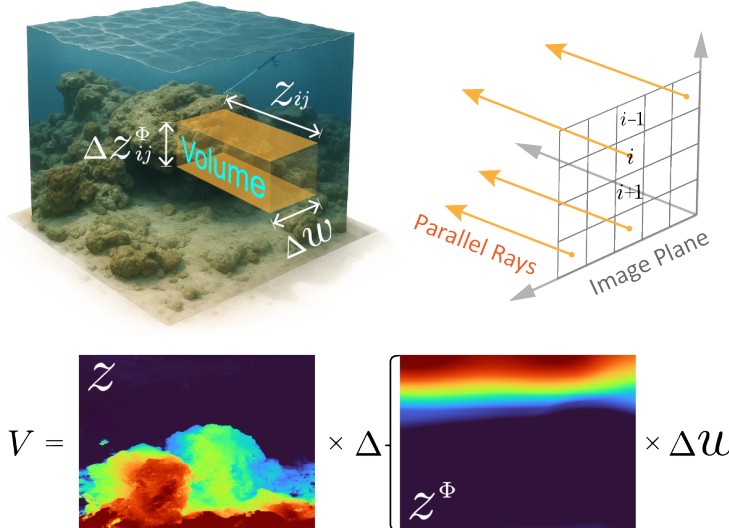

Figure 13: Illustration of subsurface water volume estimation under an orthographic projection assumption. Left: Each image pixel corresponds to a non-overlapping 3D cuboid, defined by its horizontal extent $z_{i,j}$, vertical height $\Delta z_{i,j}^{\Phi}$, and pixel-aligned width $\Delta w$. Right: All rays are assumed to be parallel and aligned along the horizontal axis, consistent with an orthographic camera model.

is denoted as $z_{i,j}$. Meanwhile, the downwelling attenuation module estimates the vertical distance from the ray to the water surface, denoted as $z_{i,j}^{\Phi}$.

To convert image coordinates into real-world dimensions, we assume that a physical scale prior is available—that is, the real-world width corresponding to the image field of view is known or can be estimated. In our case, for the underwater scene captured in Okinawa, we set the real-world width to six meters based on a rough field measurement of a subsea anchor present in the scene. As illustrated in Figure 13, each pixel is thus associated with a unique cuboid in 3D space, defined by its horizontal extent $z_{i,j}$, vertical height $\Delta z_{i,j}^{\Phi}$, and lateral span $\Delta w$, where $\Delta w = W_{\text{real}}/W_I$ is the real-world width per pixel assuming uniform spacing. Since adjacent rays may penetrate to different depths, we compute the $\Delta z_{i,j}^{\Phi}$ of each vertical water column by taking the difference in downwelling depth between vertical neighboring pixels:

$$\Delta z_{i,j}^{\Phi} = \begin{cases} z_{i,j}^{\Phi}, & \text{if } i = 0 \\ \max(0, z_{i,j}^{\Phi} - z_{i-1,j}^{\Phi}), & \text{otherwise} \end{cases} \tag{62}$$

The resulting per-pixel water volume is given by:

$$V_{i,j} = z_{i,j} \cdot \Delta z_{i,j}^{\Phi} \cdot \Delta w \tag{63}$$

Summing over all pixels across the image yields the total visible subsurface water volume:

$$V_{\text{total}} = \sum_{i=1}^{H_I} \sum_{j=1}^{W_I} z_{i,j} \cdot \Delta z_{i,j}^{\Phi} \cdot \Delta w \tag{64}$$

Applying this formulation to our underwater scene, and using the anchor-based scale prior of six meters image width, we estimate the total traversed water volume visible in the field of view to be approximately 330.68 cubic meters.

Although we introduce certain approximations and simplified assumptions, our method retains the ability to recover physically interpretable quantities. Unlike previous approaches that operate entirely within virtual space, I²-NeRF incorporates geometric alignment with physical measurements. This enables the estimation of real-world metrics, such as volumetric quantities, in a manner consistent with the physical structure of the environment.

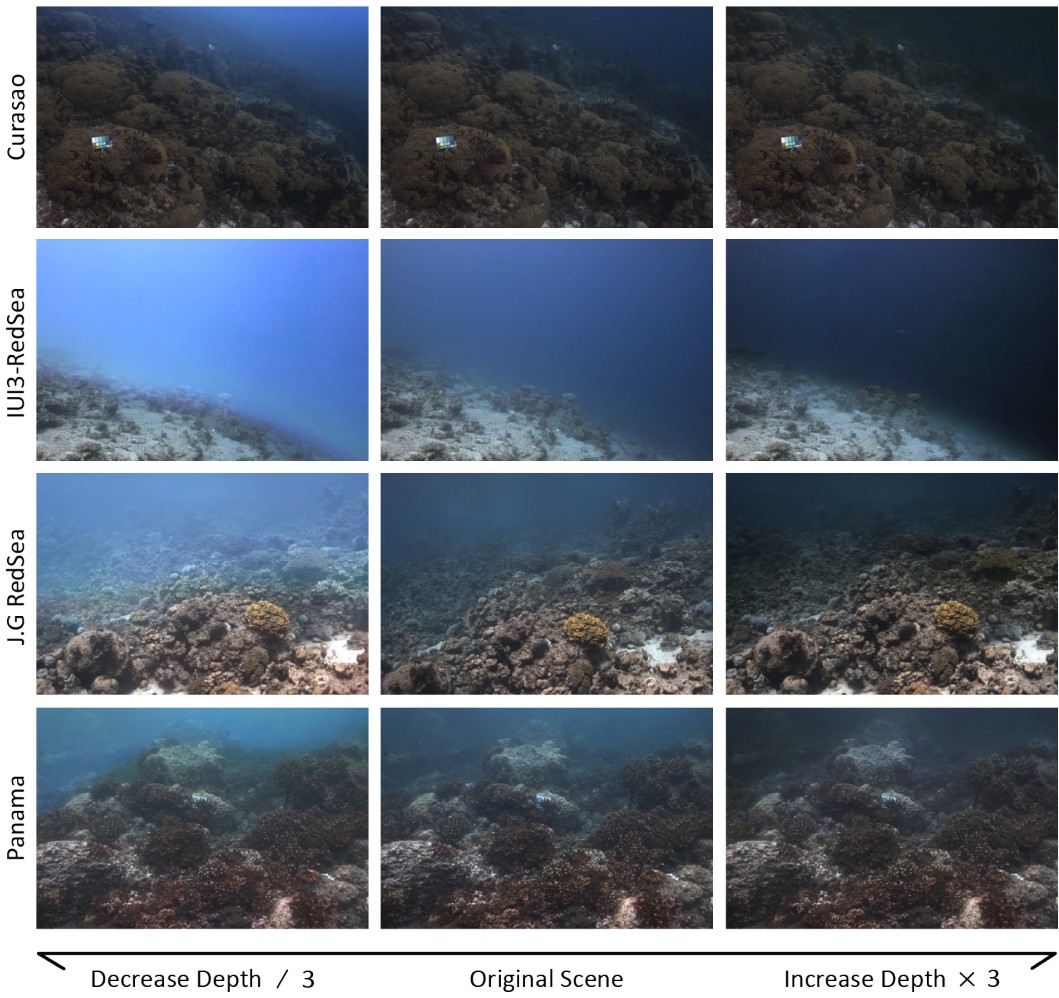

Figure 14: Physically-grounded synthesis under varying water depths. By modifying the downwelling depth estimated in our model, we simulate the same scene under shallower (left) and deeper (right) water conditions.

## H.2 Physically-Grounded Synthesis

Our framework reveals interpretable physical properties such as attenuation and scattering coefficients, sunlight parameters, and distances in the 3D volume, which enables controllable and physically grounded scene synthesis. In this example, we simulate the same scene under shallower and deeper water by scaling the downwelling depth by a factor of one-third and three, respectively. These changes affect scattering and ambient illumination, resulting in noticeable variations in color tone, contrast, and visibility. As shown in Figure 14, our model enables realistic appearance modulation that reflects physically plausible modulation in underwater conditions. This supports the potential of our approach not only for geometry-aware rendering but also for physically grounded scene editing aligned with underwater optics.

## I Implementation Details

In our implementation based on the Zip-NeRF codebase [8], we set the number of proposal sampling points to 128, the number of NeRF sampling points to 32, and the number of media upsampling points to 32, treating them equally to object samples. The sampling hierarchy level is set to 2. We employ two separate hash grid encoders for object and media components, which introduces a slight

increase in training time but is necessary to preserve their distinct geometry and spatial distributions. The output dimensionality of both hash encoders is set to 256.

Under scattering conditions, to enforce per-ray constant medium properties (i.e. $\sigma^{\mathrm{attn}}$, $\sigma^{\mathrm{scat}}$, and $z^{\Phi}$) as in the RUIF simplification [2], we pool the per-sample predictions along each ray and use the pooled values in subsequent volume rendering. For low-light conditions that favor spatially varying medium distribution, we instead preserve the per-sample media densities, allowing each sampled point to maintain its distinct value during volume rendering.

For the LOM dataset [16], we use a batch size of 4096. For the SeaThru-NeRF dataset [45], the batch size is set to 2048, and for our captured underwater scenes, it is set to 1024. Each batch size is scaled proportionally to the total number of pixels in the respective dataset. The maximum number of training steps is set to 25,000. We use the Adam optimizer with an initial learning rate of $10^{-2}$ and a final learning rate of $10^{-4}$.

Input images are loaded into the NeRF model at their original resolutions. We follow the same train-test splits provided by the datasets. For the SeaThru-NeRF dataset [45], we adopt the common practice in previous studies by assigning every 8th view as the test view. All experiments are conducted on a single NVIDIA RTX A6000 Ada GPU. Training time are provided in Appendix L.

# J  Sanity Check

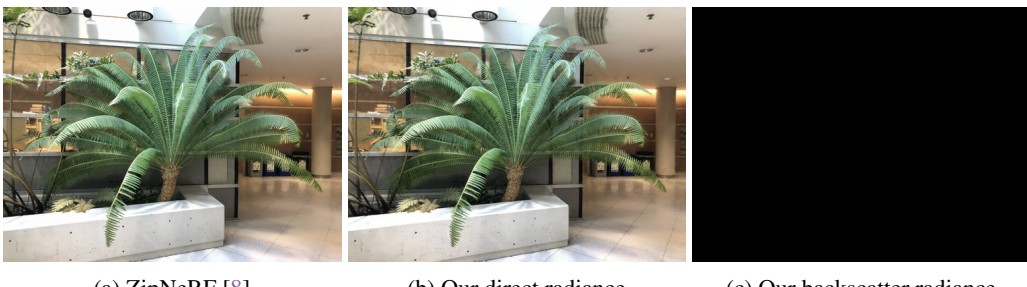

(a) ZipNeRF [8]  (b) Our direct radiance  (c) Our backscatter radiance

Figure 15: Sanity check on clear-air Fern scene from NeRF dataset [56].

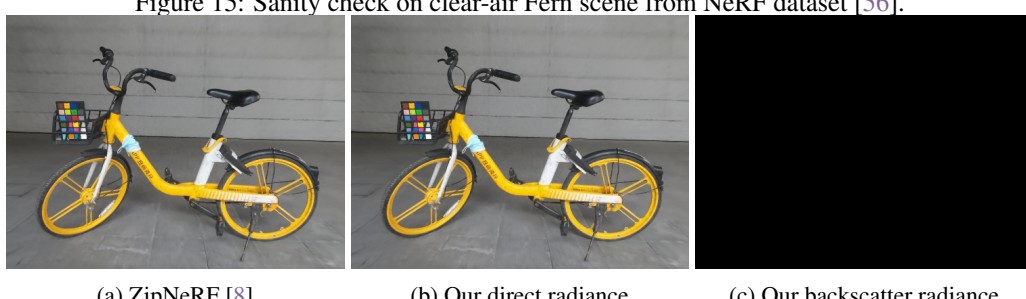

(a) ZipNeRF [8]  (b) Our direct radiance  (c) Our backscatter radiance

Figure 16: Sanity check on clear-air bike scene from LOM dataset [16].

Following SeaThru-NeRF [45], we conduct a similar sanity check by applying our model to clear-air scenes while enabling the media branch. As shown in Figure 15 and Figure 16, we compare the novel-view synthesis of direct radiance and backscatter radiance of our method with the ZipNeRF [8] baseline. Our approach produces no backscatter component when scattering is absent in clear-air scenes.

# K  Multiple Running

Table 6: Multiple running statistics on the *bike* and *Curasao* scenes.

| Metrics | PSNR | SSIM | LPIPS |
|---------|------|------|-------|
| bike | $22.82 \pm 0.05$ | $0.798 \pm 0.01$ | $0.279 \pm 0.00$ |
| Curasao | $30.55 \pm 0.04$ | $0.860 \pm 0.01$ | $0.142 \pm 0.00$ |

Due to computational costs, we present statistical results based on multiple runs for two scenes: the low-light *bike* scene from the LOM dataset [16], and the underwater *Curasao* scene from the SeaThru-NeRF dataset [45]. As shown in Table 6, we report the mean and standard deviation of PSNR, SSIM, and LPIPS metrics across five independent runs.

## L    Training Time Analysis

In this section, we present a training time analysis of our method and baseline approaches on the low-light [16] and underwater [45] datasets. As shown in Figure 17, our method achieves fair training efficiency, benefiting from hash encoding to accelerate convergence. Gaussian Splatting-based approaches demonstrate significantly faster training times due to their rasterization-based rendering process.

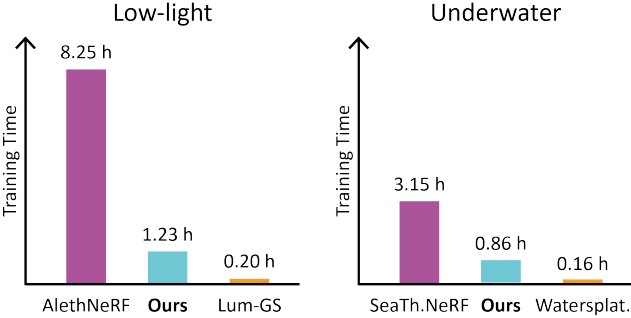

Figure 17: Training time analysis of our method and baseline approaches on low-light and underwater scenes. For each model, the training time is recorded using a single NVIDIA RTX A6000 Ada GPU.

## M    Dataset Diversity

In this study, we evaluate our method on benchmarks that encompass various types of media degradation. For low-light scenes, we use the LOM dataset [16], which contains five real-world scenes captured as underexposed multi-view images, along with corresponding ground-truth well-lit images. For underwater scenes, we employ the SeaThru-NeRF dataset [45], which includes four real-world underwater scenes without ground-truth clean object radiance. Additionally, we capture two underwater scenes ourselves—one under well-illuminated conditions and another exhibiting a hybrid of low-light and underwater scattering. For hazy environments, we utilize the synthetic hazy Fern scene provided in [45].

The datasets used in our evaluation cover a wide range of degradation scenarios, allowing for a fair comparison and clearly demonstrating the superior performance of our method over baseline approaches. Nonetheless, evaluating on a larger number of scenes would further strengthen the generality of our conclusions. Given the limited availability of multi-view datasets with media degradation, collecting more real-world data, especially with paired degraded and ground-truth images, is considered as future work.

## N    Future Works

Apart from real-world paired datasets that facilitate the study of 3D reconstruction under degraded conditions, several key directions deserve further investigation. First, integrating physically grounded 3D representations into large-scale foundation models offers a promising path toward spatially and physically aware reasoning. Such integration could enhance the model's ability to understand geometry, material properties, and environmental interactions in complex scenes.

Second, there remains a lack of evaluation metrics that reflect physical correctness. Existing image-based metrics such as PSNR and SSIM are limited to appearance similarity and fail to capture geometric fidelity or physical plausibility. While comparisons based on point cloud or mesh completeness are more suitable for geometric evaluation, they require high-quality ground-truth data and are not applicable to implicit volumetric representations, which do not produce point- or surface-based

outputs. Developing evaluation metrics that account for radiometric consistency, media-aware realism, and structural integrity is increasingly important as scene representations move toward physically based modeling.

Finally, most reconstruction pipelines assume access to accurate camera poses. However, in real-world degraded environments, such priors are often unavailable. Traditional structure-from-motion methods and deep learning-based pose estimators struggle under strong scattering, blur, and noise. Recent advances in pose-free methods alleviate this dependency by learning jointly from unposed images, but their performance remains limited in heavily degraded conditions. Improving the robustness of such methods under physical degradation is critical for real-world deployment.

