# OpenReview forum: "I2-NeRF: Learning Neural Radiance Fields Under Physically-Grounded Media Interactions"
_NeurIPS.cc/2025/Conference — NeurIPS 2025 poster_

### Official Review · Reviewer_TDRQ · 2025-06-29

**Clarity:** 3
**Significance:** 3
**Originality:** 3
**Rating:** 4
**Confidence:** 4

**Summary:**

The paper proposes a NeRF variant (built on ZipNeRF) that adjusts the sampling strategy along each ray, and the forward model, to flexibly model scenes with translucent media like seawater or fog. The paper also proposes to model low-light scenes in the same framework, by modeling a virtual absorptive medium. The paper also introduces several regularizers in the loss function, to encourage a clean separation of their separate models for objects and translucent media. This enables more physically plausible and separable (object vs medium) reconstructions compared to baselines that are designed separately for each task (underwater NeRF, hazy/foggy NeRF, and low-light NeRF).

**Questions:**

Some questions are embedded in the discussion of weaknesses; remaining questions are below. For the rebuttal, my primary concerns are the points mentioned under weaknesses.
- Line 131: Should “regresses” be “reduces”? Similarly in equation 2 I think there is an $i$ that is supposed to be in the subscript instead of on the main level.
- Are the $u_j$ in equation 8 and the $u_J$ in line 196 related to each other? I’m concerned this might be a notation clash because the former would appear to be a sample position and the latter would appear to be an illuminance value.
- Will the novel underwater dataset be released? The authors state that code will be released, but do not specify release of data.

**Ethical Concerns:**

["NO or VERY MINOR ethics concerns only"]

**Final Justification:**

The authors have engaged with my review during the rebuttal and discussion phase, and have promised to discuss some assumptions and inefficiencies in their method that may be removed in future work. My only remaining concern that was not fully addressed is about the paper naming/framing that focuses on isometry and isotropy, which I feel does not really reflect the focus of the work, which is on modeling absorptive and scattering media. However, this naming concern is not significant enough for me to change my score, so I continue to recommend acceptance.

**Limitations:**

Limitations are adequately discussed. One question here is that the authors mention that their method takes longer training time than some others, but doesn’t specify a rough estimate of what the training time is or how it might be improved.

**Quality:**

3

**Strengths And Weaknesses:**

Strengths:
- Section 3.2 is very clearly written, and clarified many of my questions after reading the abstract and introduction. I also found the derivations and modeling descriptions in section 3.3 to be clear and elegant.
- The experiments are comprehensive, including 3 types of challenging scenes (underwater, low light, and hazy) as well as ablation studies. The paper shows competitive and in some cases SOTA performance on these three types of scenes in terms of quantitative metrics, and shows qualitatively better scene decomposition into object vs medium layers, compared to baselines.
- The evaluation includes both existing public datasets and a new underwater dataset; if this dataset is released it would also be a valuable contribution.

Weaknesses / Suggestions for improvement:
- The $I^2$ in the title would appear to be referring to “isometric” and “isotropic”, but (a) this title choice is never actually stated so I am just inferring it, and (b) I’m not sure that isometric and isotropic really describe the method clearly. As I understand it, “isometric” means having the same dimensions as something else, and “isotropic” means having some property remain fixed in all directions. It’s not clear how either of these is descriptive of the proposed method. I think the connection for “isometric” might be that the proposed method promotes metric consistency with the real world, up to some global scaling, but this is only demonstrated in the context of the depth of an underwater scene, not as a general property of the method. I’m not sure what is “isotropic” about the method, especially since it actively considers one-dimensional depth of a medium (water, mist, etc.) when modeling scattering, and this is a quantity that only makes sense in the vertical dimension, not all 3 dimensions. Note that this is not so much a weakness of the proposed method or results, as it is a concern over how the method is presented. Relatedly, the second bullet point contribution on line 78 describes the sampling strategy as “metric-preserving” but it’s not clear what this means (what’s the connection between the sampling strategy and the metric accuracy of the reconstruction? In the results it seems that sampling in the medium allows for better decomposition of the scene into medium and object layers, but I’m not sure why this is “metric-preserving”).
- For most settings (at least for underwater and fog/haze scenes, perhaps also for low-light scenes?), it seems that the properties of the medium are fixed regardless of spatial position. This makes perfect sense as an approximation that is likely close to accurate in reality. But, if this is the case, it seems wasteful to spend a lot of ray samples in the medium, and to spend a lot of memory representing the density parameters throughout the medium. Why not just model the distance the ray travels between the camera and the object, and apply a closed-form estimate with a single set of spatially-invariant medium parameters instead of estimating an integral with individual samples?
- The dependence of downwelling attenuation on vertical depth seems to be modeled based on a single depth $z^\Phi$ per ray. For rays that are not perfectly horizontal, won’t the vertical depth vary along the length of the ray? Shouldn’t this be included in the model?

Minor weaknesses / suggestions for improvement:
- The term “matting” is used several times between the abstract and introduction, but not defined until the top of page 4. I thought this was a typo / misspelled other word until I had already read ⅓ of the paper.
- Figure 1 uses blue points for samples in the medium, and orange points for samples of the object. This should be included in the caption rather than deferred to lines 42-44.
- The discussion of related work is brief but generally sufficient. My suggestion here is to add a sentence or two in each paragraph explaining how the methods you just described are related to your proposed approach: what is similar and what is novel in your approach.
- The sampling strategy in equations 7 and 8 should be augmented with a description of how they result in increased medium sampling only in front of the object (between the object and the camera) instead of on both sides of the object. It’s not obvious from the equations that this is the case.
- $\mathcal{L}_{recon}$ and $\mathcal{L}_{geo}$ are losses borrowed from prior work, but their formulas should still be reproduced in the main text for the readers’ convenience.
- Line 203 describes avoiding “the trivial solution”--please specify what this is. Is it both object and medium densities going to zero?
- Line 206-207 describes a “monotonic decay in media density” but this statement is unclear without knowing what the monotonicity is with respect to. Similarly, the notation on line 208 is unclear; it uses a ReLU with 3 arguments but usually ReLU takes a single argument and returns $max(\cdot, 0)$.
- Line 212 is a sentence fragment; the meaning is not clear. I suspect a word is missing?
- Can the authors provide some description/intuition for why $\lambda_{struct}, \lambda_{trans}$ are both zero for both underwater and hazy scenes?
- What is the bright channel in a low-light scene? Is there a different color space being used other than RGB? What is $B^c$ in equation 13?
- Line 260 describes a baseline as “the GS-based method” but in the associated figure (fig. 4) the label is “Watersplatting”--making these namings consistent would help readers understand the figure.

---

> ### Author Rebuttal · Authors · 2025-07-26
>
> **We would like to express our deepest gratitude for receiving such detailed and thoughtful review of our paper. We greatly appreciate the concrete comments and suggestions.**
>
> ### **W1. Isometry and Isotropy**
>
> Thank you for the insightful question. In our paper, we introduce the notation $I^2$ to denote two complementary desiderata in neural field modeling: isometry, the ability to recover metric distances in space, and isotropy, the capacity to model properties uniformly in all directions.
>
> In vanilla NeRF, weights and density samplings concentrate exclusively on object surfaces, treating free space as void. This simplification makes rendering efficient, but it also erases any notion of “distance through air”: an object one meter away is treated identically to one ten meters away, and transmittance through the intervening volume is ignored. In situations where accounting for the space in between is essential, such as when optical scattering occurs in the surrounding medium, this becomes a critical limitation. Works such as Aleth-NeRF [1] and SeaThru-NeRF [2] attempt to mitigate this by assigning additional weights along the ray to capture radiance contributions beyond object.
>
> We consider this challenge as an opportunity toward a fully metric-aware method by explicitly allocating samples in non‑object regions, thereby restoring optical length scales. In an absorption field in low-light scenes, distant objects appear darker in accordance with a distance‑dependent attenuation law; in a scattering field, further objects appear blurrier and shift toward the scattering color. By letting the model attend to these samples, we recover an approximate metric estimation (up to a global scale) under scattering conditions, and similarly reflect physical distance as low‑illumination in low-light scenarios.
>
> By definition, isometry refers to a mapping that preserves distances. In our work, we want the neural field to recover physical length scales throughout the volume rather than collapsing all free space into an inert background. To achieve this, we allocate sampling points not only on object surfaces but also in the surrounding medium. The network then learns to accumulate attenuation and scattering along each ray, effectively measuring line‑of‑sight (LoS) distance and downwelling depth in scattering environments. Although we do not yet recover absolute metric units, this sampling lets distant objects exhibit the correct degree of blurring, color shift, or illumination loss that arises from true volumetric propagation.
>
> Isotropy denotes uniform behavior in all directions. In vanilla NeRF and existing scattering‑aware extensions [2–4], media properties (color) are learned implicitly along each viewing ray, without a coherent physical model for propagation in other directions. We extend the Beer–Lambert attenuation law both along the LoS and vertically toward the media boundary (i.e., the water surface), making the “downwelling distance” an explicit, learnable quantity. Although we cannot apply the optical attenuation in all 3 dimensions, the law is consistent along the ray and in vertical direction.
>
> By combining these two concepts, we introduce the $I^2$ concept, aimed at enabling AI models not only to perceive the spatial positions of objects but also to sense and interpret whole space metric. This capability is critical for advancing current AI systems, particularly in the context of Embodied AI, World Models, and Generative Models. These areas demand a deeper integration of spatial awareness and metric understanding, which our $I^2$ concept seeks to provide.
>
> In submitting this work to NeurIPS, a conference that brings together experts from Mathematics, Physics, Machine Learning, and Embodied AI, we hope to gather insights from diverse perspectives and engage in interdisciplinary discussions. We would like to discuss with NeurIPS's community and explore how the proposed $I^2$ concepts can contribute to encoding a complete metric perception of 3D space. Through constructive feedback, we seek to examine the potential for establishing a comprehensive framework encompassing key concepts, benchmark datasets, evaluation metrics, and technical methods for this emerging research direction.
>
> ### **W2. Closed-form solution between camera and object**
>
> Thank you for raising this important question. In scattering conditions, where the media color and media scattering coefficients are constant, we find it possible to apply a closed-form solution to directly calculate the backscatter radiance component and add it to the direct radiance to compute the pixel color. In this condition, we only need scalar values for each ray, and this will approximately reduces 4 GB of GPU memory given the training batch size of 2048 (Supp Line 377). We think the current sampling design has three advantages presented below, and we would like to hear your opinion.
>
> First, as noted in W1, the objective of our work is to explicitly model the entire 3D space, including both the dense object and the surrounding volume. This comprehensive modeling enables metric measurements and fosters a physically grounded understanding of the 3D space. Motivated by this intention, we explicitly allocate media sampling points throughout the space.
>
> Second, the inherent ill-posedness of the inverse rendering problem requires separating the direct and backscatter radiance given only the degraded observations. Explicit spatial sampling, such as that provided by RSU, is therefore crucial for the successful optimization of the NeRF model. Accurate media sampling helps prevent the collapse of the media field during learning. Collapse can results a failure case where backscatter radiance is incorrectly attributed to object direct radiance, leaving the medium field empty (as demonstrated in Fig. 6 left and Supp. Fig. 1 without RSU). This collapse becomes particularly significant when the ambient light has close color to the scene color, such as in shallow underwater scene (Supp Fig. 1) and hazy scene (Supp Fig. 4). In such case, the guidance from media geometry positions becomes important for convergence.
>
> Finally, in low-light environments (no scattering only absorption), a spatially variant absorption field is required (Line 226) due to the highly-dynamical scene illumination. As a result, the media density becomes inconsistent along each ray; therefore, there is no closed-form solution in low-light conditions. We visualized the density distribution of two representative rays in Fig. 3, the absorption media field monotonic decays and terminate at the object surfaces.
>
> ### **W3. Rays Assumption**
>
> We appreciate your thoughtful question. Under scattering conditions, we build upon the Revised Underwater Image Formation (RUIF) model [5], which simplifies the computation of backscattered radiance by assuming that all rays are horizontal (as noted in Line 59 of our Supp).
>
> In precise underwater imaging, rays are not parallel in horizontal directions, and this introduces an inclination angle $\theta$ in the nadir direction. As a result, the backscatter radiance at each point along the ray becomes dependent on $\theta$, substantially increasing the ill-posedness of the problem (Eq. 1 in RUIF paper [5]). To make the problem tractable, RUIF [5] proposes a simplified formulation by setting $\theta=90^{\circ}$, effectively removing the inclination dependence as shown in Supp Eq.2. (Eq.2 in RUIF paper as well). This approximation is widely adopted in previous studies [2-4]. Consequntly, each point along the ray will have the approximated same downwelling depth $z^{\Phi}$. **We will add this explanation in Sec 3.1 to emphasize this horizon simplification**. More elaboration of RUIF [5] is provided in Supp Sec. A.2.
>
> ### **Q1**
> Should be "reduces" and $\hat{C}_i^{obj}$
>
> ### **Q2**
> It is a notation clash and we will change symbol.
>
> ### **Q3**
> Thank you for the reminder. We will definitely release the dataset.
>
> ### **Minor: trivial solution**
>
> It should be rephrased to "helps avoid the trivial solution of zero media density" in Line 203.
>
> ### **Minor: monotonic decay**
>
> Refer to W2., we have spatially-variant absorption media in low-light conditions. We want this media field to be monotonic decay as shown in Fig.3 such that it ends on the object surface. Line 208 has typo, it should be $ReLU(\sigma_{k,i}^{med}-\sigma_{k,{i-1}}^{med})$.
>
> ### **Minor:  $\lambda_{struct}$ and $\lambda_{trans}$ are zero in underwater and hazy scenes**
>
> $\lambda_{struct}$ is weight of compensated SSIM loss, which takes into account the luminance and contrast differences, and are particularly important in low-light scenes where the rendered well-lit and low-light observations present large discrepancy. In scattering conditions, such SSIM do not really help to improve the scene reconstruction when there are no large illumination/contrast gaps. $\lambda_{trans}$ is activated when there are any transmittance prior. In low-light scenes, we can unsupervisedly retrieve such prior from BCP to facilitate the learning of spatial-varient absorption media. In underwater/haze scenes, there is unfortunately no such prior available, so we disable this loss term.
>
> ### **Minor: BCP definition in Eq. 13**
>
> We provide detailed derivation of BCP in Supp Sec A.4. $B^c$ means the channel-wise value of ambient light, $c \in (r,g,b)$ defined in Eq.13.
>
> **Due to the word limitations, we answer all questions, and will carefully revise the paper according to your rest of the suggestions.**
>
> ### Reference
>
> [1] "Aleth-nerf: Illumination adaptive nerf with concealing field assumption." AAAI, 2024.
>
> [2] "Seathru-nerf: Neural radiance fields in scattering media." CVPR, 2023.
>
> [3] "Neural underwater scene representation." CVPR, 2024.
>
> [4] "Watersplatting: Fast underwater 3d scene reconstruction using gaussian splatting." 3DV, 2025.
>
> [5] "A revised underwater image formation model." CVPR, 2018.

---

> > ### Comment · Reviewer_TDRQ · 2025-08-05
> >
> > Thanks for the detailed response to my review. The word limits are unfortunate but they are what they are; please do incorporate the remaining suggestions in the revised paper. Below I give a bit more detail on a few points raised in the response, but overall I am inclined to maintain my score.
> >
> > Naming: I appreciate why the authors see their work as promoting isometry and isotropy, but I'm still not really convinced that these are the most appropriate descriptors of the method. To me it seems the key innovation is a physics-based modeling of media rather than only of objects. Without any absorptive or scattering media present, prior models also achieve metric-preserving reconstructions, so I think the incorporation of physics-inspired media modeling is really the key novelty of this work.
> >
> > Modeling media parameters as fixed vs spatially-varying: For settings like low-light scenes in which the learned media parameters are actually spatially varying, it makes sense to model the entire volume--though even in this case the same effect could perhaps be modeled as a fixed absorption parameter with a nonlinear effect as a function of ray distance. There might also be other applications (e.g. microscopy, optical tomography) where there really is spatially varying absorption and scattering in the medium. However, for the underwater and fog/haze settings, it seems there would be a substantial improvement in memory and computation cost by assuming constant media parameters. This might also have the bonus of introducing a form of regularization since it reduces the number of free parameters, so it could even help stabilize the reconstructions under sparse-view conditions. I think this would be an interesting direction to explore, but it's ok if it is beyond the scope of the current paper (e.g. this could be worth mentioning in the discussion as potential future work).
> >
> > Please do add the discussion of the assumption that rays are horizontal. I understand if this assumption was borrowed from prior work, but I think it is worth at least discussing what would be involved in removing it. Given that your method does take samples along the ray in the medium, I could imagine incorporating this within-ray depth variation into the modeling would be possible for your method.

---

> > > ### Author Response · Authors · 2025-08-06
> > >
> > > Thank you very much for your invaluable comments and suggestions.
> > >
> > > Our physics-inspired media modeling was initially motivated by the goal of enabling isometry and isotropy in 3D reconstruction. Building on this foundation, we further extended our work of media modeling to support estimation of water depth (Fig. 4, Supp. Fig. 3), rendering of physically plausible media color (Fig. 4, Supp. Fig. 4), synthesis of novel views under varying water depths (Supp. Sec. H, Supp. Fig. 8), and modeling of hybrid media scenarios (Supp. Sec. G.4, Supp. Fig. 6).
> > >
> > > To advance this direction, we hope to engage with experts at NeurIPS like you and seek for discussions toward a more complete perception of 3D space. We consider our current work a first step and are eager to explore further developments.
> > >
> > > We also appreciate your suggestion to explore broader applications, such as microscopy and optical tomography that open promising directions to extend our study. The idea of applying a regularization term to the media field is particularly interesting for improving memory and computational efficiency. We will include this in our discussion of future work and its potential extensions.
> > >
> > > We will also add a discussion on the horizontal simplification in backscattering to thoroughly present our paper. As you pointed out, explicitly modeling the downwelling distance sounds particularly interesting and feasible, and we will highlight this as a promising direction for future studies.
> > >
> > > Thank you again for your time and thoughtful input, which have greatly helped us improve our work.

---

> > > > ### Comment · Reviewer_TDRQ · 2025-08-07
> > > >
> > > > Thanks for clarifying, and I appreciate that the authors will include these discussions in the revision. Overall I am satisfied, with the one exception that I still feel that the naming/framing focus on isometry and isotropy is not very precise as a descriptor of this work. Something focused on absorptive/scattering media would be more precise. However, this is a minor point so I continue to recommend acceptance.

---

> ### Author Response · Authors · 2025-08-07
>
> Thank you for taking the time to review our paper! We look forward to speaking with you if we have the chance.

---

### Official Review · Reviewer_teTt · 2025-07-01

**Clarity:** 3
**Significance:** 2
**Originality:** 3
**Rating:** 5
**Confidence:** 2

**Summary:**

The paper proposes a new method for modeling non-clear air medium for 3D reconstruction through NeRFs. This is done through an explicit modeling of the medium, its density, absorbtion and emitted radiance and a stratified sampling mechanism that increases the number of samples taken outside of the surface of the object. The experimental results show a significant improvement in modeling under-water, hazy and dark scenes.

**Questions:**

- What is the runtime of the model for an average-sized scene (~50 images)
- In table 3 the structural similarity loss seems to affect the rendering quality by a lot, why is that? Usually this loss adds <0.5 db when added to the photometric reconstruction loss.

**Ethical Concerns:**

["NO or VERY MINOR ethics concerns only"]

**Final Justification:**

All of my concerns were sufficiently addressed in the rebuttal. I believe the paper has practical contributions to the field, therefore I am still leaning toward acceptance.

**Limitations:**

The limitations are discussed.

**Paper Formatting Concerns:**

None.

**Quality:**

3

**Strengths And Weaknesses:**

Strengths:
- The proposed model follows the physical modeling of the Beer–Lambert Law and allows for backscatter color and light absorbtion
- The proposed model allows for isotrpoy and isometry that leads to prediction of actual depth, given the scale, in non-clear air medium.
- The model achieves 3D reconstruction in challenging scenes with minimal additional supervision from depth foundation models.

- Specifically the method improves on previous methods like Sea-Thru NeRF and Watersplatting by being applicable to different challenging mediums like hazey and dark settings (not just water), handling vertical attenuation on top of LoS, having a more effective sampling than object surface sampling, and the ability to predict actual depth that emerges from accurate physical modeling. These benefits show as qualitative improvement in the results.

Weaknesses:
- The stratified upsampling adds to the complexity and the rendering time needed for the NeRF.
- I'm curious to see if the model still performs well in normal scenes with mostly having clear air medium. My concern is that the medium prediction branch of the network might add to the complexity of modeling simpler scenes and overfit.
- The quantitative results are competitive with Watersplat, however the qualitative results show a better reconstruction and depth estimation. Why is the better rendering quality not reflected in the image quality metrics?

---

> ### Author Rebuttal · Authors · 2025-07-26
>
> **We sincerely appreciate the valuable comments and suggestions that help strengthen our paper. We conduct additional experiments below to show the efficiency of the proposed method.**
>
> ### **W1. Complexity of RSU**
>
> The additional GridEncoder for media position encoding and the media radiance in volume rendering process increases both the complexity and rendering runtime. However, in adverse media conditions, such complexity tradoff are necessary to maintain the high-quality restoration and geometry reconstruction of the clean scenes. We conduct additional experiments and demonstrate that **The incorporation of medium modeling introduces a managable computation overhead.**
>
> Specifically, to investigate the computation overhead, we conduct experiments below by training ZipNeRF [1], which is our code base (Line 243), on both the "bike" scene in LOM dataset [2], and “Curaçao” scene in SeaThru-NeRF dataset [3] to compare the training and rendering time. $I^2$-NeRF introduces a reasonable computational overhead of approximately 20–30\% (10–20 minutes) additional training time compared to the baseline, while achieving substantial performance improvements under adverse conditions such as low light, haze, underwater environments, and even hybrid cases like low light + underwater shown in Supp Sec G.4 and Supp Fig. 6.
>
> ### Table 1: Computation and memory cost on "bike" of LOM dataset [2]
> | Method   | Train | Render | GPU Memory (training) |
> | -------- | -------- | --------- | --------- |
> | Baseline [1]  |    0.95 Hours      |     1.29 FPS      |    \~ 10 GB   |
> | Ours     |     1.25 Hours   |    0.58 FPS      |     \~ 22 GB     |
> (Training batch size of 4096, training epoch 20,000, image resolution 500 x 375)
>
> ### Table 2: Computation and memory cost on "Curaçao" of SeaThru-NeRF dataset [3]
> | Method    | Train | Render | GPU Memory (training) |
> | -------- | ------- | ------- | ------- |
> | Baseline [1]  |  0.69 Hours  |  0.041 FPS  |  \~ 5 GB  |
> | Ours |  0.88 Hours  |  0.028 FPS  |  \~ 10 GB  |
> (Training batch size of 2048, training epoch 20,000, image resolution 1776 x 1182)
>
> ### **W2. Performance in clear-air condition**
> We sincerely appreciate this comment, as it highlights the importance of validating the generalization of our model to clear-air environments. We conduct additional experiments showing below. **In clear air condition, our model performs very close (< 1\%) to the original baseline [1]**. Moreover, the backscatter radiance estimated by our model is nearly zero, indicating that **the media branch does not learn any signal when scattering is absent in the scene**.
>
> Specifically, we conducted sanity checks by applying our model to well-lit images from the “bike” scene in the LOM dataset [2] and the original “fern” scene from the NeRF dataset [4], using ZipNeRF [1] as the baseline for comparison.
>
> ### Table 3: Sanity check on "bike" of LOM dataset [2]
> | Method   | PSNR | SSIM | LPIPS |
> | -------- | -------- | --------- | --------- |
> | Baseline [1]  |  31.40 |  0.884  |  0.163 |
> | Ours     |  31.32  (-0.08) |   0.879  (-0.06)  |  0.166 (-0.03) |
>
> ### Table 4: Sanity check on  "fern" of NeRF dataset [4]
> | Method    | PSNR | SSIM | LPIPS |
> | -------- | ------- | ------- | ------- |
> | Baseline [1]  |  24.95  |  0.807  |  0.169  |
> | Ours |  24.80 (-0.15) |  0.799 (-0.08) |  0.171 (-0.02) |
>
> ### **W3. Experiment results**
>
> Thank you for the comment. We would like to clarify the key differences in evaluation settings and the inherent challenges of the current community that hinder us from performing fair restoration evaluations.
>
> In the low-light dataset [2], pixel-aligned (paired) low-light and well-lit images are available for testing. This allows us to render novel-view clean (well-lit) images and compute evaluation metrics directly against GT clean views. As shown in Tab. 2, the proposed method achieves SOTA performance under this setting.
>
> In contrast, real-world haze and underwater datasets, such as the SeaThru-NeRF dataset [3], only provide degraded observations, and there is no pixel-aligned clean images for evaluation. It is difficult to remove haze or oceanic water from the real-world environment to capture pair-wised images of clean views. As a result, we can only render degraded novel views (with haze or water) and compare them against the degraded inputs. Such novel-view synthesis results are shown in Table 1.
>
> Due to the absence of pair-wised clean observations in real-world scattering datasets, **Tab. 1 reflects only how well each method reconstructs the appearance of degraded scenes, as visualized in the novel view synthesis results shown in the first columns of Fig. 4 and Supp Fig. 3. However, Table 1 does not capture the accuracy of restored scene geometry (i.e., depth) or clean image synthesis (i.e., novel-view restoration), which is illustrated in the remaining columns of Fig. 4 and Supp Fig. 3.** When solely look at the novel view synthesis results (first column of Fig. 4) of our model and Watersplatting [5], the differences appear subtle. However, the rest columns of qualitative results on geometry and scene restoration clearly demonstrate that the proposed method provides more accurate geometry reconstruction and physically plausible estimation of media properties.
> **We will add this description to clarify the evaluation settings for each dataset in Sec 5.1. Line 249.**
>
> ### **Q1. Runtime of the model**
>
> For an average-sized scene (\~50 images) and a common resolution (\~720x540), training for a commonly employed iteration (\~20k), it will take roughly 1.2 \~1.5 hours to train our model.
>
> The average training time of our model is 1.23 hours on the LOM dataset [1] (compare to Aleth-NeRF [2] of 8.25 hours) and 0.86 hours on the SeaThru-NeRF dataset [2] (compare to SeaThru-NeRF [4] of 3.15 hours). The proposed method achieves substantially faster training speeds compared to previous NeRF-based approaches [2,4], benefiting from the hash encoding of spatially sampled features. In Supp Sec. K, we provide detailed information for the training time of the proposed method and baselines on both low-light and underwater scenes using the same GPU. **We will add a sentence in Line 246 to indicate the training time and refer readers to the corresponding section in supplementary material for more details.**
>
> ### **Q2. Structure similarity loss**
>
> We appreciate this question that helps improve the clarity of our paper. Structure Similarity Index Measure (SSIM) composes of luminance, contrast, and structure. It usually has limited effect on common reconstruction target. However, in the case of low-light restoration, the luminance and contrast between rendered well-lit images (by remove absorption field) and supervision signal from low-light observation are remarkably different. Low-light observations typically exhibit lower luminance (< \~0.1) and lower contrast (< \~1) compared to well-lit images, which generally have luminance \~ 0.5 and contrast \~ 10. These discrepancies cause the standard SSIM loss to struggle in recovering scene geometry from low-light inputs. Therefore, we design a **compensated SSIM loss** (Line 196) which takes into account the luminance and contrast discrepancy (Eq. 9). The detailed derivation of this $\mathcal{L}_{struct}$ loss is provided in Supp Sec E.3, and the effect of the parameter chosen is provided in Supp Sec F.2, visualized in Supp Fig 2. Without this compensated SSIM loss, our model cannot effectively restore well-lit scenes from low-light observations. As a result, the performance drops significantly without this loss term, as shown in Tab. 3. To distinguish between the conventional SSIM loss, **we will rename this loss term to Compensated SSIM Loss in Line 196, and denoted as $\mathcal{L}_{comp}$ for clearness**
>
> ### Reference
>
> [1] Barron, Jonathan T., et al. "Zip-nerf: Anti-aliased grid-based neural radiance fields." Proceedings of the IEEE/CVF International Conference on Computer Vision. 2023.
>
> [2] Cui, Ziteng, et al. "Aleth-nerf: Illumination adaptive nerf with concealing field assumption." Proceedings of the AAAI conference on artificial intelligence. Vol. 38. No. 2. 2024.
>
> [3] Mildenhall, Ben, et al. "Nerf: Representing scenes as neural radiance fields for view synthesis." Communications of the ACM 65.1 (2021): 99-106.
>
> [4] Levy, Deborah, et al. "Seathru-nerf: Neural radiance fields in scattering media." Proceedings of the IEEE/CVF conference on computer vision and pattern recognition. 2023.
>
> [5] Li, Huapeng, et al. "Watersplatting: Fast underwater 3d scene reconstruction using gaussian splatting." arXiv preprint arXiv:2408.08206 (2024).

---

> > ### Comment · Reviewer_teTt · 2025-08-04
> >
> > I thank the authors for their response. All of my concerns were sufficiently addressed. I believe the paper has practical contributions to the field, therefore I am still leaning toward acceptance.

---

> > > ### Author Response · Authors · 2025-08-04
> > >
> > > Thank you very much for your response and for taking the time to review our paper!

---

### Official Review · Reviewer_sN9j · 2025-07-02

**Clarity:** 2
**Significance:** 2
**Originality:** 2
**Rating:** 3
**Confidence:** 3

**Summary:**

This paper presents a method for learning NeRF representation in the presence of scattering media, which has applications in underwater, hazy, and low-light environments. A reverse stratified sampling technique is employed to focus sampling on the scattering medium along the ray. For underwater applications, a “downwelling distance” is predicted to guide the media color and is claimed to be related to the actual underwater depth.

**Questions:**

1. The concept of “downwelling distance” is confusing. From Figure 2, it indicates that $z^{\Phi}$ is a function of a 3D position and viewing direction. Is this view-dependent? If it is view-dependent, how does it relate to the physical depth?

2. In Figure 4, what is the difference between “Backscatter” and “Media Color”? If “Media Color” refers to $c^{med}(r)$ in equation 5, then at which point along the ray (of the pixel) is this color evaluated? Similarly, at which point is the “vertical depth” evaluated for each pixel? For a camera operating under perspective projection, rays are not parallel; therefore, they cannot all be parallel to the sea surface and maintain a constant depth.

3. On the other hand, if the Media Color $c^{med}$ is constant for each ray, it doesn’t seem necessary to apply the proposed RSU strategy to sample more points in the scattering media for the underwater application. The assumption of homogeneous distribution already implies that the media density is constant (in lines 215-216). Together with the constant color, the NeRF’s volume gathering wouldn’t change with the number of samples. Moreover, in Figure 6 (left), what is the “media density distribution” if it is already held constant?

4. As indicated in equation 5, $c^{med}(r)$ monotonically decreases as the “downwelling distance” increases. However, this does not correspond to Figure 4, where the bottom of the media appears brighter while the vertical depth is deeper. How is that interpolated?

The questionable concept prevents me from engaging with this paper; more technical details rather than just the formulation would help address my concerns.

**Ethical Concerns:**

["NO or VERY MINOR ethics concerns only"]

**Final Justification:**

After extensive discussion with the authors, several concerns were addressed and I raised my rating to borderline reject. Key agreements:

* The RSU (Reverse-stratified Upsampling) alters training gradients by querying density values with positional embeddings and pooling to keep them constant per ray. While seemingly inefficient, this approach is acceptable.
* View-dependent media densities: I accept this paradigm for underwater NeRF modeling despite limited justification. It is uncommon in other NeRFs and rare in volume-rendering literature, but plausible.

Remaining issue:

* View-dependent downwelling distance: As a central claim for demonstrating “isometry,” treating downwelling distance as view-dependent undermines its physical interpretation. The authors should either revise the model or provide experiments showing view consistency.

Overall, technical details are clearer, but I remain negative due to limited and insufficiently justified technical contributions. Given other reviewers’ positive recommendations, I could accept the paper for its inclusion of low-light scenes and the completeness of the work, provided the authors address consistency in the downwelling depth analysis in a revision.

**Limitations:**

yes

**Paper Formatting Concerns:**

N.A.

**Quality:**

2

**Strengths And Weaknesses:**

Quality: The method is fairly sound; however, the claims are not well supported due to the ambiguous key concept of “downwelling distance,” as elaborated in the questions section.

Clarity: While the paper is structured, it is difficult to distinguish its novelty from previous work. In Section 3, it would be beneficial to incorporate existing models in the "Preliminaries" and then highlight the original modifications. The key novelty is not presented clearly enough.

Significance: The quantitative evaluation shows only marginal improvement. It is challenging to assess the physically-grounded properties given the ambiguity of the key concept.

Originality: The novelty of this work compared to, for example, SeaThru NeRF [1], is not very clear, as the formulation in Section 3.3 shares many similarities. Two noticeable original designs are the reverse stratified sampling and the use of “downwelling distance,” both of which lack robust evaluation.

[1] SeaThru-NeRF: Neural Radiance Fields in Scattering Media

---

> ### Author Rebuttal · Authors · 2025-07-26
>
> **We appreciate the feedback and questions, which help clarify our NeRF‑based model for a broader conference audience. Below, we provide our explanation and responses to each point of the questions and concerns.**
>
> ### **Q1. Concept of downwelling distance**
>
> Thank you for the question that helps clarify the description of our model design. The downwelling distance $z^{\Phi}(r)$ is not explicitly a 3D function; instead, it is view-dependent and associated with each ray. Our model as well as previous studies [2-4] are all built upon the Revised Underwater Image Formation (RUIF) model [1] (Line 136/Sup Sec A.2), which plays an important role in forming the basis of scattering modeling. In precise underwater imaging, the line-of-sight (LoS) is not parallel in horizontal directions, and this introduces an inclination angle $\theta$ in the nadir direction. This causes the backscatter radiance at every point along the LoS to depend on $\theta$, which significantly increases the ill-posedness of the problem (Eq. 1 in RUIF paper). To make the problem tractable, RUIF makes a core contribution to the community by proposing a simplified formulation by setting $\theta=90^{\circ}$, which reduces to an inclination-free interpretation (Eq.2 in our Supp/Eq.2 in RUIF paper). **This simplification assumes all LoS are horizontal when computing the backscatter radiance** (also shown in Line 59 of our Supp). In this approximated model, each point along LoS will have the same downwelling depth. Note that this simplification only applies to backscatter radiance. **We will add this explanation in Sec 3.1 to emphasize horizon simplification, and revise the bottom-right portion of Fig. 1 to improve clarity.**.
>
> ### **Q2. Backscatter and media color & Points of downwelling depth and media color**
>
> Backscatter describes a behavior that particles redirect incident light into the camera’s viewing direction. Backscatter radiance is the cumulative radiance (multiplying transmittance) collected from those scattering events along a ray. Media color is $c^{med}(r)$, represents the scattering albedo (of a scattering particle) quantifying how much ambient light is redirected toward the camera in a single scattering assumption (Line 127, and Supp Sec. A.1/A.2). Under RUIF model, rays are approximated to be parallel when computing backscatter radiance, and all sampling points on a ray is simplified to have the same $c^{med}(r)$ and $z^{\Phi}(r)$.
>
> ### **Q3.1 Clarification of $c^{med}(r)$**
>
> In the scattering conditions such as underwater scenes, by adopting RUIF, the media color $c^{med}(r)$, scattering coefficient $\sigma^{scat}(r)$, and downwelling depth $z^{\Phi}(r)$ become constant along each ray. These coefficients are associated to each ray, and can vary across different rays. To accurately reflect this modeling, we revise Eq. 5 in our paper as follows:
>
> $c^{med}(r) = \Phi\ \cdot \exp\bigl(-(\sigma^{attn}(r) + \sigma^{scat}(r))z^{\phi}(r)\bigr)$
>
> Here $r$ represents individual ray and $\Phi$ is the scene constant global illumination.
>
> ### **Q3.2 Necessity of RSU strategy**
>
> Thank you for the comment. Under scattering conditions, where the medium color and scattering coefficients are constant, it is  possible to apply a closed-form solution to directly compute the backscatter radiance component of a ray. **We find that utilizing this closed-form approach can reduce GPU memory by approximately 4 GB (at a training batch size of 2048).** However, our spatial sampling design presents two advantages:
>
> First, inverse rendering in scattering conditions is an inherent ill-pose problem requires separating the direct and backscatter radiance given only the degraded observations. Explicit spatial sampling, such as provided by RSU, is therefore crucial for the successful optimization of the NeRF model. **Accurate media sampling helps prevent the collapse of the media field** (Line 177/292), a failure case where backscatter radiance is incorrectly attributed to direct object radiance, leaving the medium field empty (as demonstrated in Fig. 6 left and Supp. Fig. 1 w/o RSU). This collapse becomes particularly significant when the ambient light has close color to the scene color, such as in shallow underwater scene (Supp Fig. 1) and hazy scene (Supp Fig. 4). Geometrically accurate spatial sampling facilitates the proper separation and convergence of the two distinct fields (media and object), resulting in remarkably improved restoration quality, as demonstrated in Fig. 4, Fig. 6, and Supp Figs. 1, 3, and 4.
>
> Second, in low-light environments (no scattering only absorption), a spatially variant absorption field is required (Line 226) due to the highly-dynamical scene illumination. As a result, **the media density becomes inconstant along each ray, and there is no closed-form solution.** We visualize the density distribution of two representative rays in Fig. 3. To employ a unified model architecture capable of handling all adverse media conditions, we adhere to the current design of spatially sampling media points along each ray. When a constant value is needed, we apply additional pooling to retrieve a scalar value (Line 220). Moreover, as shown in Tab. 3 and right panel of Fig. 6, RSU also plays a crucial role under low-light conditions. Without this geometrically accurate sampling, the absorption field tends to spatially overlap with scene objects, resulting in visible artifacts in the restored outputs.
>
> ### **Q3.3 Explanation of Fig.6**
>
> Fig. 6 shows the distribution of density $\sigma^{scat}(r)$ for all rays from the entire image, plotted over sampling depth. As noted in Q3.1, $\sigma^{scat}(r)$ varies across rays, and thus the distribution is non-uniform.
>
> ### **Q4. Relation of media color and downwelling depth**
>
> As noted in Q3.1, the medium color $c^{med}(r)$ depends on the global constant $\Phi$, the attenuation and scattering coefficients $\sigma^{attn}(r)$ and $\sigma^{scat}(r)$, and the downwelling depth $z^{\Phi}(r)$. Since these coefficients are ray-dependent, different pixels/rays can have different values. Consequently, $c^{med}(r)$ does not necessarily decrease monotonically with $z^{\Phi}(r)$, as it is also dependent to $\sigma^{attn}(r)$ and $\sigma^{scat}(r)$. However, when these two coefficients are held constant, the backscatter radiance does decrease with increasing $z^{\Phi}(r)$, as demonstrated in Supp Fig. 8.
>
> ### **Relation to SeaThru-NeRF's formula on Section 3.3**
>
> Vanilla NeRF models only the radiance contributed by scene objects, neglecting the non-contributive clear air in the space. SeaThru-NeRF [2] was the first to attribute radiance and weights to the surrounding 3D volume, extending the formula of RUIF [1] to multi-view underwater reconstruction. Formula of Section 3.3 in our work builds upon this physical model, further generalizes to a broader range of media scenarios including low-light, haze, underwater, and even hybrid conditions (as shown in Supp Sec G.4) using a unified model formulation. $I^2$-NeRF incorporates physical plausibility and enables interpretable estimation of media properties. Specifically, our work differs from SeaThru-NeRF [2] in the following aspects:
>
> - We propose a unified physically-grounded formulation that models both low-light and scattering effects within a single framework (Lines 7/50/56/76). This allows our model to handle diverse media conditions, even including hybrid scenarios combining low-light and scattering showing in Supp Sec. G.4. In contrast, SeaThru-NeRF models only scattering.
> - Our model applies the Beer-Lambert attenuation law along the vertical direction, enabling physically plausible recovery of media properties, including water depth (Lines 13/62/81). By comparison, SeaThru-NeRF yields implausible estimations, as illustrated in Fig. 4.
> - We introduce RSU, a precise media sampling strategy that facilitates the separation of medium and object geometry (Lines 5/36/78) and prevents collapse of the media field, which otherwise leads to artifacts and vanishing media content (Lines 177/292/304). Without this sampling strategy, the model can converge to a trivial solution with low/zero media field, shown in Fig.6 and Supp Fig. 1.3.4.
>
> ### **Experiment performance**
>
> On the highly contested low-light restoration task [5], the proposed method archive SOTA performance compared to recent approaches. On the challenging SeaThru-NeRF dataset [2], it demonstrates competitive results in novel view synthesis. Beyond numerical performance, our approach provides physically plausible reconstructions, as discussed above, including sampling, restoration, and media property estimation. Furthermore, in Supp Sec.H, we presented two potential applications of $I^2$-NeRF for scientific estimation and visualization, highlighting its potential to extend physically grounded neural rendering to a broader range of real-world and scientific domains.
>
> We are willing to engage in further discussion.
>
> ### References:
>
> [1] Akkaynak, Derya, and Tali Treibitz. "A revised underwater image formation model." CVPR, 2018.
>
> [2] Levy, Deborah, et al. "Seathru-nerf: Neural radiance fields in scattering media." CVPR, 2023.
>
> [3] Tang, Yunkai, et al. "Neural underwater scene representation." CVPR, 2024.
>
> [4] Li, Huapeng, et al. "Watersplatting: Fast underwater 3d scene reconstruction using gaussian splatting." 3DV, 2025.
>
> [5] Cui, Ziteng, et al. "Aleth-nerf: Illumination adaptive nerf with concealing field assumption." AAAI, 2024.

---

> > ### Comment · Reviewer_sN9j · 2025-08-05
> > **Questions and Justification**
> >
> > Thank you for the authors' response—it helps address some of my concerns, though a few issues remain.
> >
> > * Physical Interpretation of View Dependency for $\sigma^{attn}$, $\sigma^{scat}$, and $z^{\Phi}$
> >
> > I still find it difficult to interpret these densities—and the corresponding downwelling depth—as being view-dependent. This formulation implies that a single 3D location could possess different densities (and downwelling depths) when observed from various directions. What would be the physical interpretation for this effect? I note that a similar view dependency of back-scatter densities appears in both SeaThru-NeRF and WaterSplatting, yet neither work provides convincing physical justification. Particularly,  $z^{\Phi}$  is proposed by this paper, if this is view-dependent and not 3D-attached, does it means the water depth at vanishing line can vary across views? What is the consistency of this depth across views?
> >
> > *  RSU Strategy
> >
> > I agree that the RSU strategy could be valuable in low-light environments where media density varies spatially. However, Figure 3 appears to show limited spatial variance in media density. I remain unconvinced that RSU is crucial for distinguishing object and media in underwater scenarios. If the media density is constant and object density is zero in the media region, increasing the sample count along these ray segments does not seem to meaningfully affect the gradients used for NeRF supervision. i.e. suppose you have two samples along the ray, with $\sigma_{scat}=\sigma$ and $\sigma_{obj}=0$, sampled at $t_1$ and $t_2$. Accumulating the two sample points is exatly the same as directly having one sample point on the sencond position:
> >
> > $ \hat{C}_1^{med} + \hat{C}_2^{med}  = (1-e^{-\sigma t_1}) c_i^{med} + e^{-\sigma t_1} (1-e^{-\sigma (t_2 - t_1)})c_i^{med}  = (1-e^{-\sigma t_2})c_i^{med}  $
> >
> > . I suppose it is only critical to sample a point where $\sigma_{obj}=0$ and being close enough to the ojbect. Moreover, it seems also that simple additional uniform sampling along the ray might serve just as well, without the added complexity of reverse-stratified sampling.
> >
> > Other Clarifications Needed:
> >
> > * Downwelling Distance:
> >
> > Since $z^{\Phi}(r)$ now depends solely on view direction, it would be helpful to update Figure 4c to avoid any misinterpretation about its dependence on the 3D position.
> >
> > Overall, after considering the rebuttal, I am reaching a borderline—with concerns leaning negative. I find the paper’s two key contributions—(1) explicit modeling of media color via downwelling distance, and (2) reverse-stratified sampling for the media region—somewhat trivial, and key issues remains. And the performance is not significant in quantity metric. Nevertheless, its extension to low-light application seems valid.

---

> ### Author Response · Authors · 2025-08-06
>
> We sincerely appreciate the thoughtful follow-up questions. This external scrutiny is essential for identifying potential blind spots in the foundation of our work.
>
> ### **Concern over comparable performance on quantity metric**
>
> Our quantity metric on haze/underwater is not significant due to inherent limitations of the benchmark setting. We included them in Tab. 1 only to enable a fair comparison with baseline approaches [2-4]. The scattering dataset [2] does not provide ground-truth restoration results, i.e., paired clean images with water/haze removed. So we can only evaluate our method indirectly through novel-view synthesis on degraded views. Moreover, our qualitative evaluations in Fig. 4 and Supp Figs. 1, 3, and 4 show superior geometry reconstruction and scene restoration compared to baseline approaches. However, again, due to the lack of ground-truth images, we can only illustrate these improvements qualitatively.
>
> **More importantly**, our explicit media sampling and unified modeling of light transport endows unique abilities that no previous research like SeaThru-NeRF [2] and Watersplatting [3] has touched yet. Our method enables estimation of water depth (Fig. 4 and Supp. Fig. 3), rendering of physically plausible media color (Fig. 4 and Supp. Fig. 4), synthesis of novel views across varying water depths (Supp. Sec. H and Supp. Fig. 8), and effective modeling of hybrid media scenarios (Supp. Sec. G.4 and Supp. Fig. 6).
>
> Regarding low-light illumination conditions, we are proud to achieve SOTA performance. Thanks to the availability of paired ground-truth data, this highly competitive task has received wide attention across numerous top-tier conferences. We are pleased that our spatially-varying media field provides a unique advantage in this setting.
>
> ### **Q1: Physical Interpretation of View Dependency**
>
> Our attenuation and scattering coefficients $\sigma^{attn}$, $\sigma^{scat}$, media color $c^{\text{med}}$, and downwelling depth $z^{\Phi}$ are both view-dependent and spatially dependent, just like how vanilla NeRF treats point color  $c^{\text{obj}}$ on the object. Vanilla NeRF and our method achieve this by using both spatial encoding and view direction encoding.
>
> On the contrary, previous studies like SeaThru-NeRF [2] and Watersplatting  [3]  **indeed**  could possess different densities for a single 3D location when observed from various directions. This is because they only encode the viewing direction when predicting media properties, without incorporating spatial information. This purely view-dependent modeling can lead to a trivial solution where the media radiance collapses to zero, as we discussed in the rebuttal (Fig. 6 left, Supp. Figs. 1, 3, and 4).
>
> In our work, as shown in Fig. 2c, we incorporate both explicit spatial encoding and view direction encoding to predict media properties,  just like the object color in vanilla NeRF, which preserves the spatial consistency from multiviews.
>
> ### **Q2. RSU Sampling Strategy**
>
> Thank you for the question. In underwater scattering conditions, we illustrate the intuition behind our approach: using multiple samples injects multiple non-zero gradient pathways at distinct positions, whereas a single-sample strategy collapses all learning into a single scalar gradient, which often vanishes in low-density regions.
>
>
> Given $\sigma$ and $c^{med}$ are constant along a ray. Consider the sampling interval $[0, Z]$, and only consider the media contributed radiance. We comparing the gradient between using a single sampling point and multiple sampling points, as shown below:
>
> With a single sample over interval $[0, Z]$,
>
> $$
> \hat C_{\rm med} = \bigl(1 - e^{-σZ}\bigr) c\~,
> \qquad
> \frac{\partial \hat C_{\rm med}}{\partial σ}
> = e^{-σZ}Zc\~,
> $$
>
> this single scalar gradient can vanish when $e^{-σZ}\approx0$ (e.g., in deep or highly absorbing regions) or when its weight $(1 - e^{-σZ})$ is near zero (shallow, low-density regions).
>
> When using multiple sampling points, with $N$ samples at place $t_i$ and segment lengths $\Delta t_i$,
>
> $$
> \hat C\_{\rm med}
> = \sum\_{i=1}^N T(t\_i)\bigl(1 - e^{-\sigma\Delta t\_i}\bigr)c\~,
> \quad
> T(t\_i)=\exp\Bigl(-\sigma\sum\_{j<i}\Delta t\_j\Bigr),
> $$
>
> $$
> \frac{\partial \hat C\_{\rm med}}{\partial \sigma}
> =c\sum\_{i=1}^N \Bigl[
> \underbrace{\frac{\partial T(t\_i)}{\partial \sigma}\bigl(1-e^{-\sigma\Delta t\_i}\bigr)}\_{\text{gradient via transmittance}}
> +\underbrace{T(t\_i)\Delta t\_ie^{-\sigma\Delta t\_i}}\_{\text{gradient via local attenuation}}\Bigr].
> $$
>
> **Each term contributes a non-zero gradient at its own depth, and even if some weights vanish, others can still remain.** So the network consistently receives learning signals across the medium and can reliably recover the constant $σ$ and disentangle medium from object contributions.

---

> > ### Comment · Reviewer_sN9j · 2025-08-06
> >
> > Thanks for the response.
> >
> > For Q1: I feel the reply didn’t quite address my main concern. Specifically, why do the  $\sigma^{attn}$, $\sigma^{scat}$, and $z^{\Phi}$ need to be View-dependent at all? To me, it seems natural for these terms to be **only** spatially dependent. For instance, in vanilla NeRF, density is spatially dependent only. In the context of underwater scenes, the initial rebuttal material implied that these properties are view-dependent under the constant assumption, especially "The downwelling distance
> >  is not explicitly a 3D function; instead, it is view-dependent and associated with each ray."
> >
> > For Q2: I appreciate this deeper analysis on the gradient. However, if we go deeper in the equation, most of the terms in the summation cancels out exatly and it just goes back to the single point sampling: ( using your notation with $N$ samples at place $t_i$）
> >
> > $\frac{\partial \hat{C}_{med}}{\partial \sigma} =$
> >
> > $$
> >  c \sum_i^N{( -t_{i-1} e^{-t_{i-1}\sigma} + t_i e^{-t_i\sigma} )} = c ( 1- t_N e^{-t_N\sigma})
> > $$
> >
> > Therefore I am regarding this is more like a numerical issue if it indeed vanishes somehow, but the actual signal for gradient descent should remains the same regardless of the number of points. Also, within the nomalized range of $t_N \in [0,1]$, and $\sigma$ typically around 1-2 (e.g. in Fig 3). It doesn't seem to be likely to goes cracy towards $e^{-t_N\sigma} \approx 0$. And if $\sigma$ is too small to cause $1-e^{-t_N\sigma} \approx 0$, it would lead to $\hat{C}_{med} \approx 0$ anyways regardless of the number of samples.

---

> ### Author Response · Authors · 2025-08-07
>
> We sincerely appreciate your timely feedback and the detailed questions.
>
> ### **Q1. Scattering**
>
> Thank you for the question. And yes, all these media properties are encoded using both viewing direction and spatial encoding, as $f\_\theta(\mathbf{x}, \phi)$. $\theta$ here represents the NeRF network.
>
> The essence of scattering in a medium is a view-dependent process. Light propagates through water, it interacts with small particles, causing scattering with directional bias. The intensity and color of this scattering not only depend on the spatial position, but also relates to which direction the light comes from and which direction it goes to. This is similar to how BRDF describes surface reflection:
>
> - BRDF $f(\phi_{in},\phi_{out})$ tells us that the incoming direction $\phi_{in}$​ and the outgoing (viewing) direction $\phi_{out}$ together determine the intensity and color of reflected light on a surface.
>
> - The phase function is also a bivariate function, similarly considering the relationship between incoming and outgoing directions.
>
> In scattering models, previous work usually assumes: (1) fixed light source direction (e.g., sunlight, or the probe always captures light from a fixed direction), and (2) constant medium properties at coarser scale/per ray. Therefore, medium properties (or phase function parameters) are modeled using the viewing direction. The network learns a mapping from direction to medium response.
>
> Our method retains direction dependence, while also learning spatial position $\mathbf{x}$ (i.e., learning $f_{\theta}(\mathbf{x}, \phi))$. This enables capturing: (1) which location causes differences in appearance and scattering characteristics, and (2) which direction causes variations in color and intensity. In this way, while preserving the BRDF-like directional dependence, we can also realistically model the spatial variation of the medium. Moreover, our model can also support inconstant media distribution per ray, allowing full spatial variance.
>
> ### **Q2. Gradient**
>
> We really appreciate your derivation of the gradient. We agree with you that one can strictly telescope-sum the partial derivative with respect to the constant $\sigma$ per ray as:
>
> $$
> \hat C^{med}
> = c \sum_{i=1}^N T(t_i)\bigl(1 - e^{-\sigma\Delta t_i}\bigr)
> = c \sum_{i=1}^N\bigl(e^{-\sigma t_{i-1}} - e^{-\sigma t_i}\bigr)
> = c\bigl(1 - e^{-\sigma Z}\bigr),
> $$
>
> therefore, the gradient of $\sigma$ with respect to $\hat{C}^{med}$ is
>
> $$
> \frac{\partial\hat C^{med}}{\partial \sigma}
> = cZe^{-\sigma Z}.
> $$
>
> But we found that the acutally gradient term should be $\frac{\partial \hat C^{med}}{\partial \theta}$, where $\theta$ is the NeRF netowrk. So the gradient presented in our previous response should be expanded by the chain rule, and we apologize for this oversight.
>
> In network training, we truly optimize the parameters $\theta$, and each sample point $t_i$ produces an independent gradient path:
>
> $$
> \frac{\partial \hat C^{med}}{\partial \theta}
> = \sum_{i=1}^N \frac{\partial \hat C^{med}}{\partial \sigma_i}\frac{\partial \sigma_i}{\partial \theta}
> = \sum_{i=1}^N w_i\frac{\partial \sigma_i}{\partial \theta},
> $$
>
> where
>
> $$
> w_i
> = \frac{\partial \hat C^{med}}{\partial \sigma_i}
> = c\Bigl[-\Bigl(\sum_{j<i}\Delta t_j\Bigr)T(t_i)\bigl(1 - e^{-\sigma\Delta t_i}\bigr) + T(t_i)\Delta t_ie^{-\sigma\Delta t_i}
> \Bigr].
> $$
>
> We can telescope-sum $\sum_{i=1}^N w_i$ to obtain $cZe^{-\sigma Z}$; but here each term $w_i$ is multiplied by its own $\partial \sigma_i / \partial \theta$, forming $N$ parallel paths. This step cannot be merged into a single-point route. Under the chain rule, network parameter gradients accumulate via multiple paths.
>
> On the other hand, we realize that during the rendering process, we can indeed follow your suggestion to use the above simplification to interpret backscatter radiance. Specifically, for the object term $\hat{C}^{obj}$, which depends on the per-sample transmittance $T_i^{D}$  involving the attenuation coefficient $\sigma^{attn}$ (Eq. 6), we can factor and compute $T_i^{D}$ separately as:
>
> $$
> T_i^D =
> \underbrace{\exp\Bigl(-\sigma^{attn}z_i\Bigr)}\_{T_i^{med}}
> \times
> \underbrace{\exp\Bigl(-\sum_{j=1}^{i-1}\sigma_j^{obj}\delta_j\Bigr)}\_{T_i^{obj}}.
> $$
>
> For the media term $\hat{C}^{med}$ that involves $T_i^{B}$, we can apply the similar factorization.
>
> This can slightly improve the rendering efficiency specific to scattering scenarios, and we really appreciate for raising this point.

---

> ### Author Response · Authors · 2025-08-07
>
> Sorry for the additional response and we really appreciate your time to discuss with us. We also found it helpful to clarify the fundamental objectives here, as you have raised an important question from the boarder community. We would like to elaborate a bit more on why our method, as well as previous approaches, introduces view dependency.
>
> The “density” of a participating medium is not its mass density but rather its optical density. That is, the coefficient that determines how the medium scatters and absorbs light. For an intuitive example, imagine a clear plastic bottle of water: although the water’s physical mass density stays the same, its apparent opacity and hue change dramatically when viewing it under direct sunlight versus viewing it on the table. In vanilla NeRF, all these complex light-matter interactions are implicitly aggregated into the object color term $c^{\mathrm{obj}}$. This works well for opaque objects with negligible scattering or reflection, where the scene remains consistent in 3D space. However, in strongly scattering or reflective environments, as shown by Ref-NeRF (CVPR 22 Best paper candidate), ScatterNeRF (ICCV 23), and SeaThru-NeRF (CVPR 23), vanilla NeRF fails to maintain multi-view consistency.
>
> By incorporating our method, we explicitly disentangle view-dependent components (such as scattering and absorption effects) from the underlying scene geometry. This ensures that the object geometry remains stable across viewpoints, while the variable light transport is modeled separately. We would like to remind that the ultimate objective of 3D reconstruction is to recover accurate scene geometry, and our approach, along with previous efforts, achieves this by isolating view-dependent light effects from the consistent spatial structure. In Fig. 4 and Supp Fig 3–5, we clearly demonstrate how object geometry is faithfully recovered in novel views. We should not revert to vanilla NeRF’s limitations and assuming everything to be solely spatial consistent, as in real-world situations, rays are inherently view-dependent and lighting conditions vary across views, so a robust model must separate these transient effects from the concrete, multi-view consistent geometry. **So this view-dependency for media in this line of research is not the limitation, but our efforts to ensure the geometry restoration from challenging lighting conditions.**

---

> ### Comment · Reviewer_sN9j · 2025-08-07
>
> Thank you for your response.
>
> **The Physical Interpretation of Direction Dependency of $\sigma^{attn}$ and $\sigma^{scat}$:**
>
> It is informative to discuss these key concepts. However, I don't think interpolating the view dependency of $\sigma^{attn}$ and $\sigma^{scat}$ using the phase function is valid. These parameters are physical properties of the medium itself and are independent of the phase function. The phase function only affects the rendered color, which explains why the color in vanilla NeRF is spatially and view-dependent. That said, I found supporting material in a textbook [1] stating: “In general, the absorption cross section (densities) may vary with both position and direction, although it is **normally just a function of position**.” It makes sense to me to interpret the direction-dependence as a result of **anisotropy in the medium's mass density**, as demonstrated in the smoke rendering example in the textbook [1]. Additionally, I came across another paper discussing anisotropic modeling [2], which seems to be a useful reference. I also want to point out that your cited refs, Ref-NeRF (CVPR 2022 Best Paper candidate) and ScatterNeRF (ICCV 2023), maintain the density modeling as dependent only on position.
>
> Now that I am convinced it is physically plausible for the densities to be view-dependent due to anisotropy of the medium, I am not entirely sure that this applies to seawater, which appears to be quite isotropic. Related literature or experimental validation would be helpful. Also, I am not fully clear on the example of the bottle of water and what you intend to illustrate with it. But I think this concern is partially addressed, as view-dependence is a paradigm in underwater domains like SeaThru-NeRF and WaterSplatting, though further evidence is needed.
>
>
> **The Physical Interpretation of Direction Dependency of  $z^{\Phi}(r)$:**
>
> Regarding the physical interpretation of the proposed "downwelling distance" $z^{\Phi}(r)$, it remains unclear to me. Since you relate it to physical distance to demonstrate the method’s "Isometry," it is confusing why it would be **view-dependent only** (which is suggested in the original rebuttal material).
>
> **Gradient of Sampling**
>
> Breaking it down further is helpful. I believe the key question is whether $\frac{\partial \sigma_i}{\partial \theta}$ remains constant along the ray. In your simplification of the underwater scene, you enforce that $\sigma^{attn}$ and $\sigma^{scat}$ are constant along each ray. I assume that these densities are modeled as $\sigma_{\theta}(\phi)$, where $\phi$ represents the ray direction. Therefore, for any given ray, $\sigma$ is queried with the same parameters and direction, resulting in identical values. Under this assumption, I would expect $\frac{\partial \sigma_i}{\partial \theta}$ to be the same across sampled points along that ray.
>
> **References:**
>
> [1] https://pbr-book.org/3ed-2018/Volume_Scattering/Volume_Scattering_Processes
> [2] Anisotropic Neural Representation Learning for High-Quality Neural Rendering

---

> ### Author Response · Authors · 2025-08-07
>
> Thank you for your comments and your input is really important to improve our work. Due to word limits, we break our reply into two parts. We also sincerely appreciate your patience to discuss with us.
>
> ### **Q1. Physical Interpretation of Direction Dependency**
>
> Thank you for pointing out that ScatterNeRF (ICCV 2023) employs only spatial encoding for the haze medium; we are sorry having overlooked this detail in our previous description. This provides helpful insight when comparing with other scattering modeling approaches. We also greatly appreciate the detailed references you provided, which have broadened our literature review and understanding, and we will be sure to include them into our paper.
>
> We also thank you for the question regarding the view-dependence in sea water, which helps to consolidate our work. There are authoritative materials in ocean optics showing that natural waters can exhibit either isotropic or anisotropic scattering (or even both), depending on their constituents. For example, **Petzold et al. (1972)** measured volume scattering functions (VSFs) across a range of open-ocean and coastal waters, and showed a transition from nearly flat (isotropic) in clear water to sharply forward-peaked (anisotropic) in turbid/coastal water. **Jerlov et al. (1968)** (also cited as [34] in our paper) classified water types by inherent optical properties and found that molecular (Rayleigh) scattering in the clearest waters (Type I) is near isotropic, and particulate  (Mei) scattering in greener waters dominates and becomes highly anisotropic. **Mobley et al. (1994)** (also cited as [56] in our paper) reported similar results. Based on these references, we hope using both viewing and spatial encoding in our method sounds reasonable to you and address your concerns about viewing-dependency in scattering.
>
> Moreover, we would like to elaborate a little bit more on the scattering coefficient and phase-function. In scattering media, the view-dependence of scattering coefficient $\sigma$ actually arises from phase-function. As explained in the helpful reference you provided [1]:*“phase functions actually define probability distributions for scattering in a particular direction,”*, *"the in-scattering portion of the source term is the product of the scattering probability per unit distance $\sigma$"*, and *“the amount of added radiance at a point is given by the spherical integral of the product of incident radiance and the phase function”.* In contrast, mass density is a scalar quantity that has no direction in physics. Any directional component therefore comes from the phase function. So Ref-NeRF becomes a helpful reference that presents one possible solution to tackle the view-dependence in phase-function under reflection scenario using BRDF.
>
> ### References
>
> [1] Volume Scattering Processes
>
> [2] T. J. Petzold, Volume Scattering Functions for Selected Ocean Waters, 1972.
>
> [3] N. G. Jerlov, Irradiance Optical Classification, 1968.
>
> [4] C. D. Mobley, Light and Water: Radiative Transfer in Natural Waters, 1994.

---

> ### Author Response · Authors · 2025-08-07
>
> ### **Q2. Downwelling distance $z^{\Phi}$**
>
> Thank you for the question. In our vertical attenuation model, we interpret $c^{med}$ as the radiance from ambient light, i.e., sunlight, traveling through the water surface before intersecting the viewer’s line of sight (LoS). Although real-world scattering involves multiple interactions and the true $c^{med}$ at each point is the sum of many contributions, the dominant component is downwelling irradiance from the surface (Chapter 3 Optical Properties of Water by Mobley et. al., *“About 90% of the diffusely reflected light from a water body comes from a surface layer of water of depth 1/Kd; thus Kd has implications for remote sensing”*, Kd is defined in Supp Eq. 29). Our model therefore explicitly relates depth to $c^{med}$ using the same radiative transfer formulation, which enables the estimation of the vertical traveling distance of light. Because scattering depends on viewing direction, any depth estimate based on scattering intensity is inherently view dependent. By adopting a parallel-ray approximation for backscatter, we interpret “downwelling depth” simply as the vertical distance from that horizontal ray to the water surface. While these simplifications introduce approximation error, they permit plausible depth recovery from only RGB images without additional sensor data. Building on this foundation, explicitly incorporating the viewing angle $\theta$ into our model becomes a promising direction for future work to improve both backscatter and depth estimation accuracy (although they will still be view/ray-depdent). If you have suggestions for additional cues or properties that could alternatively lead to more accurate depth estimates, we would be very grateful to hear and discuss to improve our work.
>
> **In short:** $E_d(z, \lambda) = E_d(0^{+}, \lambda) e^{-K_d(\lambda) z}$ (Supp Eq. 29) is a well-established formula in literature (Eq. 3.19 and Eq. 3.21, Mobley et al.). $E_d(z,\lambda)$ is the irradiance, i.e., *a hemispheric integral of radiance with a cosine weight over the downward directions*. It is a scalar value with no direction. But we have no way to estimate this value without acutual sensing data. So we can only approximate its magnitude via the in-scatter component $c^{med}$ (IOCCG Report 5, Eq. 1.11–1.12), and this in-scatter term is view-dependent.
>
> **Minors:** We would like to clarify that we have not stated the downwelling depth $z^{\Phi}$ is view-dependent **only**. For clarity, we provide our previous explanations below, and **we apologize if they might cause any confusion**. The most important thing to us is to address your confusion.
>
> - *The downwelling distance is not explicitly a 3D function; instead, it is view-dependent and associated with each ray*
>
> - *These coefficients are associated to each ray, and can vary across different rays.*
>
> - *In our work, as shown in Fig. 2c, we incorporate both explicit spatial encoding and view direction encoding to predict media properties.*
>
> - *Our attenuation and scattering coefficients $\sigma^{attn}$, $\sigma^{scat}$, media color $c^{\text{med}}$, and downwelling depth $z^{\Phi}$ are both view-dependent and spatially dependent, just like how vanilla NeRF treats point color  $c^{\text{obj}}$ on the object.*
>
> - *all these media properties are encoded using both viewing direction and spatial encoding, as $f\_\theta(\mathbf{x}, \phi)$.*
>
> Associating with ray $r$ does not mean they become view-dependent only. Ray spatially exists in 3D space, and encoded with viewing direction. Similarly, the object color in vanilla NeRF is also view-dependent, but not view-dependent only.
>
> ### **Q3. Gradient**
>
> Thank you for the detailed analysis; it revealed a blind spot in our presentation. We sincerely appreciate this in-depth discussion that helps to clarify our work.
>
> As we noted in previous replies, "media properties are encoded using both viewing direction and spatial position via $f\_\theta(\mathbf{x}, \phi)$”. In our implementation, low-light environments, which require a spatially varying media field, use the per-sample outputs $\sigma$ in subsequent volume rendering. In scattering conditions, where the medium should be constant along the ray (make the ill-posed problem tractable), we simply pool the results to obtain constant $\sigma$ values for each ray to ensure a unified model architecture and enable encoding both spatial and viewing information. The constant $\sigma$s are then participating in volume rendering. Although $\sigma$ is constant in volume rendering, its gradient with respect to $\theta$ decomposes into N parallel paths, since each $\partial\sigma_i/\partial\theta$ in $f_{\theta}(\mathbf{x}_i,\phi)$ remains distinct. We will include this explanation in the paper to clarify this back-propagation behavior.
>
> ### Reference
>
> [1] Mobley, “Optical Properties of Water"
>
> [2] IOCCG Report 5, "Remote Sensing of Inherent Optical Properties: Fundamentals, Tests of Algorithms, and Applications"

---

> ### Comment · Reviewer_sN9j · 2025-08-08
>
> Thanks for your response.
>
> **Regarding the Gradient Issue**
>
> I find this explanation reasonable. The explicit dependence on spatial position followed by the pooling operation does indeed alter the gradient flow. It now makes sense that this would impact the training via the proposed RSU. I will adjust my score accordingly if this analysis is clearly presented and elaborated in the revised paper.
>
> **Regarding View Dependency of Densities**
>
> I respectfully disagree with the assertion that the view dependency of density arises solely from the phase function. It should be noted that when discussing "color" in NeRF rendering, we are actually referring to the "emitted (exitant) radiance," not the "incident radiance." Therefore, the "color" itself already results from the integration of the phase function (or BRDF in the case of reflection) with the incident radiance. This is precisely the case in your referenced Ref-NeRF (Or I should say, Ref-NeRF indeed modeled the incident radiance at reflected direction, but all view-dependent BRDF integration ramains within the emitted radiance, not on the density). That said, I think it is okay to follow the paradigm established by previous works like SeaThru-NeRF and WaterSplatting. I will review the related material you provided, but at a glance, they seems to disscuss the phase fuction.
>
> **Regarding View Dependency of $Z^{\Phi}$**
>
> I appreciate your connection of the backscatter color $c_i^{med}$ with downwelling distance $Z^{\Phi}$. However, I feel the view dependency is not thoroughly justified. From my perspective, it makes more sense to define $Z^{\Phi}(\mathbf{x})$ as purely dependent on position, and then incorporate an additional view-dependent encoding to derive $c_i^{med}(\mathbf{x}, r)$. As you mention in Appendix C: “**the ambient light at infinity** is **proportional to the irradiance**, which models the downwelling light reaching the medium along vertical paths.” I note that irradiance is a direction-independent measure of radiant flux per unit area, whereas $c_i^{med}$ represents **radiance**, which is direction-dependent, after integration with the phase function. This gives the freedom that $c_i^{med}$ can be view dependent while $Z^{\Phi}$ is not.
>
> Otherwise, if $Z^{\Phi}$ is view-dependent and not anchored in 3D space, does that imply that the water depth at the vanishing line varies across views? How is this depth consistency maintained across different viewpoints?

---

> ### Author Response · Authors · 2025-08-08
>
> Thank you very much for your reply. We greatly appreciate your invaluable input and your patience throughout this discussion. **As the discussion period is near its end, we would like to summarize the key points and confirm whether your concerns have been sufficiently addressed.** We fully respect your perspective and understand that different opinions exist. Nonetheless, this exchange has truly helped us re-examine our project and identify important directions for future research.
>
> ### **Q1. RSU Gradient**
>
> We appreciate your feedback. We will clearly incorporate the derivation from our discussion into the paper so that our claim is explicitly supported by this reference. **We believe the concern regarding RSU is now sufficiently addressed**, and we sincerely thank you for the thorough and insightful discussion.
>
> ### **Q2. View-dependency**
>
> We are grateful for the discussion regarding Ref-NeRF and the phase function. If it is reasonable to follow previous studies in treating scattering as view-dependent, **then we consider the concerns about the plausibility of view-dependence to be addressed.** We will still carefully review the entire textbook, as it is not possible to fully digest its contents in such a short period of time.
>
> ### **Q3. Depth $z^{\Phi}$**
>
> Our method encoded both viewing and spatial information for $z^{\Phi}$, so its prediction has connection to the viewing components. We sincerely thank you for the suggestion that making $z^{\Phi}$ purely relies on position $\mathbf{x}$, and learning a separate phase function $\Omega(\lambda, \theta(r))$ such as:
>
> $$
> c^{med}(\mathbf{x}, \mathbf{r}, \lambda) =
> \Phi(\lambda)e^{-K\_dz^{\Phi}(\mathbf{x})}
> \underbrace{\Omega(\lambda, \theta(\mathbf{r}))}_{\text{view dependent}},
> $$
>
> For a short test, we chose a native isotropic phase-function $\Omega(\lambda, \theta)=1$ and found it not providing on-par scattering estimation as our current solution. A likely reason is $c^{med}$ needs more view dependence that an naive phase-function cannot provide.
>
> From this discussion, we truly believe your suggestion opens a promising research direction and brings valuable insights to this field. Future work could explore explicit integration of phase-functions, such as low-order angular models or higher-order Henyey-Greenstein functions, or implicitly learned phase-functions via neural networks. Building upon our work, future studies can explore a purely 3D-anchored depth estimation. We will include these content as the future work in our paper.

---

### Official Review · Reviewer_sErE · 2025-07-03

**Clarity:** 3
**Significance:** 3
**Originality:** 3
**Rating:** 5
**Confidence:** 3

**Summary:**

This paper presents I2-NeRF, a neural radiance field framework designed to improve 3D perception in challenging environments with media degradation such as underwater, hazy, or low-light conditions. The method introduces a reverse-stratified upsampling strategy for near-uniform 3D sampling to preserve geometric properties (isometry) and develops a unified radiative formulation based on the Beer-Lambert law that handles emission, absorption, and scattering effects in degraded media.

**Questions:**

N/A

**Ethical Concerns:**

["NO or VERY MINOR ethics concerns only"]

**Final Justification:**

My initail concerns, especially regarding the theoretical foundation of RSU in low-light conditions and the view-dependent density modeling, have been adequately addressed through the authors' detailed responses. For this I continue to recommend acceptance. The authors provided clear explanations that:

- RSU in low-light settings is theoretically sound - the method models absorption (not scattering) in low-light conditions, where the "density" represents optical attenuation coefficients rather than physical density. The separation of absorption fields from object geometry allows proper restoration while preserving underlying scene structure.
- View-dependent modeling is scientifically justified - the authors provided authoritative references, with detailed explanation, convincing me that that scattering phase functions are inherently view-dependent. Their statement that NeRF tries to recover *optical density* of the scene, not explicit geometrical density, was especially crucial in coming to this understanding. I agree with this statement, as NeRF ultimately is trained on photometric loss between the representation and its rendered image.

The rebuttal successfully resolved my primary theoretical concerns about the method's foundation. The authors demonstrated good understanding of light transport theory and provided appropriate scientific backing for their design choices. While some complexity remains in the view-dependent architecture, this appears to be a necessary component for handling real-world scattering scenarios rather than a fundamental flaw. In this light, I recommend acceptance for this paper.

**Limitations:**

Yes.

**Quality:**

4

**Strengths And Weaknesses:**

**Strengths:**
- The paper is well-written, organized and easy to follow.
- The paper takes into account absorption as well as scattering in volumetric rendering, allowing for improved performance in low-light condition or condition with semi-transparent medium.
- Proposal of reverse stratified sampling (RSU) for modeling on semi-transparent medium is novel and intuitive, and is demonstrated to be effective in the ablation experiment.
- Overall, the method is robust and clear in its objective, and the method seems well-designed and meets the objective well.

**Weaknesses:**
- The paper's main focus is on describing novel sampling strategy and volumetric rendering strategy in case of non-translucent medium, but the setting where the method achieves SoTA is low-light setting, in which external supervision signal $T_p$ plays a heavy role. In other settings, the method is unable to achieve complete SoTA. This makes me question whether the performance improvement of the method derives from incorporation of $T_p$ rather than its novel reverse stratified sampling or medium-modeling volumetric rendering. I ask the authors to clarify this question for me, demonstrating that its major performance gains come from the novel methodology the method proposes.
- The method is designed to implicit neural representation-based settings where sampling of the network is conducted: can this method be extended to 3DGS for improved speed and rendering speed? What is the computation overhead and optimization time required to train a single scene? Please elaborate.

---

> ### Author Rebuttal · Authors · 2025-07-26
>
> **We are grateful for the valuable comments, especially the suggestion to extend the proposed method to 3DGS. Please find our detailed responses below. We look forward to hearing your opinion.** (Supp denotes supplementary matrial)
>
> ### **W1. Ablation studies of $T_P$ and RSU**
>
> Thank you for the question. As shown in Sec. 5.4 ablation study, we demonstrate that **while $T_P$ plays an important role in low-light conditions, the physical constraints and RSU sampling are also essential for accurate modeling**. Our ablation study in Tab. 3 shows that with $L_{trans}$ that incorporates external supervision signal $T_P$, the performance of our model on LOM dataset [1] increases for 1 dB in PSNR. RSU also proves crucial, and it increases PSNR for approximately 2 dB.
>
> In low-light environments, the proposed method integrates several key elements to enhance restoration performance. Specifically, we incorporate an additional supervision signal $T_P$ derived from the Bright Channel Prior (Supp Sec A.4), which is used to define the transmittance loss. In addition, the model includes geometry-guided depth supervision (Line 193), a specially designed SSIM loss to compensate for luminance and contrast discrepancies (Line 196), and the RSU sampling strategy (Sec 3.4). By employing all of these components, the proposed method achieves SOTA performance on the competitive low-light restoration task [1].
>
> The importance of RSU is further illustrated in the right panel of Fig. 6. Without explicit media sampling, the estimated media field tends to spatially overlap with the object, as shown in the sub-figures of density distribution. This leads to noticeable artifacts in the restored novel views.
>
> On the other hand, in real-world scattering environments, it is inherently difficult to remove the scattering medium from a scene to obtain clean ground-truth restoration views. As a result, datasets like SeaThru-NeRF [2] lack ground-truth images for evaluating restoration performance. Therefore, in this setting, we report novel view synthesis (NVS) results instead. As shown in Tab.2, $I^2$-NeRF achieves competitive performance in NVS under scattering conditions.
>
> **In scattering conditions, by incorporating our geometric constraints and RSU sampling strategy, we achieve notable improvements in both restoration quality and geometric accuracy compared to their counterparts without these components.** In scattering scenarios, we do not incorporate $L_{trans}$ as there is not external supervision signal on transmittance, i.e., $T_P$. To qualitatively evaluate the effect of RSU on restoration performance, we illustrate the restoration outcomes with and without  RSU in the left panel of Fig. 6. In scattering conditions, insufficient media sampling along the ray causes the backscatter radiance to collapse into the direct radiance emitted by scene objects. This collapse disrupts the separation between the media field and object geometry, leading to empty media field and visible artifacts in restored views.
>
> In Supp Sec F.1, we provide more visualizations (Fig. 1) and a quantitative chromatic analysis for each key designs of $I^2$-NeRF under scattering conditions.
>
> ### **W2.1 Training time and computation overhead**
>
> The average training time of our model is 1.23 hours on the LOM dataset [1] (compare to Aleth-NeRF [1] of 8.25 hours) and 0.86 hours on the SeaThru-NeRF dataset [2] (compare to SeaThru-NeRF [2] of 3.15 hours). $I^2$-NeRF achieves substantially faster training speeds compared to previous NeRF-based approaches [1-2], benefiting from the hash encoding of spatially sampled features. Regarding computational overhead, compared to ZipNeRF [5] (our codebase), $I^2$-NeRF requires approximately 30% more training time (\~20 minutes) on the low-light dataset [1], and approximately 25% more (\~12 minutes) on the underwater dataset [2] on same device.
>
> In Supp Sec. K, we provide detailed information for the training time of the proposed method and baselines on both low-light and underwater scenes using the same GPU. **We will add a sentence in Line 246 to indicate the training time and refer readers to the corresponding section in supplementary material for more details.**
>
> ### **W2.2 Potential extension to 3DGS**
>
> We greatly appreciate this brilliant idea. After investigation, we find that a potential solution lies in integrating rasterization-based rendering techniques with point/voxel-based representations derived from NeRF, such as the recently proposed Sparse Voxel Rasterization [6]. This hybrid approach would enable geometry-aware sampling for both objects and participating media, while leveraging the rendering efficiency of rasterization. By doing so, it has the potential to significantly accelerate training without compromising reconstruction quality, offering the best of both implicit and explicit representations.
>
> We are particularly excited about the implications of this direction for real-time and resource-constrained applications. Achieving such efficiency would pave the way for deploying the proposed method on embedded systems, edge devices, and long-term environmental monitoring platforms, where both computational speed and physical realism are critical. We look forward to exploring this promising avenue in future work and believe it could significantly broaden the impact of physically-grounded neural rendering.
>
> ### References:
>
> [1] Cui, Ziteng, et al. "Aleth-nerf: Illumination adaptive nerf with concealing field assumption." Proceedings of the AAAI conference on artificial intelligence. Vol. 38. No. 2. 2024.
>
> [2] Levy, Deborah, et al. "Seathru-nerf: Neural radiance fields in scattering media." Proceedings of the IEEE/CVF conference on computer vision and pattern recognition. 2023.
>
> [3] Cui, Ziteng, Xuangeng Chu, and Tatsuya Harada. "Luminance-GS: Adapting 3D Gaussian Splatting to Challenging Lighting Conditions with View-Adaptive Curve Adjustment." Proceedings of the Computer Vision and Pattern Recognition Conference. 2025.
>
> [4] Li, Huapeng, et al. "Watersplatting: Fast underwater 3d scene reconstruction using gaussian splatting." arXiv preprint arXiv:2408.08206 (2024).
>
> [5] Barron, Jonathan T., et al. "Zip-nerf: Anti-aliased grid-based neural radiance fields." Proceedings of the IEEE/CVF International Conference on Computer Vision. 2023.
>
> [6] Sun, Cheng, et al. "Sparse Voxels Rasterization: Real-time High-fidelity Radiance Field Rendering." Proceedings of the Computer Vision and Pattern Recognition Conference. 2025.

---

> > ### Comment · Reviewer_sErE · 2025-08-07
> >
> > I thank the authors for their detailed response. After reviewing their response and other reviewers' comments, I have additional follow-up questions:
> >
> > 1. Theoretical basis of RSU in low-light conditions: I remain unclear how RSU, which models scattering due to spatial media, theoretically relates to low-light enhancement. Media properties should remain constant regardless of lighting conditions—normal NeRF models these as near-zero density in typical settings. Does your model attribute higher density values to media in low-light conditions? It seems counterintuitive that scene lighting and medium scattering are not disentangled, yet RSU's performance improvement in low-light settings suggests this coupling. Please provide further theoretical justification.
> >
> > 2. View-dependent density modeling concerns: Following Reviewer sN9j's observations, I question the view-dependent architecture for scattering modeling. While I understand the motivation based on light source and viewing direction relationships, this design makes density modeling view-dependent and transient, potentially introducing additional complexity for the NeRF optimization and producing artifacts. More fundamentally, this approach conflicts with NeRF's volumetric rendering framework, which assumes view-independent density for light transport. How do you reconcile this architectural choice with established volumetric rendering principles?

---

> ### Author Response · Authors · 2025-08-07
>
> Thank you for your follow-up questions. We really appreciate the chance to elaborate on the fundamental theory and objectives behind our model as well as the previous scattering modeling to the broader community.
>
> ### **Q1. Theoretical basis on low-light condition**
>
> In low-light conditions there is no scattering effect, only absorption (Line 139/144, top-right of Fig. 1). The absorptive medium attenuates the radiance that passes through it. The “density” of the medium is actually its attenuation coefficient in light-transport theory, which specifies how rapidly radiance decays per unit distance. In a well-lit scene, vanilla NeRF models each object with a spatially fixed density and learns its color. Placing that same object into a low-light environment should not change its true (physical) density, since the object itself is unchanged. However, from the observer’s viewpoint, low light causes the camera to receive fewer photons, for example, due to short exposure time, so vanilla NeRF treats this as the object emitted fewer radiance, and thus assign lower weights for objects along rays. The object’s property, e.g., color, varies by moving from well-lit to low-light.
>
> To overcome this, our method inserts an explicit absorption field between the object and the camera, applying the attenuation factor to the bypass radiance. With a properly learned absorption field, the underlying object preserves the same density and color it had under well-lit conditions. **We can then restore the density/appearance of the object/scene under normal lighting simply by rendering without the absorption field.** As shown in Fig. 5 and Table 2, this strategy makes objects in low-light scenarios appear same to their well-lit counterparts. (Rendering without absorption field means rendering without media branch, which becomes exactly same as ZipNeRF's pipeline. So the object's density/color in well-lit and low-light are same)
>
> For a clear separation of absorption field and object density, Fig. 3 shows that the absorption field occupies regions along each ray that do not overlap with the object’s volume density.
>
> Our RSU strategy is applied to the sampling stage, which allocate the media sampling intervals. These intervals then participating into the volume rendering stage to predict the attenuation (in low-light)/attenuation + scattering (in scattering) coefficients of the media. In low-light/scattering environments, these media intervals can employ either spatially-variant or spatially-invariant media "density" (coefficients) depending to the use-case.
>
> ### **Q2. View-dependency**
>
> **“Media properties should remain constant regardless of lighting conditions” (Q1) does not hold true.** The “density” of a participating medium is not its mass density but rather its optical density. That is, the coefficient that determines how the medium scatters and absorbs light. For an intuitive example, imagine a clear plastic bottle of water: although the water’s physical mass density stays the same, its apparent opacity and hue change dramatically when you view it under direct sunlight versus viewing it on your table. In vanilla NeRF, all these complex light–matter interactions are implicitly absorbed into the object color term $c^{\mathrm{obj}}$. This works well for opaque objects with negligible scattering or reflection, where the scene remains consistent in 3D space. However, in strongly scattering or reflective environments, as shown by Ref-NeRF (CVPR 22 Best paper candidate), SeaThru-NeRF (CVPR 23), Proposed-T (CVPR 24), **vanilla NeRF fails to maintain multi-view consistency.**
>
> By incorporating our method, we explicitly disentangle view-dependent components (such as scattering and absorption effects) from the underlying scene geometry. This ensures that the object geometry remains stable across viewpoints, while the variable light transport is modeled separately. We would like to remind that the ultimate objective of 3D reconstruction is to recover accurate scene geometry, and our approach, along with previous efforts, achieves this by isolating view-dependent light effects from the consistent spatial structure. In Fig. 4 and Supp Fig 3–5, we clearly demonstrate how object geometry is faithfully recovered in novel views. **We should not revert to vanilla NeRF’s limitations and requesting everything to be solely spatial dependent, as in real-world situations, rays are inherently view-dependent and lighting conditions vary across views, so a robust model must separate these transient effects from the concrete, multi-view consistent geometry.** So this view-dependency for media in our model is not the limitation, but our efforts to ensure the geometry restoration from challenging lighting conditions.

---

> ### Author Response · Authors · 2025-08-08
>
> We apologize for the additional reply and greatly appreciate the time you have devoted to reviewing our paper.
>
> Since Reviewer sN9j has provided valuable references, and we have also found additional materials that clarify the inherent view-dependency of scattering, we would like to elaborate further on the foundational concepts of scattering.
>
> From the material [1] provided by Reviewer sN9j, it is explained that *"phase function describes the angular distribution of scattered radiation at a point."*,  *"phase functions actually define probability distributions for scattering in a particular direction."*, *"The in-scattering portion of the source term is the product of the scattering probability per unit distance $\sigma$"*, and *"the amount of added radiance at a point, which is given by the spherical integral of the product of incident radiance and the phase function."*
>
> The phase function here defines the probability of scattering in a particular direction, which is a source of view-dependency in scattering. In the case of water scattering, authoritative references in ocean optics showing that natural waters exhibit view-dependence. For example:
>
> - Petzold et al (1972). measured volume scattering functions (VSFs) across a range of open-ocean and coastal waters, and showed a transition from nearly flat (isotropic) in clear water to sharply forward-peaked (anisotropic) in turbid/coastal water.
>
> - Jerlov et al. (1968) (also cited as [34] in our paper) classified water types by inherent optical properties and found that molecular (Rayleigh) scattering in the clearest waters (Type I) is near isotropic, and particulate (Mei) scattering in greener waters dominates and becomes highly anisotropic.
>
> - Mobley et al. (1994) (also cited as [56] in our paper) defines the scattering coefficient $b$ and the phase function $\beta(\theta)$, then presents measured VSFs for pure water (nearly isotropic) vs particulate-dominated water (forward-peaked).
>
> **Therefore, the view-dependency of scattering is well-established in the scientific literature.** Re-NeRF offers a solution to model such view-dependent phase functions in reflective scenarios (no media in this case, BRDF only). In our work and this line of research, we aim to separate this view-dependent scattering component from the multi-view consistent scene geometry, thereby addressing the limitations of vanilla NeRF discussed above.
>
> ### Reference
>
> [1] Volume Scattering Processes
>
> [2] T. J. Petzold, Volume Scattering Functions for Selected Ocean Waters, 1972.
>
> [3] N. G. Jerlov, Irradiance Optical Classification, 1968.
>
> [4] C. D. Mobley, Light and Water: Radiative Transfer in Natural Waters, 1994.

---

> > ### Comment · Reviewer_sErE · 2025-08-09
> >
> > I thank the authors for their detailed response. My concerns have been addressed, especially with using RSU for low-light settings. I understood the authors' comments as that the density value predicted by NeRF is relative to the lighting at hand - in which case, in low-light settings where lighting is dimmed, the relative (optical) density of the medium has stronger effect on the optimization / reconstruction process. I will maintain my score.

---

> > > ### Author Response · Authors · 2025-08-09
> > >
> > > Thank you very much for your response and for taking time reviewing our paper!

---

### Note · Authors · 2025-08-12

## Rebuttal Summary

The paper proposes a physically grounded NeRF model with 3D spatial modeling for media conditions such as haze/underwater. It introduces a reverse-stratified upsampling (RSU) that samples media points between the camera and objects. By introducing a virtual media field that absorbs light, this model achieves SOTA on competitive low-light restoration tasks. Moreover, extending the optical attenuation law to the vertical direction enables unique capabilities such as estimating water depth and producing plausible underwater/haze rendering.

The paper sparked substantial discussion among the reviewers, particularly interested in its theoretical assumptions, the newly proposed 3D spatial media modeling, and the effectiveness of RSU as its solution.

**Parallel ray assumption in water scattering:** Reviewers sN9j, TDRQ were satisfied with the rebuttal’s explanation that this assumption, adopted from the RUIF model, provides a tractable, inclination-free solution with constant backscatter radiance and vertical depth per ray.

**Effectiveness of RSU:** Reviewers sErE, teTt acknowledged the rebuttal’s notion of Fig.6, Tab.3, and Supp. Sec.F/Fig.1, showing RSU’s benefits in preserving plausible media estimation. Extensive discussion with Reviewer sN9j also established a theoretical foundation for RSU’s gradient effect

**RSU’s computational tradeoffs:** Reviewers sErE, teTt confirmed RSU incurs a manageable ~0.3 hour training overhead compared to ZipNeRF while delivering superior performance in challenging media condition

Reviewers TDRQ and sN9j suggested a closed-form alternative to RSU. Authors found removing the spatial media branch and adopting this form can reduce ~4 GB memory but can cause sampling collapse. TDRQ further proposed adding regularization in future work to mitigate this issue

**View-dependence of scattering:** Reviewers sErE, sN9j agreed it was acceptable to follow the previous paradigm to use view-dependent encoding. Reviewer N9j suspected view-dependence is not only caused by phase functions.

**Potential inconsistent vertical depth:** Reviewer sN9j suggested using position-only depth prediction and explicit phase function. Rebuttal experiment indicated this is a promising direction for addressing common scattering issues and will be noted in future work

Authors agree with TDRQ that naming has space to discuss. The rebuttal discussion will be added to improve the paper. The data will be released following TDRQ’s reminder.

---

### Decision · Program_Chairs · 2025-09-17

**Decision:**

Accept (poster)

**Comment:**

The paper proposes a NeRF variant that considers media interactions along the rays. This enables modeling 3D scenes in underwater environments, in foggy conditions and in low light conditions. The paper demonstrates more physically plausible and separable reconstructions compared to baselines that are designed separately for each task (underwater NeRF, hazy/foggy NeRF, and low-light NeRF).

The reviewers agreed that the paper is well written and that the comprehensive presentation of the method provides useful insights to the community.

There were a few issues that were discussed during the discussion period, including:

1. The level of contribution related to similar prior work.

2.  The RSU (Reverse-stratified Upsampling) mechanism and the flexibility in media parameters across the media.

3. Should media densities be view dependent.

4. Assumptions made on the modeling of downwelling distance, its physical interpretation and its dependence on viewing angle.

5. The $I^2$ in the title and its importance in the method.

Most of the concerns including points 1, 2 and 3 were resolved. Points 4 and 5 remain in some level of disagreement.

Based on the authors response most reviewers recommended acceptance, and the sole reviewer that was leaning for rejection agreed to accept the paper provided the authors address consistency in the downwelling depth analysis (point 4 above) in the final revision.